# Natural proteome diversity links aneuploidy tolerance to protein turnover

Julia Muenzner[1,15], Pauline Trébulle[2,3,15], Federica Agostini[1], Henrik Zauber[4], Christoph B. Messner[2,5], Martin Steger[6,14], Christiane Kilian[1], Kate Lau[1], Natalie Barthel[1], Andrea Lehmann[1], Kathrin Textoris-Taube[1,7], Elodie Caudal[8], Anna-Sophia Egger[2], Fatma Amari[1,7], Matteo De Chiara[9], Vadim Demichev[1,2], Toni I. Gossmann[10], Michael Mülleder[7], Gianni Liti[9], Joseph Schacherer[8,11], Matthias Selbach[4], Judith Berman[12✉] & Markus Ralser[1,2,3,13✉]

Accessing the natural genetic diversity of species unveils hidden genetic traits, clarifies gene functions and allows the generalizability of laboratory findings to be assessed. One notable discovery made in natural isolates of *Saccharomyces cerevisiae* is that aneuploidy—an imbalance in chromosome copy numbers—is frequent[1,2] (around 20%), which seems to contradict the substantial fitness costs and transient nature of aneuploidy when it is engineered in the laboratory[3–5]. Here we generate a proteomic resource and merge it with genomic[1] and transcriptomic[6] data for 796 euploid and aneuploid natural isolates. We find that natural and lab-generated aneuploids differ specifically at the proteome. In lab-generated aneuploids, some proteins—especially subunits of protein complexes—show reduced expression, but the overall protein levels correspond to the aneuploid gene dosage. By contrast, in natural isolates, more than 70% of proteins encoded on aneuploid chromosomes are dosage compensated, and average protein levels are shifted towards the euploid state chromosome-wide. At the molecular level, we detect an induction of structural components of the proteasome, increased levels of ubiquitination, and reveal an interdependency of protein turnover rates and attenuation. Our study thus highlights the role of protein turnover in mediating aneuploidy tolerance, and shows the utility of exploiting the natural diversity of species to attain generalizable molecular insights into complex biological processes.

Assessing the natural diversity of species has the potential to uncover new genetic traits and gene functions. Moreover, natural isolate collections are increasingly powerful tools for probing reproducibility and the generalizability of experimental findings. In budding yeast, sequencing collections of natural isolates[1,2,7–9] revealed that around 20% of the isolates exhibit stable aneuploidies—imbalances in chromosome copy numbers. This finding contrasts with earlier studies of laboratory strains, in which aneuploidies were unstable and exacted fitness costs[3,5,10]. The high natural prevalence implies that aneuploidies are beneficial, at least under some conditions, and that natural isolates have strategies to mitigate fitness costs[1,4]. Indeed, aneuploidies can increase stress resistance and drug tolerance in yeast[11–16], virulence and immune escape in protists[17] and malignancy, invasiveness and drug tolerance in human cancer cells[18–22]. However, how cells mediate aneuploidy tolerance is not well understood. One potential mechanism is dosage compensation—the attenuation of altered gene dosage. Dosage compensation of either individual genes or whole chromosomes at the proteome level has been observed in tissue-cultured mammalian cancer cells and in pathogens such as *Leishmania donovani*[23–31]. However, the diverging results obtained with two collections of lab-engineered aneuploid yeast strains have triggered a debate about whether dosage compensation is associated with aneuploidy across species, and have raised questions about which proteins are dosage compensated and through which mechanisms[3,5,10,32]. One of these collections consists of 'synthetic' disomes, in which each haploid strain carries a duplication of one of the 16 *S. cerevisiae* chromosomes, maintained under selection[3]. In these disomes, the gene dosage from a chromosome duplication yielded, on average, a twofold increase in mRNA and protein levels, but a fraction of proteins—namely, components of protein complexes—were attenuated specifically at the proteome level[3,10]. The second collection of strains was produced by inducing the meiosis of triploid or pentaploid strains and subsequent selection for stable aneuploid progeny[5].

[1]Department of Biochemistry, Charité Universitätsmedizin, Berlin, Germany. [2]Molecular Biology of Metabolism Laboratory, Francis Crick Institute, London, UK. [3]Centre for Human Genetics, Nuffield Department of Medicine, University of Oxford, Oxford, UK. [4]Max Delbrück Center for Molecular Medicine, Berlin, Germany. [5]Precision Proteomics Center, Swiss Institute of Allergy and Asthma Research (SIAF), University of Zurich, Davos, Switzerland. [6]Evotec (München), Martinsried, Germany. [7]Core Facility High-Throughput Mass Spectrometry, Charité Universitätsmedizin, Berlin, Germany. [8]Université de Strasbourg, CNRS GMGM UMR 7156, Strasbourg, France. [9]Université Côte d'Azur, CNRS, INSERM, IRCAN, Nice, France. [10]Computational Systems Biology, Faculty of Biochemical and Chemical Engineering, TU Dortmund University, Dortmund, Germany. [11]Institut Universitaire de France (IUF), Paris, France. [12]Shmunis School of Biomedical and Cancer Research, George S. Wise Faculty of Life Sciences, Tel Aviv University, Ramat Aviv, Israel. [13]Max Planck Institute for Molecular Genetics, Berlin, Germany. [14]Present address: NEOsphere Biotechnologies, Martinsried, Germany. [15]These authors contributed equally: Julia Muenzner, Pauline Trébulle. ✉e-mail: jberman@tauex.tau.ac.il; markus.ralser@charite.de

Aneuploidies in this collection were more stable, but dosage compensation was minimal, and not enriched for protein-complex subunits[5]. Moreover, the two collections differ in that the *SSD1* gene, which has been linked to aneuploidy tolerance, is truncated in the disomic collection[33]. Finally, contradictory aneuploidy-associated transcriptional signatures have been reported for lab-engineered aneuploid strains, which suggests that aneuploids respond dissimilarly to the presence of aneuploid chromosomes[3,10,34,35].

Lab-generated synthetic aneuploids represent an early state of acquired aneuploidy; that is, they have not undergone long-term selection with the aneuploid chromosome. Dosage compensation might thus be different in natural isolates, in which most aneuploids are assumed to have been present over longer timescales[36]. The transcriptomes of natural aneuploids, however, resembled lab-generated aneuploids: transcripts that cause toxicity when overexpressed are dosage compensated, but the average chromosome-wide gene dosage is not attenuated and follows the chromosome copy number change[4,37,38].

On the other hand, although dosage compensation is linked mainly with protein levels, the proteomes of natural isolates have yet to be investigated. Here we developed cell growth, sample preparation, data acquisition and data preprocessing strategies to generate precise proteomes for 796 environmental and domesticated natural isolates of *S. cerevisiae*. By integrating these proteomes, isolate by isolate, with genomes[1] and transcriptomes[6], we obtained a multi-omic dataset covering 613 natural isolates, including 15.5% aneuploids. Furthermore, for eight isolates, we generated a ubiquitinome dataset[39], and for 55 isolates, we measured protein turnover using pulse labelling with stable amino acid isotopes[40]. We found that aneuploid proteomes differ markedly between natural and lab-engineered yeast. Natural isolates attenuate the aneuploid gene dosage broadly, affecting about 70% of proteins encoded on aneuploid chromosomes, across protein functions and on a chromosome-wide level. Attenuation of the proteome in natural isolates can be attributed to the induction of the ubiquitin–proteasome system (UPS) and to increases in overall rates of protein turnover.

## Multi-omic data on natural yeast

We produced a multi-omic dataset for natural isolates by generating proteomes for isolates that were obtained globally from environmental and industrial niches (Supplementary Table 1), and integrating them with genomes[1] and transcriptomes[6] of the same isolates. For the proteomes, isolates were arrayed in 96-well plates and cultivated in a minimal synthetic medium; 933 strains reached sufficient biomass (Supplementary Table 2). We generated proteomes by adapting a SWATH mass spectrometry (SWATH-MS)-based high-throughput proteomics pipeline[41,42] and by developing a computational preprocessing pipeline that accounts for the genetic diversity of natural isolates (Fig. 1a and Extended Data Fig. 1a). After extensive quality filtering, we retained 7,946 precursors that quantified 1,576 proteins across 796 isolates (Fig. 1a, Extended Data Fig. 1b,c and Supplementary Table 3). With less than 4% missing values, the proteins were quantified with a coefficient of variation (CV) of 16% across the 77 control samples, and the natural diversity was reflected in a CV of 32.8% across the isolates (Extended Data Fig. 1d). Processing of the disomic strain collection[3] over the same pipeline yielded quantities for 1,377 proteins at high completeness and precision (less than 2.3% missing values; median CV of 9.5% within replicates, and 26.7% median CV across samples; Extended Data Fig. 1e,f and Supplementary Table 4).

For integrating the proteomes with genomes and transcriptomes, we included isolates that either were euploid or had whole-chromosome aneuploidies[1], and excluded all isolates with potential inconsistencies within or between the genome and transcriptome (Extended Data Fig. 1g, Supplementary Tables 5 and 6 and Methods). The integrated dataset represents 613 haploid to pentaploid natural isolates, including

95 with at least one aneuploid chromosome (Fig. 1b and Extended Data Fig. 1h). Aneuploidy was seen for all chromosomes (Extended Data Fig. 1i). Chromosome gains were more common (88.4%) than losses (11.6%). Chromosomes 1, 9 and 3 were most frequently aneuploid, whereas aneuploidy was rare for chromosomes 2, 4, 10 and 13 (Fig. 1b and Extended Data Fig. 1i). The low aneuploidy frequency of these chromosomes could, at least partially, be linked to their large size[36]. Furthermore, aneuploidy was more frequent in isolates with higher basal ploidy, and 26 strains had complex aneuploidies (Fig. 1b, Extended Data Fig. 1h–j and Supplementary Table 7).

## Dosage compensation in natural aneuploids

Gene-by-gene level analysis of transcript and protein levels relative to chromosome copy number changes revealed a high degree of dosage compensation at the proteome level in natural isolates. For example, Gvp36 (a Golgi vesicle protein), Ccr4 (a component of the CCR4–NOT complex) and Rps20 (a component of the small ribosomal subunit) all had minimal attenuation at the mRNA level (Fig. 2a; mRNA attenuation slopes of 0.9, 0.98 and 1.11, respectively; Supplementary Table 8). At the protein level, Gvp36 was not attenuated, but Ccr4 and Rps20 were partially and strongly attenuated, respectively (Fig. 2a; protein attenuation slopes of 0.97, 0.69 and 0.12, respectively; Supplementary Table 8). In total, 70.5% of all proteins encoded by genes on the aneuploid chromosomes were dosage compensated at the proteome level (Fig. 2b). Consistently, the correlation between mRNA- and protein-level attenuation was weak (Spearman's $\rho = 0.20$, $P = 4.9 \times 10^{-9}$; Extended Data Fig. 2a). The fraction of attenuated proteins was consistently higher than the fraction of attenuated mRNAs (Fig. 2b), and the median attenuation slope across all genes expressed on aneuploid chromosomes was markedly stronger for proteins than for mRNAs (0.65 versus 0.92; Extended Data Fig. 2b).

By contrast, for the disome collection, the fraction of attenuated proteins was lower than it was in the natural isolates at any given threshold (Fig. 2b and Supplementary Table 9), whereas median attenuation slopes across all genes for proteins and mRNAs were similar (0.87 versus. 0.94; Extended Data Fig. 2b). Furthermore, when comparing synthetic disomic strains ($1n + 1$) to natural isolates with chromosome duplications ($1n+1$ or $2n+2$; Extended Data Fig. 2c), most proteins attenuated in disome strains were attenuated in natural isolates (around 69%), whereas proteins that were not attenuated in the synthetic aneuploids were also attenuated in the natural disomes (Extended Data Fig. 2d,e). Thus, attenuation is specific to the proteome, and is more prevalent in natural isolates than in lab strains, both qualitatively and quantitatively.

Previous work on the synthetic disomes and in mammalian cell lines explained dosage compensation with the attenuation of surplus subunits of protein complexes[10,25]. In natural isolates, functional analysis of attenuated proteins found that 44% of attenuated proteins are components of macromolecular complexes (Extended Data Fig. 2f). However, most (63%) proteins that are not known to be macromolecular-complex components are also attenuated in natural isolates (Fig. 2c,d). For instance, proteins involved in metabolic pathways such as the TCA cycle, glycolysis and gluconeogenesis are attenuated (Supplementary Table 10). By contrast, only some metabolic processes associated with growth were not attenuated, including the glutathione, pyrimidine, pantothenate and CoA, and glycerophospholipid pathways, as well as the alanine, aspartate and glutamate pathway (Supplementary Table 10). Thus, we asked whether attenuation is determined by other factors. We found that, in addition to protein-complex membership, attenuation at the protein level correlated with the number of potential ubiquitination sites. The significance of the observed relationship was not robust to multiple testing correction in our dataset, but has been reported previously[26] (Fig. 2e and Extended Data Fig. 3). Also, consistent with the importance of protein complexes, the numbers

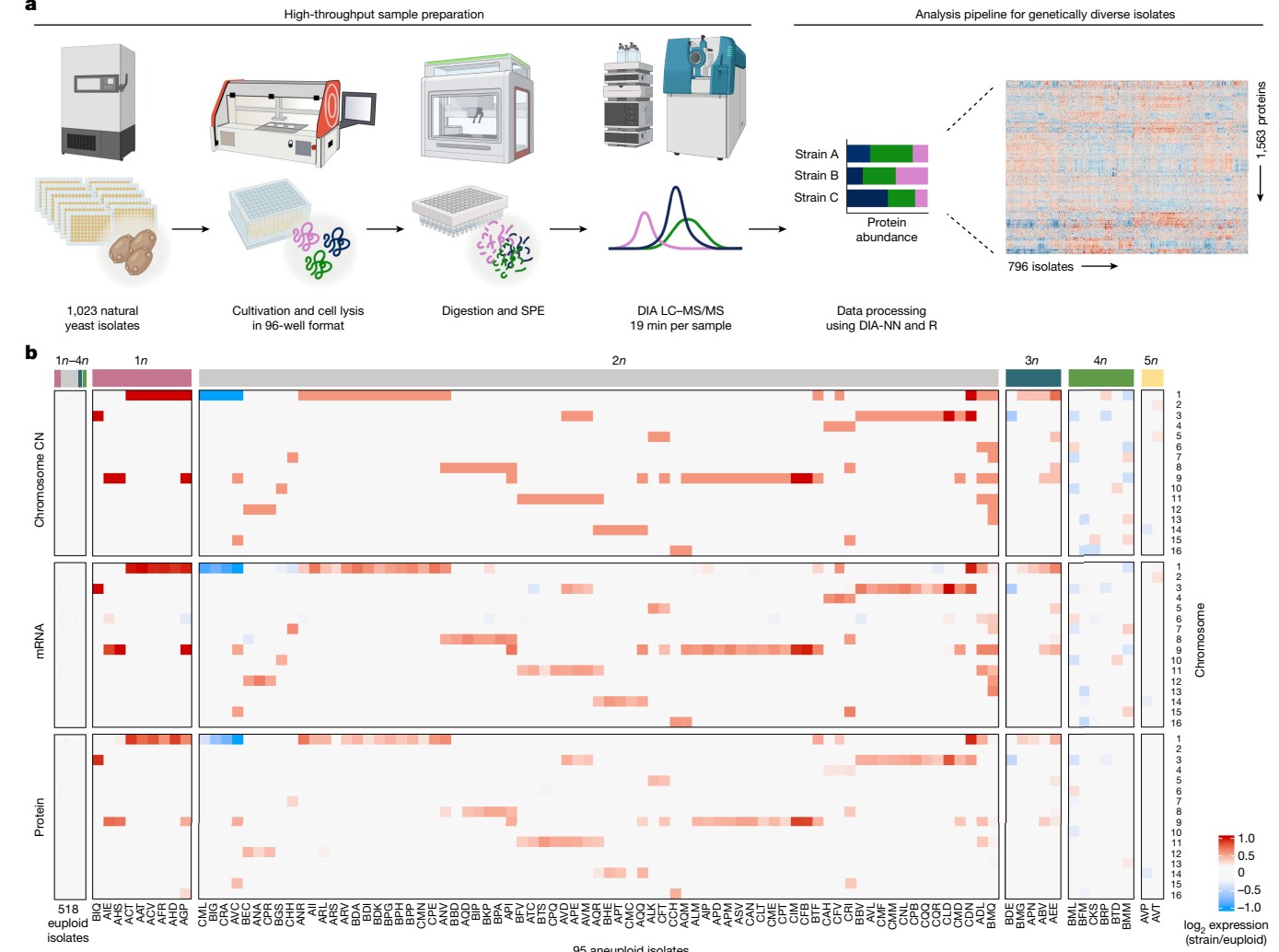

**Fig. 1 | High-throughput proteomics pipeline and assembly of a cross-omics dataset for studying aneuploidy in natural yeast isolates. a**, *S. cerevisiae* isolates were cultivated in synthetic minimal medium in 96-well format. Cells were collected by centrifugation at mid-log phase and lysed by bead beating under denaturing conditions. The lysate was then treated with reducing and alkylating reagents and digested with trypsin. The resulting peptides were desalted by solid-phase extraction (SPE) and analysed by liquid chromatography–tandem mass spectrometry (LC–MS/MS) in data-independent acquisition (DIA) mode using SWATH-MS. Data were integrated using DIA-NN. **b**, Integrated dataset of 613 natural *S. cerevisiae* isolates. Relative chromosome copy number (CN), relative median mRNA levels and relative median protein abundances per chromosome between isolates and euploid reference (log$_2$ ratios strain/euploid) are shown. The median across all euploid isolates was used as the reference for ratio calculations. Isolate ploidy is indicated at the top of the heat map.

of protein–protein interactions showed a similar trend. Moreover, proteins that exhibited lower variability in abundance across euploid isolates were more likely to be attenuated (Fig. 2e and Extended Data Fig. 2g,h). Homologues of proteins that are non-exponentially degraded in mammalian cells[25] were also more likely to be attenuated (Extended Data Fig. 2i).

By analysing effect sizes across isolates, we compared chromosome-wide dosage compensation in the natural isolates and the disomes (Extended Data Fig. 4a–c and Supplementary Table 11). As in the synthetic disomes and as reported for a small selection of natural isolates[3,4,10,43], the relative transcript abundances of aneuploid chromosomes centred around the relative change in copy number. In the disomes, relative protein levels measured here also centred around the relative copy number increase, with the attenuation of specific proteins (for example, complex subunits) having a shouldered distribution, confirming previous studies[10] (Extended Data Fig. 4a). By contrast, for the natural isolates, the mean relative protein distribution was shifted towards the euploid state, indicating chromosome-wide dosage compensation (Extended Data Fig. 4b,

Supplementary Table 11 and Supplementary Fig. 1). For example, disome 9 and natural isolate AHS have the same karyotype and similar average mRNA distributions, but only in the natural isolate was the median relative protein abundance shifted towards the euploid state (Fig. 3a). Across all isolates, absolute attenuation was stronger with lower overall ploidy and for larger chromosomes, but also diverged widely between isolates (Fig. 3b, Extended Data Fig. 4d–f and Supplementary Tables 11 and 12). For example, relative protein levels in diploid isolate CFV were attenuated by 58%, whereas in triploid isolate ABV they were attenuated by only 14% (Fig. 3c and Supplementary Table 12). Chromosome-wide attenuation was significant for all chromosome gains and losses (Extended Data Fig. 4a–c and Supplementary Table 11, except for 3*n*−1 chromosomes), as were the differences between the proteome and the transcriptome (Extended Data Fig. 4a,b). These differences were robust across different growth stages (Extended Data Fig. 4g,h and Supplementary Table 13). Quantitatively, relative protein expression in the natural isolates was attenuated by an average of 25% across relative chromosome copy number changes (Fig. 3b).

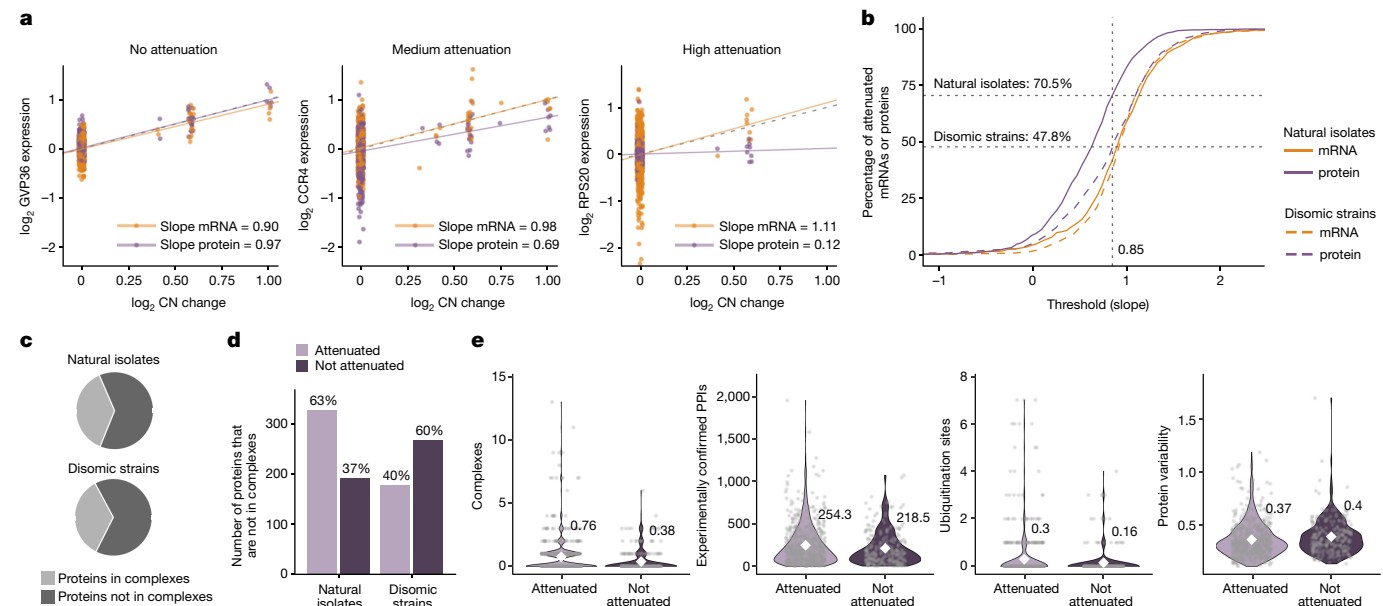

**Fig. 2 | Gene-by-gene quantification of dosage compensation. a**, Linear regression of relative mRNA and protein levels against relative copy number changes for three exemplary genes with different extents of protein-level dosage compensation. Each point shows the log₂ expression when expressed from an aneuploid chromosome in an isolate (GVP36: chr. 9, relative CN change > 0 in 24 points; CCR4: chr. 1, CN change > 0 in 29 points; RPS20: chr. 8, CN change > 0 in 9 points). The expected behaviour of a non-buffered gene ($y = x$, dotted grey line) is shown. **b**, Cumulative distribution analysis for natural isolates and disomic strains, comparing the fraction of genes attenuated at the mRNA or protein level on the basis of the distribution slopes. The vertical dotted grey line denotes an analysis threshold of 0.85, and the horizontal dotted grey lines and numbers indicate the fraction of proteins attenuated at this threshold. **c**, Proportion of genes in natural isolates and in synthetic disomes that are part of macromolecular complexes. **d**, Proportion of proteins that are not part of macromolecular complexes and attenuated or not in natural isolates or synthetic disomes. For **a**–**d**, the regression analysis was performed separately for natural isolates and disomic strains (827 and 680 proteins included in the analysis, respectively). **e**, For proteins (dots) that are either attenuated ($n = 583$) or not ($n = 244$) when expressed from aneuploid chromosomes of natural isolates, comparison of the number of complexes a protein is part of ($P = 4.1 \times 10^{-8}$ (without adjustment); $P = 9.3 \times 10^{-7}$ (with adjustment)), the number of experimentally confirmed protein–protein interactions (PPIs) ($P = 0.056$; $P = 0.20$), the number of ubiquitination sites per protein obtained from Uniprot (integrating both experimental and computational resources; $P = 0.023$; $P = 0.13$) and the general variability in protein abundance ($P = 4.4 \times 10^{-3}$; $P = 0.034$). White diamonds represent means. $P$ values refer to significance testing by two-sample, two-sided Wilcoxon tests without or with Benjamini-Hochberg adjustment.

## UPS activation across natural aneuploids

Aneuploidy affects gene expression in *trans*; that is, on euploid chromosomes. Three *trans* expression signatures potentially provide insights into the physiological responses to aneuploidy: the environmental stress response (ESR)[3,35,44], the aneuploidy-associated protein signature (APS)[10] and the common aneuploidy gene expression (CAGE)[34]. However, because these signatures differed between collections of aneuploid lab strains, they were fiercely debated[10,34,35]. Notably, the natural isolates exhibited transcriptional signatures that resembled the ESR, APS and CAGE signatures. However, the direction of up- or downregulation was highly isolate dependent, with clusters of isolates showing completely opposite patterns of regulation (Fig. 4a and Supplementary Fig. 2). Moreover, these transcriptional signatures were largely mitigated at the proteome level. Thus, the presence of ESR, APS and CAGE signatures differs in different isolates, and these signatures are not necessarily translated to the proteome.

We then performed gene set enrichment analysis (GSEA) to ask whether other *trans* signals were consistently seen across the proteomes of natural isolates. The KEGG gene set 'Proteasome' was highly enriched among the upregulated proteins across all natural aneuploids relative to all euploids, and was not enriched in the proteomes of the disomes (Fig. 4b,c and Extended Data Fig. 5a). Furthermore, 'Proteasome' was the only enriched KEGG term that was related neither to growth-related metabolism, which is also slightly altered in the natural aneuploids[1,36], nor to transcription-associated processes that are plausibly associated with aneuploidy owing to their increased need for DNA and RNA synthesis. The enrichment of the KEGG term was due to upregulation of structural components of the proteasome, such as the core and regulatory particle (Fig. 4c, Extended Data Fig. 5b,c and Supplementary Table 14), and was seen across ploidies (Extended Data Fig. 5d,e), although the effect sizes varied between isolates (Extended Data Fig. 6a). The degree of enrichment for proteasome components did not correlate with the degree of aneuploidy (Extended Data Fig. 6b), and did not seem to be driven by the transcription factor Rpn4, a transcriptional regulator of proteasome components in lab strains (Extended Data Fig. 6c–j). Thus, proteasome components are induced in natural aneuploids but not in the lab-engineered disomes.

## Protein turnover and dosage compensation

We hypothesized that the proteasome could mediate dosage compensation by accelerating protein degradation and thus protein turnover. Ubiquitinomics experiments using K-GG remnant peptide profiling[39] revealed an increased total amount of K-GG-modified peptides in proteins encoded on aneuploid chromosomes (Extended Data Fig. 7a and Supplementary Table 15). Ubiquitination levels correlated with increased gene dosage, but were not found super-stoichiometrically on proteins encoded on the aneuploid chromosome (Extended Data Fig. 7b). Next, we analysed the proteomes of around 4,800 deletion mutants in the lab strain BY4741, of which more than 100 had acquired aneuploidies[45]. When encoded on an aneuploid chromosome, proteins with a faster turnover[46] were more strongly dosage compensated than were proteins with a slower turnover (Fig. 5a).

We asked whether we could adapt dynamic SILAC[40] to generate protein turnover data for natural isolates. We exploited our previous

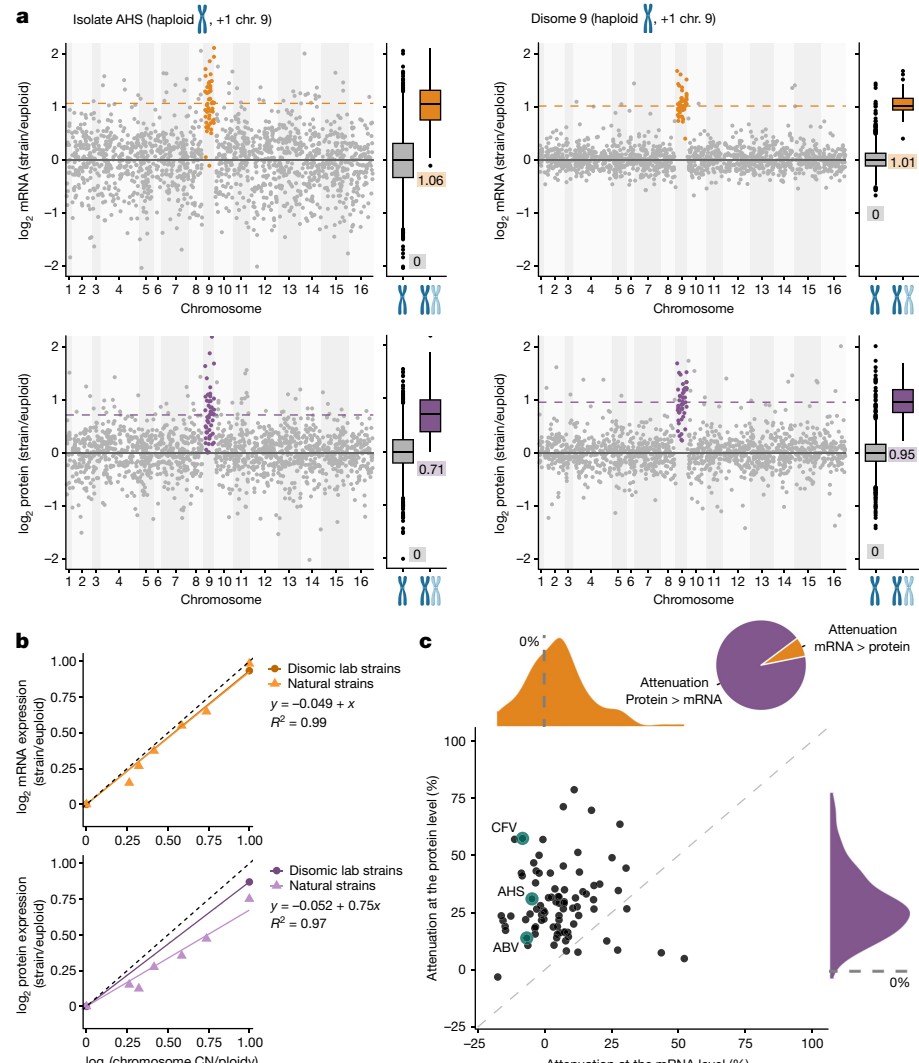

**Fig. 3 | Dosage compensation across isolates. a**, Comparison of relative mRNA and protein expression between natural isolate AHS and a disomic lab strain, both disomic for chromosome 9. Gene-by-gene $\log_2$ mRNA or protein ratios are shown, sorted by chromosomal location. Genes located on aneuploid chromosomes are coloured. The solid grey line marks 0, and dashed coloured lines indicate the medians of the aneuploid mRNA or protein expression distributions, respectively. Box plots show the distributions of $\log_2$ mRNA or protein ratios for all genes encoded on euploid or duplicated chromosomes (AHS: $n = 1,513/48$ euploid/aneuploid; disome 9: $n = 1,322/45$ euploid/ aneuploid). In box plots, the centre marks the median, hinges mark the 25th and 75th percentiles and whiskers show all values that, at maximum, fall within 1.5 times the interquartile range; the median of the distributions is shown below the boxes. Only $\log_2$ ratios between −2 and 2 are shown to improve readability. **b**, Quantification of relative dosage compensation at the mRNA level (top) and

at the protein level (bottom) for natural isolates and disomic lab strains. The medians of the $\log_2$ mRNA and $\log_2$ protein distributions in Extended Data Fig. 4a,b are plotted against the relative chromosome copy number change. Linear regressions for mRNA (orange) and protein (purple) levels in natural isolates and disomic lab strains are shown. Linear models and $R_2$ values are shown for natural isolates. The dotted black line indicates the expected relative expression levels under no dosage compensation ($y = x$). **c**, Scatter plot comparing the isolate-wise extent of attenuation at the mRNA and protein level. Isolates AHS, CFV and ABV are highlighted. Distributions of mRNA-level (orange) and protein-level (purple) buffering across all aneuploid isolates are shown. The pie chart shows the number of isolates that are buffered more strongly at the protein or at the mRNA level. Isolates with complex aneuploidies of different relative copy number changes, as well as isolates that probably reverted to the euploid state, were excluded.

finding that prototrophic yeasts switch from self-synthesis to uptake for lysine[47], and confirmed the uptake of lysine using stable-isotope-labelled lysine followed by liquid chromatography–selective reaction monitoring (Extended Data Fig. 7c–e). Natural isolates consumed lysine in preference to endogenous synthesis, indicating a full switch to lysine uptake. Indeed, most natural isolates had a higher lysine uptake than did a lysine-auxotrophic lab strain. Next, we conducted dynamic SILAC experiments on 48 haploid or diploid aneuploid isolates, including all diploid isolates with a single chromosome gain (trisomic strains), and 12 haploid or diploid euploid isolates with similar ranges of growth rates (Extended Data Fig. 8a and Supplementary Table 16). SILAC ratios

were obtained for 2,400-4,800 peptides per time point (Extended Data Fig. 8b), providing information about the turnover rates for a median number of around 1,100 proteins across 55 isolates (Fig. 5b, Extended Data Fig. 8c and Supplementary Table 17).

Natural isolates exhibited a wide range of turnover rates. Isolates with a high degree of attenuation had significantly higher rates of turnover (Fig. 5c). Furthermore, the rates of turnover correlated with the degree of attenuation across the isolates (Fig. 5d). When quantile-normalized rates of turnover were analysed as a function of whether the protein was expressed from a euploid or an aneuploid chromosome, we found that proteins expressed from euploid chromosomes, in either euploid

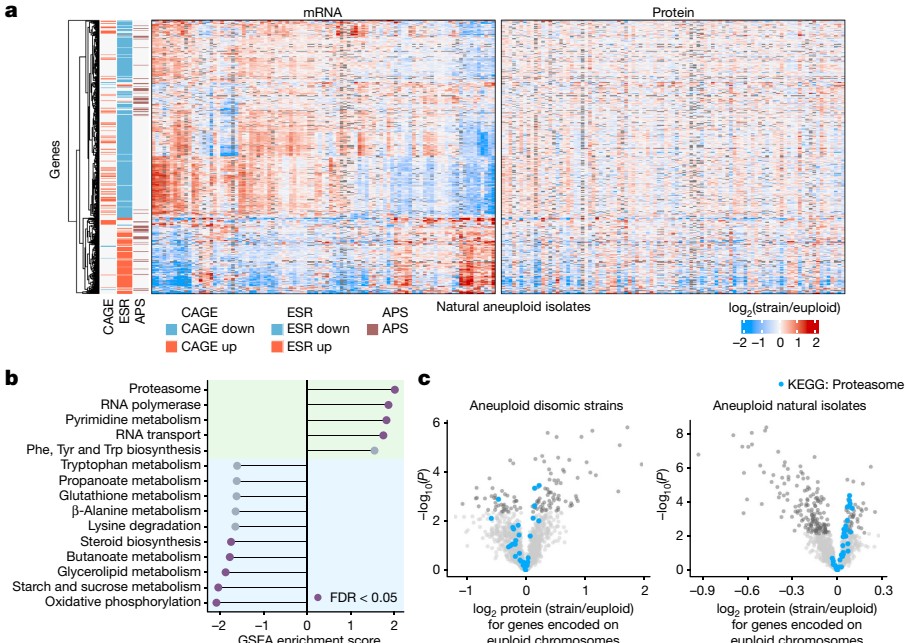

**Fig. 4 | Analysis of the *trans* expression response in natural aneuploid isolates. a**, mRNA and protein *trans* expression (log$_2$ isolate/euploid) of genes previously implicated in the global response to aneuploidy across natural aneuploid isolates (*n* = 95). Genes annotated as CAGE genes[34], ESR genes[3,44] and APS genes[10] are clustered to the left of the heat maps, with the direction of the regulation described in the reference papers indicated in red (up) or blue (down). Genes that are located on aneuploid chromosomes in a respective isolate are omitted from *trans* expression analyses and are therefore shown in grey. **b**, GSEA of median log$_2$ protein expression ratios (isolate/euploid) for genes encoded on all euploid chromosomes across aneuploid natural isolates

(*n* = 95, genes in *trans* of aneuploid chromosomes). Statistically significant enrichment scores (false discovery rate (FDR) < 0.05) are coloured in purple. The green background highlights gene sets with positive enrichment scores, the blue one gene sets with negative enrichment scores. **c**, Volcano plots for natural isolates (*n* = 95) and disomic strains (*n* = 9, biological triplicates) showing the results of one-sample, two-sided *t*-tests comparing the mean log$_2$ protein ratios to $\mu$ = 0. Proteins with statistically significant differential expression after multiple hypothesis correction (Benjamini–Hochberg) are coloured in dark grey. Structural components of the proteasome are highlighted in blue.

or aneuploid strains, exhibited comparable turnover rates (Fig. 5e). By contrast, when a protein was expressed from an aneuploid chromosome, its turnover rate was more frequently higher compared with when it was expressed from a euploid chromosome in either euploid or aneuploid strains (Fig. 5e). This suggests that, in natural isolates, proteins expressed from aneuploid chromosomes are often degraded more rapidly than they are when expressed from euploid chromosomes.

The relationship between protein abundance and the overall protein turnover of isolates was different for many proteins when they were expressed from an aneuploid rather than a euploid chromosome (Fig. 5f and Extended Data Fig. 8d) For example, in euploid isolates, the abundance of Age2 did not correlate with isolate turnover; however, when Age2 was expressed from an aneuploid chromosome, its protein levels were more attenuated with increasing turnover (Extended Data Fig. 8d). Such negative correlations between protein levels and turnover rates were observed across genes expressed on aneuploid chromosomes (Fig. 5f), and were particularly evident for subunits of protein complexes (Fig. 5g). By contrast, the correlation coefficient distribution for relative protein abundances and isolate-wise turnover rates centred around 0 for proteins expressed from euploid chromosomes of euploid isolates (Fig. 5f). Notably, proteasome components were enriched among the genes with positive correlation coefficients across euploid isolates (Extended Data Fig. 8e). This is consistent with increased protein turnover being—at least partially—mediated through an increase in proteasome abundance.

## Discussion

The advent and success of next-generation sequencing technologies have greatly facilitated molecular biology investigations beyond traditional model organisms, and increasingly enable the generalizability of laboratory findings to be tested[1,48,49]. However, investigations with natural strains had not yet been extended to the proteome. A key finding from libraries of yeast natural isolates was the frequent occurrence of aneuploidy[1,2,4]. This contrasts with lab strains, in which aneuploidies are often transient and impose fitness costs[3,12]. Here we adapted high-throughput and data-independent acquisition proteomics[41,42] to quantify the proteomes of hundreds of natural *S. cerevisiae* isolates at high precision, and with a low number of missing values. To capture additional proteomic properties for the isolates, we generated ubiquitination profiles and protein turnover data using dynamic SILAC[40]. Integrating these proteomic datasets with genomes[1] and transcriptomes[6] generated a large, systematic, openly available community resource, which is applicable, for instance, to questions related to protein expression dynamics, smaller copy number variations, the genetic basis of protein expression and gene function.

By comparing natural aneuploids with natural euploids, and comparing the natural aneuploids with lab-generated aneuploids, we report three major observations. First, in agreement with previous literature[3,4,10,37,38,43,50], at the transcriptional level, chromosome-wide average mRNA expression levels are largely unaltered, and overall transcript abundance follows chromosome copy number in all yeast isolates, with the caveat that some mRNAs diverge from a clear one-to-one relationship with the DNA copy number. Moreover, our results provide a global perspective on the effects of changes in gene expression on euploid chromosomes. Here, three *trans* expression signatures (ESR, APS and CAGE) were considered common responses to aneuploidy[3,34,44]. Although all three transcription signatures are evident amongst the aneuploids, they are sometimes up- and sometimes downregulated, and are dampened considerably at the proteome. Of

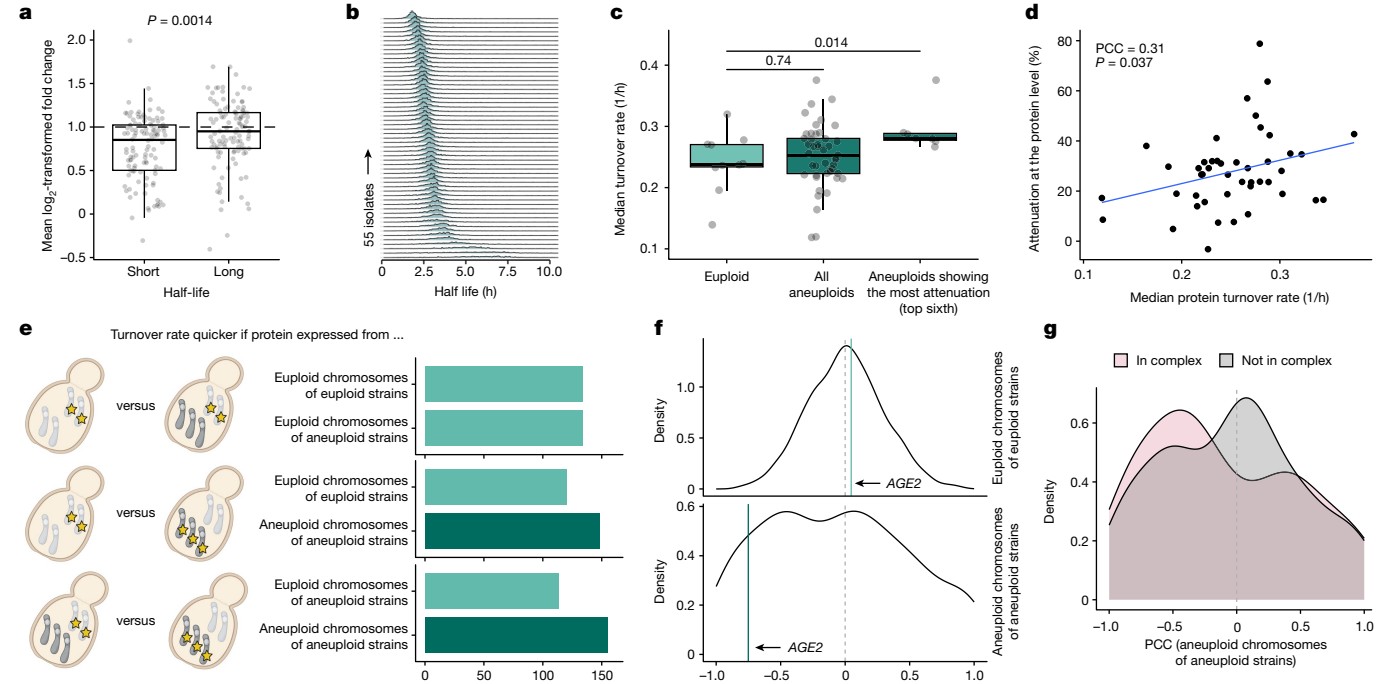

**Fig. 5 | Increased protein turnover in natural isolates is linked to dosage compensation. a**, Changes in protein abundance after inadvertent chromosomal duplications in aneuploid strains of the yeast deletion collection[45] for proteins with short and long half-lives. Fold changes are defined as ratios between protein abundances and the median abundances of the respective protein across all strains. Long and short half-lives are defined as being >75% and <25% quantile (*n* = 110), respectively. *P* values were obtained by two-sided *t*-test. **b**, Stacked distributions of protein half-lives calculated for 55 isolates. **c**, Comparison of median turnover rates in euploid isolates (*n* = 10) versus all aneuploid isolates (*n* = 45) or isolates exhibiting high attenuation (*n* = 7). *P* values were determined using two-sample, two-sided Wilcoxon tests. **d**, Correlation between median turnover rate and protein-level dosage compensation in aneuploid isolates. Pearson correlation coefficient (PCC),

*P* value and linear regression (blue line) are shown. The PCC and *P* value of the shown correlation do not change when genes expressed on aneuploid chromosomes in aneuploid isolates are excluded (PCC = 0.31, *P* = 0.037, two-sided). **e**, Comparison of median quantile-normalized turnover rates when a protein is expressed from aneuploid chromosomes, euploid chromosomes of aneuploid strains or euploid chromosomes of euploid strains. **f**, Distribution of PCCs between relative protein expression and isolate turnover rate determined for proteins expressed from euploid chromosomes of euploid isolates or aneuploid chromosomes. **g**, Distribution of PCCs for proteins expressed from aneuploid chromosomes split by protein-complex membership. In box plots, the centre marks the median, hinges mark the 25th and 75th percentiles and whiskers show all values that, at maximum, fall within 1.5 times the interquartile range.

course, transcriptional responses are likely to be dynamic with respect to different environmental and growth conditions as well, but these results indicate a strong background dependency of transcriptional signals, and their mitigation at the proteome suggests that some of them might not be functional.

Second, at the proteome level, where dosage compensation was previously either attributed to the attenuation of surplus protein complexes or considered minimal[5,10,38], in the natural isolates we find that dosage compensation applies at a broad scale. Indeed, 70.5% of proteins encoded on aneuploid chromosomes across functional classes, including proteins that are not part of protein complexes, show attenuation, with protein levels shifting to the euploid state all across the aneuploid chromosome. The attenuation effect sizes in aneuploids remove, on average, one-quarter of the additional relative gene dosage provided by the extra chromosome. Of note, individual isolates can differ markedly from the average, with some attenuating up to half of the extra proteomic mass. When it comes to chromosome-wide dosage compensation, natural yeast isolates more closely resemble the protein attenuation observed for mammalian tissue culture cells or *L. donovani*, which also show dosage compensation at the chromosome-wide level[26–31]. A likely explanation is that the synthetic yeast aneuploids represent a state in which cells are not yet adapted to aneuploidy, whereas natural strains might have stably carried aneuploidies over long periods of time[1,2]. Our findings thus propose a timeline for adaptive processes that could result in chromosome-wide dosage compensation in mediating

aneuploidy tolerance. When aneuploidy offers a selective advantage, as demonstrated in certain environmental or stress conditions, or in the presence of toxic substances and antifungal drugs[4,5,11–14,16], the fitness benefits can outweigh the costs associated with aneuploidy. These 'naive aneuploids' may at first lack chromosome-wide dosage compensation and suffer fitness costs, making them transient in temporary environmental or stress conditions[3,12]. However, if environmental conditions continue to favour aneuploidy, selective pressure may select for strains with dosage compensation, to increase the stability of the aneuploid state.

Third, the finding that dosage compensation dominates proteins that are not necessarily members of complexes requires new mechanistic explanations. Here, the data recorded for natural isolates highlight the role of the UPS[51] and protein turnover rates: structural components of the proteasome were increased in aneuploids relative to euploids, and proteins encoded on aneuploid chromosomes showed an increased level of ubiquitination in absolute terms. The data are also consistent with the notion that the synthetic strains present a pre-adapted state, in comparison to natural isolates. Although the disomes do not show increased expression of structural components of the proteasome, they show signs of proteotoxic stress[21,43,52]. Furthermore, when evolved in vitro under selective antibiotic pressure to maintain the aneuploid chromosome, they accumulate adaptive mutations that influence UPS activity and increase the levels of protein degradation[53]. Because this data pointed to the role of protein turnover in dosage compensation,

we turned to dynamic SILAC experiments. We found that proteins with a high turnover are better dosage compensated, and in natural isolates with the highest attenuation levels, we observe a faster protein turnover. Moreover, when a protein is encoded on an aneuploid chromosome and is thus more likely to be attenuated, it is more likely to exhibit a higher turnover compared with when it is encoded on a euploid chromosome in either a euploid or an aneuploid strain.

How could an increase in total protein turnover mediate the attenuation of gene dosage? First, at a faster turnover rate, the principle of non-exponential degradation[25] might extend from stable protein complexes to more transient interactions, as well as to interactions between proteins, small molecules or metal ions. For instance, metabolite-bound proteins, which are consequently more stable, might be degraded more slowly than unbound proteins[54]. This hypothesis is supported by previous observations from high-throughput proteomic experiments: proteins that turn over rapidly are less likely to be differentially expressed across the yeast deletion collection strains, probably because they are better buffered[45]. In addition, proteins that are generally more tightly regulated across the *S. cerevisiae* species have a higher propensity to be attenuated. Second, single-cell diversity and rapid clonal selection might give an advantage to those cells with a proteome closer to the most optimal level. Consistent with this idea, aneuploid isolates with chromosome losses (for example, monosomic chromosomes in a diploid strain) also compensate for these aneuploidies by increasing the protein levels closer to the euploid state. Thus, dosage compensation specifically, and buffering effects that act in *trans* on euploid chromosomes, could be achieved by degrading a broad range of proteins faster, and then by natural selection for cells with optimal protein levels or by regulatory processes that promote proteostasis. Our results therefore add to increasing evidence that the proteome level buffers transcriptional events. For example, we previously found that proteomic changes resulting from genetic perturbations are buffered[45], and others have found that co-expression signals resulting from the three-dimensional structure of the genome are buffered as well[55].

Finally, this study reveals considerable diversity in the natural yeast proteome. This diversity is evident across several dimensions, including protein abundance, ubiquitination, dosage compensation, turnover rates and responses to aneuploidy in different strains. Our findings serve as a potent reminder that outcomes observed in a single genetic background, no matter how meticulously analysed, may not be universally representative. Consequently, incorporating proteomic data alongside genomic and transcriptomic information from a broad collection of natural isolates provides a valuable resource for addressing numerous questions about the generalizability of key evolutionary, ecological, molecular and metabolic processes.

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

## Methods

### Reagents

Unless otherwise noted, reagents were purchased as follows. Bacto yeast extract (212750), Bacto peptone (211677), Bacto dehydrated agar (214010), water (LC–MS grade, Optima, 10509404), acetonitrile (ACN) (LC–MS grade, Optima, 10001334), methanol (LC–MS grade, Optima, A456-212) and formic acid (LC–MS grade, 13454279) were purchased from Fisher Chemicals. Heavy ($^{13}C_6/^{15}N_2$) lysine was purchased from Roth (2085.1) and Silantes (211604102). Trypsin (sequence grade, V511X) was purchased from Promega. D-glucose (G7021), glycerol (G2025), DL-dithiothreitol (BioUltra, 43815), iodoacetamide (BioUltra, I1149) ammonium bicarbonate (eluent additive for LC–MS, 40867), yeast nitrogen base without amino acids (Y0626) and glass beads (acid washed, 425–600 µm, G8772) were purchased from Sigma-Aldrich. Urea (puriss. P.a., reag. Ph. Eur., 33247H) and acetic acid (eluent additive for LC–MS, 49199) were purchased from Honeywell Research Chemicals. Ninety-six-well solid-phase extraction plates (MACROSpin C18, 50–450 µl, SNS SS18VL) were purchased from the Nest Group.

### Yeast strains

The natural isolate library counts 1,023 strains in total, of which 997 strains were previously described[1] to be representative of the entire *S. cerevisiae* species. A further 26 strains were described in two studies[7,56]. The isolates were arranged in a 96-well plate format according to estimated growth rates from growth on YPD agar. Aneuploidies in strains from ref. 56 and laboratory isolates were manually detected through the coverage plots of the genomic read mapping. For all other isolates, the aneuploidy annotations as described[1] were considered. All strain details including aneuploidy, phylogenetic classification, ecological origin of isolation and ploidy are provided in Supplementary Table 1.

Mat A disomic yeast strains, constructed by A. Amon's laboratory[3], were provided by R. Li (disome WT, strain 11311; disome 1, strain 12683; disome 2, strain 12685; disome 4, strain 24367; disome 5, strain 14479; disome 8, strain 13628; disome 9, strain 13975; disome 10, strain 12689; disome 11, strain 13771; disome 12, strain 12693; disome 13, strain 12695; disome 14, strain 13979; disome 15, strain 12697; and disome 16, strain 12700).

### Transcriptomic data

Raw read counts are described in a parallel study[6] and were filtered to include only genes with a mean of more than 1 count per million across measured strains (Supplementary Table 18). These filtered read counts were then normalized using the trimmed mean of M-values (TMM) method[57] as implemented in edgeR[58,59]. Non-zero, non-log$_2$-transformed counts-per-million values were used for further analysis.

Microarray gene-expression data for lab-engineered disomic strains were downloaded from the supplementary material of a previous study[3]. Only data for strains grown in batch culture were used. Raw expression profiling data for this dataset are available from the Gene Expression Omnibus database[60] under the accession number GSE7812.

### High-throughput cultivation of yeast isolates

**Natural isolates.** The yeast samples were cultivated and digested as follows: the collection was grown on agar plates containing synthetic minimal (SM) medium (6.8 g l$^{-1}$ yeast nitrogen base, 2% glucose, without amino acids). Subsequently, colonies were inoculated in SM liquid medium (200 µl) and incubated at 30 °C overnight. Then, 160 µl of the culture was transferred to 96-deep-well plates pre-filled with one borosilicate glass bead in each well and diluted 10× in SM liquid medium to a total volume of 1.6 ml per well. Plates were sealed using an oxygen-permeable membrane and grown at 30 °C to exponential phase (Supplementary Table 2), shaking at 1,000 rpm for 8 h. Then, 1.5 ml of cell suspension was transferred to a new deep-well plate and collected by centrifugation (3,220$g$, 5 min, 4 °C). The supernatant was

discarded and plates were immediately cooled on dry ice, then stored at −80 °C until further processing.

**Lab-engineered synthetic disomic strains.** Samples were grown using SD-His+G418 agar and medium selecting for the duplicated chromosomes (6.7 g l$^{-1}$ yeast nitrogen base without ammonium sulfate, Difco 233520; 20 g l$^{-1}$ glucose; 1 g l$^{-1}$ monosodium glutamate, VWR 27872.298; 0.56 g l$^{-1}$ CSM-His-Leu-Met-Trp-Ura, MP Biomedicals 4550422; 0.02 mg ml$^{-1}$ uracil; 0.06 mg ml$^{-1}$ leucine; 0.02 mg ml$^{-1}$ methionine; 0.04 mg ml$^{-1}$ tryptophan; 200 µg ml$^{-1}$ G418, Gibco 11811023). Each disomic strain and the euploid wild type were set up in triplicate. The procedure for cultivation and lysis of the disomic strains was as described above, except that the collection by centrifugation was performed at 2,700$g$, 10 min and 4 °C.

### Preparation of proteomics samples

**Natural isolates.** The samples for proteomics were prepared in 96-well plates as previously described[41,61], with up to four plates processed in parallel. For yeast lysis, 200 µl of lysis buffer (100 mM ammonium bicarbonate and 7 M urea) and around 100 mg glass beads were added to each well, followed by 5 min bead beating at 1,500 rpm (Spex Geno/Grinder). For reduction and alkylation, 20 µl of 55 mM DL-dithiothreitol (1 h incubation at 30 °C) and 20 µl of 120 mM iodoacetamide (incubated for 30 min in the dark at ambient temperature) were used. Subsequently, 1 ml of 100 mM ammonium bicarbonate was added per well, followed by centrifugation (3,220$g$, 3 min) and 230 µl of this mixture was transferred to plates pre-filled with 0.9 µg trypsin per well. The samples were incubated for 17 h at 37 °C and the digestion was subsequently stopped by adding 24 µl of 10% formic acid (FA). The mixtures were cleaned up using C18 96-well plates, with 1-min centrifugations between the steps at the described speeds. The plates were conditioned with methanol (200 µl, centrifuged at 50$g$), washed twice with 50% ACN (200 µl, centrifuged at 50$g$) and equilibrated three times with 3% ACN/0.1% FA (200 µl, centrifuged at 50$g$, 80$g$ and 100$g$, respectively). Then, 200 µl of the digested sample was loaded (centrifuged at 100$g$) and washed three times with 3% ACN/0.1% FA (200 µl, centrifuged at 100$g$). After the last washing step, the plates were centrifuged at 180$g$. Subsequently, peptides were eluted in three steps, twice with 120 µl and once with 130 µl of 50% ACN (180$g$), and collected in a plate (1.1 ml, square well, V-bottom). The collected material was completely dried on a vacuum concentrator and redissolved in 40 µl of 3% ACN/0.1% FA before transfer to a 96-well plate. The final peptide concentration was estimated by absorption measurements at 280 nm with a Lunatic photometer (Unchained Labs, 2 µl of sample). All pipetting steps were performed with a liquid handling robot (Biomek NXP) and samples were shaken on a thermomixer (Eppendorf Thermomixer C) after each step.

**Lab-engineered synthetic disomic strains.** Lysis, reduction, alkylation and digestion of the disomic strains were performed as described above. The digest was quenched using 25 µl of 10% FA per sample. The conditioning of the solid-phase-extraction plates was performed as described above, but using 0.1% FA instead of the 3% ACN/0.1% FA mixture. After loading 200 µl of the digested sample, the columns were washed four times with 200 µl of 0.1% FA followed by centrifugation (150$g$). Purified peptides were collected by three consecutive elution steps using 110 µl of 50% ACN (centrifugation at 200$g$). After vacuum drying, peptides were dissolved in 30 µl of 0.1% FA. All steps of the sample preparation were performed by hand. Peptide concentrations were determined using a fluorimetric peptide assay kit following the manufacturer's instructions (Thermo Fisher Scientific, 23290).

### LC–MS/MS measurements

**Natural isolates.** For the collection of natural isolates, liquid chromatography was performed on a nanoAcquity UPLC system (Waters) coupled to a Sciex TripleTOF 6600. Peptides (2 µg) were separated on

a Waters HSS T3 column (150 mm × 300 µm, 1.8-µm particles) ramping in 19 min from 3% B to 40% B (solvent A: 1% ACN/0.1% FA; solvent B: ACN/0.1% FA) with a non-linear gradient (Supplementary Table 19). The flow rate was set to 5 µl min⁻¹. The SWATH acquisition method[62] consisted of an MS1 scan from $m/z$ 400 to $m/z$ 1,250 (50 ms accumulation time) and 40 MS2 scans (35 ms accumulation time) with a variable precursor isolation width covering the mass range $m/z$ 400 to $m/z$ 1,250 (Supplementary Table 20). Proteomic raw data were recorded using Analyst v.1.8.1.

**Lab-engineered synthetic disomic strains.** Proteomics measurements were performed on an Agilent 1290 Infinity LC system coupled to a SCIEX TripleTOF 6600 equipped with an IonDrive source as previously described[41]. Buffer A consisted of 0.1% FA in water, and buffer B of 0.1% FA in ACN. All solvents were LC–MS grade. Five micrograms of peptides per sample were separated at 30 °C with a 5-min active gradient starting with 1% B and increasing to 36% B on an Agilent Infinitylab Poroshell 120 EC-C18 column (2.1 × 50 mm, 1.9-µm particles). The flow rate was set to 0.8 ml min⁻¹ and the scanning SWATH acquisition method consisted of an $m/z$ 10-wide sliding isolation window.

### Generation of an experimental spectral library for strain S288c
Five micrograms of yeast digest was injected and run on a nanoAcquity UPLC (Waters) coupled to a SCIEX TripleTOF 6600 with a DuoSpray Turbo V source. Peptides were separated on a Waters HSS T3 column (150 mm × 300 µm, 1.8-µm particles) with a column temperature of 35 °C and a flow rate of 5 µl min⁻¹. A 55-min linear gradient ramping from 3% ACN/0.1% FA to 40% ACN/0.1% FA was applied. The ion source gas 1 (nebulizer gas), ion source gas 2 (heater gas) and curtain gas were set to 15 psi, 20 psi and 25 psi, respectively. The source temperature was set to 75 °C and the ion spray voltage to 5,500 V. In total, 12 injections were run with the following $m/z$ mass ranges: 400–450, 445–500, 495–550, 545–600, 595–650, 645–700, 695–750, 745–800, 795–850, 845–900, 895–1,000 and 995–1,200. The precursor isolation window was set to $m/z$ 1 except for the mass ranges $m/z$ 895–1,000 and $m/z$ 995–1,200, for which the precursor windows were set to $m/z$ 2 and $m/z$ 3, respectively. The cycle time was 3 s, consisting of high- and low-energy scans, and data were acquired in 'high-resolution' mode. The spectral libraries were generated using library-free analysis with DIA-NN directly from these scanning SWATH acquisitions. For this DIA-NN analysis, MS2 and MS1 mass accuracies were set to 25 ppm and 20 ppm, respectively, and the scan window size was set to 6.

### Proteomics data processing
**Natural isolates.** Protein-wise fasta files were created by inferring single-nucleotide polymorphisms for each strain on the basis of the reference genome of the S288c strain. In cases of heterozygosity, one of the possible alleles was randomly inferred[1,7]. For non-reference genes, a single representative sequence per protein was available based on the genomes. The proteome for the reference strain S288c was obtained from UniProt (UP000002311, accessed 10 February 2020)[63]. Sequences of strains present in the original strain collections[1,7] and subject to intellectual property restrictions were excluded from our study, leading to the inclusion of 1,023 strains in the processing. To reduce the processing time and limit the search space to relevant peptides, the protein-wise fasta files were processed to select peptides that were well shared across the strain collection. The protein sequences were thus trypsin-digested in silico and missed cleavages were disregarded. Non-proteotypic peptides were excluded and only peptides shared by 80% of the strains were selected for further analysis. This list of peptides was used to filter the experimental library. Raw mass spectrometry files were processed using the filtered spectral library with the DIA-NN software (v.1.7.12)[42]. Default parameters of the software were used except for the following: mass accuracy, 20; mass accuracy MS1, 12. Because the peptides selected were not necessarily present ubiquitously in all

the strains, an additional step was required to remove false-positive peptide assignments (entries in which a peptide is detected in a strain in which it should be absent). This filter led to the exclusion of around 1% of the entries. Samples with insufficient MS2 signal quality (around $5.7 \times 10^7$) and entries with a $q$ value greater than 0.01 or a protein group $q$ value greater than 0.01 were removed. Outlier samples were detected on the basis of both the total ion chromatograms (TIC) and the number of identified precursors per sample ($z$-score > 2.5 s.d.) and were excluded from further analysis. Precursor normalized values as inferred by DIA-NN that were well detected across the samples (in at least 80% of the strains) and with CV < 0.3 in the quality control samples were retained. Subsequently, batch correction was performed at the precursor level by bringing median precursor quantities of each batch to the same value. Proteins were then quantified using the maxLFQ[64] function implemented in the DIA-NN R package, resulting in a dataset containing 1,576 proteins for 796 strains. Missing values (less than 4% of all values) were imputed using $k$-nearest neighbours (KNN) imputation[65].

**Lab-engineered synthetic disomic strains.** Mass spectrometry files were processed using the experimental spectral library obtained through gas phase fractionation for the S288c strain with the DIA-NN software (v.1.7.12). Default parameters of the software were used except for the following: mass accuracy, 20; mass accuracy MS1, 12. The output from the software was then processed in R. Entries with a $q$ value greater than 0.01 or a protein group $q$ value greater than 0.01 and non-proteotypic peptides were removed. Samples with too low an optical density (OD) (less than 0.075) were filtered out for further analysis (disome 4 and one replicate of disome 8). The precursor normalized values inferred by DIA-NN were used and precursors that were well detected across 80% of the samples were retained. Proteins were then quantified using the maxLFQ function implemented in the DIA-NN R package. The resulting dataset consists of 1,377 proteins for 38 samples. Missing values (less than 2.35% of all values) were imputed using the KNN approach. The median value of all available replicate measurements was used for each protein during all further analyses.

### Twenty-four-hour time-course proteomics
Yeast isolates were cultivated on SM medium (as above) in batch culture. In brief, colonies from across an agar plate were incubated in 5 ml medium for 16 h at 30 °C, 750 rpm. The pre-culture was diluted to an optical density at 600 nm ($OD_{600\,nm}$) of 0.1 in 30 ml medium, and incubated for 24 h at 30 °C, 750 rpm. At regular intervals, the $OD_{600\,nm}$ was recorded, and around $4 \times 10^7$ cells were collected by centrifugation (5 min, 10,000$g$, 4 °C) at five time points to cover early exponential, mid-exponential, late exponential and stationary phases of growth. Samples were lysed in screw cap tubes by adding around 100 mg of glass beads and 160 µl of lysis buffer (7 M urea and 100 mM ammonium bicarbonate (ABC)), followed by four cycles of bead beating (5 min, 1,500 rpm followed by 5 min on ice) using a GenoGrinder. Samples were centrifuged (5 min, 10,000$g$, 4 °C) and the supernatant was transferred to a 500-µl 96-well plate. Twenty microlitres of 55 mM DTT was added to each well and the samples were incubated for 1 h at 30 °C. Subsequently, the plate was cooled on ice for 5 min, and then 20 µl of 120 mM IAA was added to each well. The samples were incubated for 30 min at 25 °C in the dark. The reduced and alkylated samples were diluted by adding 500 µl of 100 mM ABC to each well. Then, 2 µg of trypsin/LysC was added to each sample and the plate was incubated for 17 h at 37 °C. The digest was stopped by the addition of 35 µl 20% FA, and peptides were purified using solid-phase extraction as described above. Purified peptides were dried using a vacuum concentrator and dissolved in 35 µl 0.1% FA, and peptide concentrations were determined using a fluorimetric peptide assay kit following the manufacturer's instructions (Thermo Fisher Scientific, 23290).

Peptide separation was accomplished in a 63-min water to ACN active gradient on an Ultimate 3000 RSLnanoHPLC coupled to a Q Exactive

Plus mass spectrometer (both Thermo Fisher Scientific) operating in data-independent acquisition (DIA) mode. Tryptic peptides (1 µg) were concentrated on a trap column (PepMap C18, 5 mm × 300 µm × 5 µm, 100 Å, Thermo Fisher Scientific, buffer containing 2:98 (v/v) ACN/water containing 0.1% (v/v) trifluoroacetic acid, flow rate of 20 µl min⁻¹) and separated on a C18 column (Acclaim PepMap C18, 2 µm, 100 Å, 75 µm, 150 mm, Thermo Fisher Scientific) in a linear gradient from 5–28% buffer B in 63 min followed by an increasing step to 98% B in 1 min and washing for 9 min with 98% buffer B before equilibration for 15 min with initial conditions with a flow of 300 nl (buffer A, 0.1% formic acid; buffer B, 80% ACN and 0.1% formic acid). The total acquisition time was 100 min. The Orbitrap worked in centroid mode with a duty cycle consisting of one MS1 scan at 70,000 resolution power with a maximum injection time of 300 ms and an AGC target of $3 \times 10^6$ followed by 40 variable MS2 scans using a 0.5-Da overlapping window pattern. The window length started with 25 MS2 scans at 12.5 Da, followed by 7 windows with 25 Da, and the last 8 windows were set to 62.5 Da. Precursor MS spectra ($m/z$ 378–1,370) were analysed with 17,500 resolution after 110 ms accumulation of ions to a target value of $3 \times 10^6$ in centroid mode. The following mass spectrometric settings were used: spray voltage, 2.1 kV; no sheath and auxiliary gas flow; heated capillary temperature, 275 °C; normalized HCD collision energy 27%. In addition, the background ions $m/z$ 445.1200 acted as lock mass.

Raw data were processed using DIA-NN v.1.8 (ref. 42) with the scan window size set to 7 and the MS2 and MS1 mass accuracies set to 20 ppm and 10 ppm, respectively. A spectral library-free approach and yeast UniProt (UP000002311, reviewed, canonical, downloaded 18 November 2021)[63] were used for annotation. The output was filtered at 1% FDR on peptide level. Log₂-transformed protein expression levels between the aneuploid and the euploid isolate were calculated per time point for each protein present in at least two of the three biological replicates of the euploid strain in the given time point, and normalized per strain and time point as described below for the natural isolate library.

## Ubiquitinomics

Selected aneuploid and euploid yeast isolates were cultivated in SM medium (6.7 g l⁻¹ yeast nitrogen base with ammonium sulfate, Difco 291920, 20 g l⁻¹ glucose) at 30 °C. Three individual pre-cultures per strain were cultured for 16 h, and used to inoculate three flasks of 30–50 ml SM medium per strain. Cultures were collected at mid-log phase by centrifugation (2,880g, 8 min, 4 °C) and pellets were frozen at −20 °C. Cells were lysed using glass beads (volume equal to pellet volume) in 200 µl freshly prepared SDC buffer (1% sodium deoxycholate, 10 mM TCEP, 40 mM chloroacetamide and 75 mM Tris-HCl, pH 8.5) by five cycles of 1 min vortexing, 1 min on ice. Samples were centrifuged (13,800g, 15 min, 4 °C) and the supernatant was collected. Protein concentrations were determined using a Pierce BCA Protein Assay Kit (Thermo Fisher Scientific, 23225). Then, 500 µg of proteins was digested with a trypsin/LysC mix (V5071 or V5072, Promega) overnight at 37 °C with a 1:50 enzyme-to-protein ratio. K-GG peptide enrichment was performed as reported previously[39]. The digestion was stopped by adding two volumes of 99% ethylacetate/1% TFA, followed by sonication for 1 min using an ultrasonic probe device (energy output of around 40%). The peptides were desalted using 30 mg Strata-X-C cartridges (8B-S029-TAK, Phenomenex) as follows: (a) conditioning with 1 ml isopropanol; (b) conditioning with 1 ml of 80% ACN/5% NH₄OH; (c) equilibration with 1 ml of 99% ethylacetate/1% TFA; (d) loading of the sample; (e) washing with 2× 1 ml of 99% ethylacetate/1% TFA; (f) washing with 1 ml of 0.2% TFA; and (g) elution with 2× 1 ml of 80% ACN/5% NH₄OH. The eluates were snap-frozen in liquid nitrogen and lyophilized overnight. K-GG peptide enrichment was performed by resuspending lyophilized peptides in 1 ml of cold immunoprecipitation (IP) buffer (50 mM MOPS pH 7.2, 10 mM Na₂HPO₄ and 50 mM NaCl). Peptides were then incubated with 4 µl of K-GG antibody bead conjugate (Cell Signaling Technology, PTMScan HS Ubiquitin/SUMO Remnant Motif

(K-ε-GG) Kit, 59322) for 2 h at 4 °C with end-over-end rotation. Beads were washed (with the help of a magnetic stand) four times with 1 ml IP buffer and an additional time with cold Milli-Q water. After removing all of the supernatant, the beads were incubated with 200 µl of 0.15% TFA at room temperature while shaking at 1,400 rpm. After briefly spinning, the supernatant was recovered and desalted using in-house-prepared, 200 µl two plug StageTips[66] with SDB-RPS (3M Empore, 2241). SDB-RPS StageTips were conditioned with 60 µl isopropanol, 60 µl 80% ACN/5% NH₄OH and 100 µl 0.2% TFA. The K-GG enrichment eluate (0.15% TFA) was directly loaded onto the tips followed by two washing steps of 200 µl 0.2% TFA each. Peptides were eluted with 80% ACN/5% NH₄OH. Peptides were Speedvac-dried and then resuspended in 10 µl of 0.1% FA, of which 4 µl were injected into the mass spectrometer.

For LC–MS measurement, peptides were loaded on 40-cm reversed-phase columns (75 µm inner diameter, packed in-house with ReproSil-Pur C18-AQ 1.9 µm resin (ReproSil-Pur, Dr. Maisch)). The column temperature was maintained at 60 °C using a column oven. An EASY-nLC 1200 system (Thermo Fisher Scientific) was directly coupled online with the mass spectrometer (Q Exactive HF-X, Thermo Fisher Scientific) through a nano-electrospray source, and peptides were separated with a binary buffer system of buffer A (0.1% FA plus 5% DMSO) and buffer B (80% ACN plus 0.1% FA plus 5% DMSO), at a flow rate of 300 nl min⁻¹. The mass spectrometer was operated in positive polarity mode with a capillary temperature of 275 °C. The DIA method consisted of an MS1 scan ($m/z$ = 300–1,650) with an AGC target of $3 \times 10^6$ and a maximum injection time of 60 ms ($R$ = 120,000). DIA scans were acquired at $R$ = 30,000, with an AGC target of $3 \times 10^6$, 'auto' for injection time and a default charge state of 4. The spectra were recorded in profile mode and the stepped collision energy was 10% at 25%. The number of DIA segments was set to achieve an average of four to five data points per peak. For details on the DIA method set-up, see a previous report[39].

Raw data-independent acquisition data files were analysed using DIA-NN (v.1.8) in library-free mode searching against the *S. cerevisiae* reference proteome (strain ATCC 204508/S288c, UniProt ID UP000002311, excluding isoforms, accessed 18 November 2021). Trypsin/P, one missed cleavage, a maximum of two variable modifications (including cysteine carbamidomethylation and diglycine remnant modification, K-GG) and a precursor charge rate between 2 and 4 were set for precursor ion generation. The MBR and remove interferences options were enabled and Robust LC (high precision) was chosen as the quantification strategy. Peptides with a diglycine remnant (UniMod: 121) were used to quantify genes using the built-in MaxLFQ algorithm in DIA-NN, with a global and run-specific FDR of 1% being applied at both the precursor and the protein group level. The resulting data were filtered to include only genes that were measured in at least two of the three biological replicates in at least one strain for further analyses.

## Intracellular lysine measurements

Isolates (ABH, AFR, AHR, AHS, AII, AIP, ALK, ALM, ALT, AMC, ANA, ANR, APD, APM, APT, AQD, ARL, ARV, ASV, ATA, ATC, AVL, BBD, BBV, BDA, BDI, BDK, BDL, BFK, BFV, BHE, BIP, BKP, BLF, BPA, BPG, BPH, BPP, BTS, CAH, CAN, CCH, CHH, CLN, CLT, CME, CMF, CMM, CMN, CNL, CPB, CPE, CPQ, CPR, CPT, CQQ, CQR, CRL, SACE.YAB and SACE.YCO, as for dynamic SILAC experiments, see below) were randomized in triplicate on two 96-well microtitre plates. Six replicates of a lysine-auxotroph lab strain (BY4742-HLU)[67] were also added (three positions randomly per plate). Colonies were picked after 48 h growth at 30 °C on SM medium + 2% agar with a Singer Rotor HDA (Singer Instruments) and pre-cultured overnight in 200 µl SM medium supplemented with labelled L-lysine (80 mg l⁻¹) for the continuous labelling experiment, or 200 µl unlabelled L-lysine (80 mg l⁻¹) for the switching experiment. The OD₆₀₀ nm was measured after 17 h (Tecan Infinite) and cells were diluted to a starting OD of around 0.1 in 1.6 ml Lys-8- or Lys-0-labelled medium, respectively. For the continuous labelling experiment, isolates were

cultured for 8 h at 30 °C whilst shaking, collected by centrifugation (2,900g, 10 min, 4 °C), and stored overnight at −80 °C. For the switching experiment, isolates were cultured at 30 °C whilst shaking for 4 h, then centrifuged (2,900g, 10 min, 30 °C), the supernatant discarded and the pellets washed using 1 ml SM medium, cultivated for another 3 h at 30 °C whilst shaking, and collected as above. The final $OD_{600\,nm}$ at collection of all samples was measured.

Amino acids were extracted by adding 200 µl pre-cooled 80% ethanol containing the internal standard D4-L-lysine (Silantes, 211113913) to each of the frozen cell pellets. The samples were incubated for 2 min at 80 °C and subsequently vortexed. This step was repeated two more times. The samples were centrifuged (2,900g, 10 min, 4 °C) and the supernatants were collected. The measurements of 1 µl per sample were performed on a triple quadrupole mass spectrometer system (Agilent 6460) as previously described[68]. Technical controls from pooled extracted metabolite samples were included and measured by LC–MS/MS after every 15th sample, in total 27 times. The analysis was performed using MassHunter Software B.07.01 (Agilent Technologies). The internal standard response ratios were calculated for each sample and normalized to the $OD_{600\,nm}$ measured at collection.

**Dynamic SILAC**

Yeast strains (46 diploid aneuploid isolates with a single chromosome gain (trisomic strains) for which we quantified attenuation, 2 randomly chosen haploid aneuploid isolates with a single chromosome gain, as well as 10 diploid euploid and two haploid euploid isolates with a similar range of growth rates to that of the aneuploid isolates, meaning isolates ABH, AFR, AHR, AHS, AII, AIP, ALK, ALM, ALT, AMC, ANA, ANR, APD, APM, APT, AQD, ARL, ARV, ASV, ATA, ATC, AVL, BBD, BBV, BDA, BDI, BDK, BDL, BFK, BFV, BHE, BIP, BKP, BLF, BPA, BPG, BPH, BPP, BTS, CAH, CAN, CCH, CHH, CLN, CLT, CME, CMF, CMM, CMN, CNL, CPB, CPE, CPQ, CPR, CPT, CQQ, CQR, CRL, SACE.YAB and SACE.YCO; see also Supplementary Table 16) were grown on synthetic medium containing 6.7 g l⁻¹ yeast nitrogen base, 2% glucose and 80 mg l⁻¹ L-lysine (SM + Lys-0). For SILAC labelling, L-lysine was swapped for 80 mg l⁻¹ heavy [$^{13}C_6$/$^{15}N_2$] lysine (SM + Lys-8). Cells were taken from cryo stocks and streaked on freshly prepared SM + Lys-0 agar plates (20 g l⁻¹ agar) and grown for 48–72 h at 30 °C. Colonies across the whole agar plate were gathered and cultivated in 5 ml SM + Lys-0 for approximately 16 h at 30 °C and 300 rpm. The overnight pre-culture was then diluted in 25 ml in SM + Lys-0 (pre-warmed to 30 °C) to a starting $OD_{600\,nm}$ of around 0.1. The culture was grown at 30 °C, 300 rpm until it reached an $OD_{600\,nm}$ of between 0.25 and 0.3. At this point, the medium was switched from SM + Lys-0 to SM + Lys-8 using the following procedure. First, 20 ml of the culture was transferred into a 50-ml Falcon tube and centrifuged for 5 min at 30 °C, 3,095g. Then, the supernatant was decanted and the pellet was washed twice with 4 ml SM + Lys-8 (pre-warmed to 30 °C). Lastly, the pellet was resuspended in 20 ml warm SM + Lys-8 and transferred into clean flasks. The heavy-labelled cultures were grown at 30 °C, 300 rpm and at three time points (90 min, 135 min and 180 min), 2 ml of the culture was collected into ice-cold screw cap tubes. The samples were centrifuged at 10,000g, 4 °C, the supernatant aspirated and the pellets stored at −80 °C. At each collection time point, the $OD_{600\,nm}$ was also recorded. Strains BPP, BDK, ATA and BFV did not grow well under the chosen conditions or were not growing exponentially when sampled, and were therefore omitted from further processing.

Cells were lysed mechanically in screw cap tubes by adding around 100 mg glass beads and 100 µl fresh lysis buffer (7 M urea and 100 mM ABC) to each sample, followed by two cycles of bead beating (5 min, 1,500 rpm, followed by 5 min on ice) using a GenoGrinder. The samples were briefly centrifuged (4,000g, 1 min) and the supernatant was transferred to a 500-µl Eppendorf 96-well plate. From this step onwards, all samples were processed together in high throughput. Reduction, alkylation and digest were performed as described in the 'Twenty-four-hour time-course proteomics' section, using 10 µl of

55 mM DTT, 10 µl of 120 mM IAA, 380 µl of 100 mM ABC and 2 µg of trypsin/LysC. Samples were digested for 17 h at 37 °C. The digest was stopped by the addition of 25 µl of 20% FA and the samples were purified using solid-phase extraction as described above. Purified peptides were dried using a vacuum concentrator and subsequently dissolved in 25 µl 0.1% FA. Peptide concentrations were determined using a fluorimetric peptide assay kit following the manufacturer's instructions (Thermo Fisher Scientific, 23290).

For each strain, 1 µg of peptide sample was separated on a Vanquish-Neo System (Thermo Fisher Scientific) by reverse-phase chromatography with a 30-min efficient gradient from 3 to 30% ACN on a self-packed 20-cm column (ID 75 µm, 1.9-µm beads), and directly injected through electrospray ionization (ESI) to an Exploris480 Orbitrap (Thermo Fisher Scientific). In brief, the MS settings for Top20 acquisition scheme were the following: ESI voltage: 2.2 kV; resolution MS1 60k; IT MS1 10 ms; RF-Lens 55; resolution MS2 15k; maxIT MS2 50 ms; isolation width 1.2 Da; HCD collision energy 28; AGC target 100%.

Raw files were analysed with MaxQuant v.1.6.7.0 using standard settings, with match between runs and requantify enabled, and the Uniprot *S. cerevisiae* protein database including isoforms (downloaded 9 February 2023) was selected for the database search. The complexity was set to 2, with Lys-8 set as the heavy label. Further processing of the data and calculation of half-lives was done in R. First, the evidence.txt was loaded with the fread function from the data.table package, filtered for lysine-containing peptides and cleaned from potential contaminants and remaining reverse hits. Owing to the fact that many proteins in yeast are very stable, a correction for doubling times is not applicable to most identified proteins. As in a previous study[46], we therefore calculated turnover rates (kdp) and the corresponding half-lives without doubling time or dilution rate correction (Supplementary Table 16). In more detail, protein turnover rates were calculated for proteins with valid SILAC ratios in at least two time points per strain by building a linear model from the different sampling time points against the log-transformed H/L ratios, thus calculating kdp. The corresponding slopes from each fit depict the kdp value for each strain. Half-lives were calculated from the resulting kdp as log(2)/kdp (Supplementary Table 17). Isolate CLN was excluded owing to a very low number of valid SILAC ratios obtained at t = 135 min. Furthermore, three proteins (ERP1, NEO1 and MDE1) were excluded from the dataset because they were measured in only a few isolates (6, 4 and 3, respectively) and exhibited very high variability in half-lives across these strains.

**Post-processing statistical analyses**

All statistical analyses were conducted in R v.3.6 unless otherwise indicated. KEGG annotations for *S. cerevisiae* genes were obtained through the KEGG database (accessed January 2021)[69]. The org.Sc.sgd. db package[70] was used to obtain chromosomal location information for genes and to map gene names to systematic open reading frame (ORF) identifiers. If no gene name was annotated in this package, the systematic ORF identifier was used instead. Standardized *S. cerevisiae* yeast strain names and systematic ORF identifiers for genes were used throughout all analyses. Heat maps were plotted using the Complex-Heatmap package[71]. In all box plot representations, the centre marks the median, box plot hinges mark the 25th and 75th percentiles and whiskers show all values that, at maximum, fall within 1.5 times the interquartile range.

**Assembly of integrated chromosome copy number, mRNA and protein expression datasets.** Gene copy numbers for natural isolates were downloaded from the 1002 Yeast Genome website (http://1002genomes.u-strasbg.fr/files/)[1], and the following loci were excluded: ribosomal DNA, Ty elements, RTM loci, ORFs located on the 2-micron plasmid, mitochondrial ORFs and non-reference material. Furthermore, the table was filtered to retain only genes with non-zero and non-missing values for further analyses. Chromosome

copy number status for all engineered disomic strains was confirmed by Torres et al.[3], meaning that all disomic strains used in our study were haploid with indicated 'disomic' chromosomes duplicated. One exception was disome 13: despite published mRNA expression values being available and proteomics data having been measured in our experiments, disome 13 was excluded from all analyses because it had undergone whole-genome duplication when reaching our laboratory (personal communication, J. Zhu).

For 761 isolates, both proteomes (this study) and transcriptomes[6] were available, and for 759 isolates, gene copy number information[1] was available. Data for gene copy number, mRNA expression and protein abundances were matched by strain name and systematic ORF identifier for both the natural isolate collection and the disomic strain collection. Only genes for which values for gene copy number, transcript and protein levels were available were used for analyses. We noticed that a number of strains in the natural isolate collection exhibited a mismatch between the median gene copy number per chromosome and the assigned aneuploidy as described in Supplementary Table 1, which is likely to be attributable to segmental aneuploidies, shorter gene copy number variations or algorithm-specific thresholds used for aneuploidy determination. We excluded all strains ($n = 80$) containing one or more of those 'mismatched' chromosomes from our analysis (Supplementary Table 5). From this point, chromosome copy numbers as given by the aneuploidy annotation were used throughout the analyses.

**Calculation of relative chromosome copy numbers, mRNA and protein expression values.** The assembled integrated dataset was used to compare the relative changes in chromosome copy number, mRNA transcript expression and protein abundances between aneuploid and euploid strains. Relative chromosome copy number changes were calculated as the $\log_2$ ratio between the chromosome copy number and the ploidy of the strain. Relative abundances for transcriptomic and proteomic data were calculated gene-wise as the $\log_2$ ratio between a gene's mRNA or protein abundance in a given strain and the median mRNA or protein expression value of the respective gene across all euploid strains ('all-euploid strain' method). In addition, ploidy-wise calculation of relative mRNA or protein abundance was tested, comparing the abundances of a haploid strain to the median abundance of that same mRNA or protein across all euploid haploid strains, each diploid strain to all euploid diploids and so forth for all basal ploidies ('ploidy-wise' method). There was a high correlation between relative expression values calculated across all euploid strains, and ploidy-wise calculated relative expression values (Extended Data Fig. 9), indicating that non-linear scaling of the proteome with ploidy[72] had no significant effect on the outcome of the used data normalization strategy. For the transcriptomic data of the lab-engineered disomic strains, we used the transcript levels as published; that is, as $\log_2$ fold changes relative to the wild-type strain (disome WT, 11311)[3]. For replicate measurements, the median value was used for further analysis.

For the $\log_2$ mRNA and protein ratios of genes encoded on euploid chromosomes, a distribution centred around 0 would be expected, representing no overall shift of relative expression values of these genes across strains. This was true for our proteomics data, and also for most strains in the transcriptomic data. Because some natural isolates showed left tails in this distribution for the transcriptomic data, presumably because of restricting the assembled dataset to genes for which we had data across all three -omics layers, we decided to normalize the relative mRNA and protein expression values. Normalization was performed in a strain-by-strain manner for both the across-euploid strains and the ploidy-wise ratio calculation methods (see above): first, the median $\log_2$ mRNA or protein ratio of all genes encoded on euploid chromosomes of a given strain was calculated. This median value was then subtracted from all $\log_2$ mRNA or protein ratios of that strain.

The proteome profiles of disome 12 and disome 14 showed no aneuploid signature, indicating that those strains, even though they were held under selective pressure, had lost their duplicated chromosome either before their arrival in our laboratory or during our experiments. Both strains were therefore excluded from our analysis. Similarly, when comparing relative chromosome copy number changes and relative mRNA expression levels in the natural isolate collection, we noticed discrepancies indicative of chromosomal instabilities in natural *S. cerevisiae* isolates. Some euploid strains had gained or lost chromosomes, evident as much higher or lower fold changes of expression values observed in the transcriptomics data than in chromosome copy numbers. Likewise, some aneuploid strains underwent changes in their karyotype, resulting in either more complex aneuploidies or in aneuploid strains reverting to euploid strains. We decided to include in our analysis only strains that showed consistent relative expression ($\log_2$ ratio) values on the chromosome copy number and the transcriptome level. Consequently, we excluded strains that had at least one chromosome for which the difference between relative chromosome copy number and the median of the normalized relative mRNA abundances differed by more than ±4 standard deviations from the mean, on the basis of all relative chromosome–mRNA comparisons (Supplementary Table 6, $n = 66$). After excluding these strains, the calculations to obtain gene-wise relative (strain/euploid) mRNA and protein expression values were repeated to avoid unintended biases towards these excluded strains.

**Gene-by-gene quantification of dosage compensation.** Linear regressions between $\log_2$ mRNA or protein expression ratios and relative chromosome copy number (CN) changes ($\log_2$ chromosome CN/ basal ploidy) were performed for all genes that were encoded on an aneuploid chromosome in at least three different natural aneuploid isolates (so genes on chromosomes 1, 3, 4, 5, 6, 8, 9, 11, 12 and 14). Isolates that had reverted to euploidy (Supplementary Table 12) were excluded from this analysis. Relative chromosome copy numbers were restricted to be greater than or equal to 0, thus including all euploid chromosomes, and all chromosome gains of aneuploid isolates, but excluding chromosome losses. Therefore, each regression was performed using data for the expression of the gene on euploid chromosomes ($\log_2$ CN change = 0) and at least three independent data points with a relative chromosome CN change greater than 0. The slopes of these gene-by-gene linear regressions were used as a measure of across-isolate dosage compensation. For lab-engineered disomic strains, a similar analysis was performed; however, it was necessarily restricted to one 'aneuploid' data point per gene and forced through 0 because each aneuploid chromosome was engineered exactly once in the disomic strain collection. Therefore, in total, the regressions were performed for 827 and 680 genes at the mRNA and protein level in natural isolates and disomic strains, respectively. For the cumulative distribution ('rolling threshold') analysis, the number of mRNAs or proteins exhibiting attenuation slopes smaller than a given threshold were counted, and effect sizes for these attenuated mRNAs or proteins were calculated as the median of the attenuation slopes smaller than the respective threshold. For the following analyses, a threshold of 0.85 was selected to define attenuated mRNAs and proteins.

For assessment of the protein properties on attenuation, the following sources were used: macromolecular-complex membership: Complex Portal of the EBI (accessed December 2020)[73]; protein–protein interactions (PPIs): STRING database (accessed November 2022)[74]; prediction of protein disorder and linear interacting peptides by Alpha-Fold, MobiDB and anchor: MobiDB (accessed October 2022)[75]; GC content and percentile mean gRSCU: calculated on the *Saccharomyces cerevisiae* S288C sequence (NCBI: GCF_000146045.2_R64) using the gc1, gc2, gc3 and gRSCU functions in BioKIT v.0.1.2 (ref. 76); ribosome occupancy: from a previous study[77]; amino acid synthesis costs and glucose cost: from a previous study[78]; absolute protein copy numbers

per cell: from a previous study[79]; protein length, mass and modification sites: UniProt (accessed October 2022, ubiquitinated residue information inferred from experimental and automatic cross-link evidence listed as 'Glycyl lysine isopeptide (Lys-Gly) (interchain with G-Cter in ubiquitin)')[63]; and protein half-life: from a previous study[46]. The internal variability of transcripts and proteins was calculated as the standard deviation of mRNA and protein abundance across all euploid isolates of the collection, respectively. Receiver operator characteristics were calculated using the pROC package[80] as described previously[26].

For assessment of mRNA- and protein-level annotation across pathways, cellular localizations, molecular functions and biological processes, KEGG annotations were obtained using the KEGG API (accessed January 2021)[69], and a GO slim mapping file was obtained from SGD (accessed January 2021)[81]. The degree of relative attenuation was determined as $100 \times (1 - slope)$ per mRNA or protein, and median attenuation levels per KEGG or GO category were calculated only for those KEGG or GO terms with at least six mRNAs or proteins for which an attenuation slope had been determined.

For the comparison of proteins non-exponentially degraded in human cells versus yeast, we identified the yeast homologues of the human proteins quantified and assigned as either exponentially degraded (ED), non-exponentially degraded (NED) or undefined in a previous report[25]. We identified 759 yeast homologues for 3,187 human proteins covered in that report[25] (around 23%), agreeing very well with the expected fraction of the human proteome that has yeast homologues. Of those 759 proteins, 146 were classified as NED, 349 classified as ED and 262 as undefined; two proteins could not be unambiguously mapped to one of these categories.

**Chromosome-wide and strain-by-strain quantification of dosage compensation.** To assess attenuation at the chromosome level, the median mRNA or protein log$_2$ ratio of all genes encoded per chromosome or relative chromosome copy number change across isolates was calculated. For both disomic lab-engineered strains and natural isolates of *S. cerevisiae*, mRNA and protein expression log$_2$ ratios between a gene's expression in a strain and the gene's expression over all euploid strains were examined in relation to the log$_2$-transformed fold change of the copy number of the chromosome on which the gene is located. For gains of chromosomes, all log$_2$ chromosome copy number changes for which fewer than 300 affected data points (genes) were quantified were excluded for distribution visualization. For chromosome losses, fewer data points were available overall, so the described cut-off was set at 50 data points (genes). The attenuation observed in the lab-engineered disomic strains measured by DIA-MS was comparable to that previously measured using SILAC[10] (Extended Data Fig. 10).

To quantify the relationship between chromosome gains (log$_2$ chromosome copy number/ploidy > 0, all relative chromosome copy number changes included) and relative mRNA or protein expression from aneuploid chromosomes, linear models were fitted between the log$_2$ chromosome copy number change and the median relative mRNA or protein expression value.

For the strain-by-strain quantification of dosage compensation, non-parametric two-sided one-sample Wilcoxon tests were performed for each relative chromosome copy number change per isolate to compare the normalized log$_2$ mRNA or protein expression distributions to the expected median (log$_2$ chromosome copy number/basal ploidy). *P* values were corrected using the Benjamini–Hochberg method. This way, for each isolate, it could be assessed whether the observed attenuation of chromosomes with the same copy number change in that isolate was significant or not. Aneuploid isolates were marked as 'reverted to euploid' if the pseudomedian of the protein-level Wilcoxon test was between −0.1 and 0.1. The attenuation at mRNA or protein level was calculated as $100 \times (1 - pseudomedian/relative$ chromosome copy number change) per isolate, with 'pseudomedian' referring to the pseudomedian obtained from the Wilcoxon test. The calculation was

performed only for isolates that had a single aneuploidy, or complex aneuploidies of the same relative chromosome copy number change of aneuploid chromosomes; that is, attenuation levels could, for example, be calculated for a diploid isolate with one extra copy of chromosome 1 (for example, isolate BDI), also, for example, for a diploid isolate with one gained copy of chromosome 1 and one gained copy of chromosome 4 (for example, isolate CFV), but not, for example, for a tetraploid isolate with a complex aneuploidy that gained one copy of chromosome 1 and lost a copy of chromosome 3 (for example, isolate BRP).

For investigating the relationship between the degree of aneuploidy and dosage compensation, we defined an additional measure of degree of aneuploidy by calculating the ploidy-adjusted absolute number of protein copies per cell of all proteins encoded on aneuploid chromosomes in a given strain (referred to as 'aneuploid protein load'). This measure correlated very well with the number of genes located on aneuploid chromosomes, a previously used measure of aneuploidy degree (for example, ref. 26; PCC = 0.96, $P \ll 0.05$).

**Assessment of the *trans* transcriptome and proteome response in natural aneuploid isolates.** *Trans* expression at the transcriptome and proteome level was defined as the mRNA or protein expression, respectively, of genes encoded on euploid chromosomes in aneuploid isolates. Genes up- or downregulated according to the ESR were mapped as described in ref. 44, genes up- or downregulated according to the CAGE signature were mapped as described in ref. 34 and genes annotated as upregulated in the APS were mapped as described in ref. 10. To find genes differentially expressed in *trans* in aneuploid strains—that is, genes encoded on euploid chromosomes of aneuploid strains that show up- or downregulation at the mRNA or protein level when compared with euploid strains—we calculated the gene-by-gene median normalized relative mRNA or protein abundances (log$_2$ ratios) of all genes encoded on euploid chromosomes in aneuploid strains ($n = 95$). KEGG-pathway GSEA of these median relative expression values was performed with WebGestalt 2019 using the default settings (accessed December 2021)[82]. In addition, one-sample *t*-tests were used to compare gene-by-gene mean normalized protein log$_2$ ratios across euploid chromosomes of aneuploid strains against the theoretical gene-by-gene mean protein log$_2$ ratio value across euploid strains ($\mu = 0$). *P* values were corrected for multiple hypothesis testing using the Benjamini–Hochberg method as implemented in the rstatix package[83]. Annotation of structural components was obtained through KEGG (accessed January 2021)[69]. Detailed proteasome component annotations were obtained from a previous study[84].

For the analysis of the role of RPN4 in mediating the increase of proteasome abundance, RPN4 transcript levels were obtained from the transcriptome data of the natural isolate collection[6], and TMM normalized and scaled as described above. RPN4 regulon targets (found in high-throughput screens and manually curated ones) were downloaded from SGD (accessed April 2023)[81], with around 50 of these targets being measured in the proteomic dataset of the natural isolates.

**Determination of ubiquitination levels.** Relative levels of ubiquitinated proteins were determined gene-wise by calculating the log$_2$ ratio between the measured abundance of a ubiquitinated protein in each strain and the median abundance of the ubiquitinated protein across all euploid strains. Assuming a distribution centred around 0 of relative levels of ubiquitinated proteins on euploid chromosomes, these relative abundances were then normalized strain-wise by subtracting the calculated median log$_2$ ratio of all genes expressed on euploid chromosomes of a strain from all log$_2$ ratios of that strain.

**Attenuation and turnover analyses.** Proteomes of the aneuploid yeast deletion collection were obtained from a previous study[45]. Fold changes were defined as ratios between protein abundances and the median abundances of the respective protein across all strains. Chromosomes

were defined as duplicated when the median $\log_2$ expression levels were greater than 0.8 across all measured proteins on the respective chromosome. $\log_2$ fold changes were averaged across the strains with duplications of the respective chromosome. Long and short half-lives were defined as being greater than the 75% and less than the 25% quantile ($n = 110$), respectively. Half-lives were taken from a reference dataset and were obtained by metabolic labelling[46].

To compare the turnover rates of proteins when expressed from aneuploid versus euploid chromosomes, we first filtered the turnover dataset to include only proteins for which we had turnover rates determined in at least 80% (44/55) of isolates and KNN-imputed the remaining missing values. We then quantile-normalized the turnover rates to correct for differences in overall turnover rates between isolates. We calculated the median quantile-normalized turnover rate for each protein when expressed from aneuploid chromosomes, from euploid chromosomes of aneuploid isolates or from euploid chromosomes of euploid isolates. These calculations were performed only for proteins for which turnover rates were determined at least three times on aneuploid and euploid chromosomes, respectively. The median turnover rates were then compared to count the number of times a protein exhibits a difference in turnover rates depending on whether it is expressed from aneuploid or euploid chromosomes.

For determining the relationship between protein attenuation and overall turnover rates of isolates, the Pearson correlation between the protein's log2 expression ratio in a given isolate (see 'Calculation of relative chromosome copy numbers, mRNA and protein expression values' section) and isolates' turnover rates was calculated. This analysis was performed for each protein expressed at least three times from aneuploid chromosomes, euploid chromosomes of euploid isolates or euploid chromosomes of aneuploid isolates. GSEA was conducted on Pearson correlation coefficients with WebGestalt 2019 using the default settings.

### Reporting summary

Further information on research design is available in the Nature Portfolio Reporting Summary linked to this article.

### Data availability

Raw and processed mass spectrometry data of the natural collection have been deposited in the MassIVE database under ID MSV000090435. Data associated with the disomes collection have been deposited at the ProteomeXchange Consortium through the PRIDE partner repository, with the dataset identifier PXD044526. Data associated with turnover experiments have been deposited at the ProteomeXchange Consortium through PRIDE with the identifier PXD048219. The RNA-sequencing data are available in the European Nucleotide Archive (ENA) under the accession number PRJEB52153. The following publicly deposited data sources have been used: gene copy number annotation for natural isolates from the 1002 yeast genome website (http://1002genomes.u-strasbg.fr/files/); microarray gene-expression data for lab-engineered disomic strains from ref. 3; SILAC proteome data for lab-engineered disomic strains from ref. 10; proteomes of the aneuploid yeast deletion collection from ref. 45 (https://y5k.bio.ed.ac.uk/); gene ontology annotation from SGD (https://www.yeastgenome.org/); yeast reference proteome databases from UniProt (https://www.uniprot.org); KEGG annotation from KEGG (https://www.genome.jp/kegg/); protein–protein interaction data from STRING (https://string-db.org); prediction of protein disorder and linear interacting peptides by AlphaFold, MobiDB and anchor from MobiDB (https://mobidb.bio.unipd.it/); ribosome occupancy data from ref. 77; amino acid synthesis costs and glucose cost from ref. 78; absolute protein copy numbers per cell from ref. 79; protein half-life data from ref. 46; exponential degradation annotation from ref. 25; annotation of protein-complex membership from the Complex Portal of the EBI (https://www.ebi.ac.uk/complexportal/

home); annotation of ESR genes from ref. 44; annotation of CAGE genes from ref. 34; annotation of APS genes from ref. 10; and proteasome component annotations from ref. 84.

### Code availability

This manuscript did not generate new software, but the work informed DIA-NN developments, which are available from GitHub (https://github.com/vdemichev/DiaNN).

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

**Acknowledgements** We thank P. Beltrao, G. Fischer, J. Hartl, S. Kamrad and V. Farztdinov for discussions throughout the project; J. L. Steenwyk for providing codon optimization and gRSCU values; and U. Ben-David for critical reading of the manuscript. J. Zhu, S. Ramachandran and R. Li sent us the lab-engineered disomic strains. We thank the Core Facility High Throughput Mass Spectrometry of the Charité for support in sample preparation, acquisition and analysis. Elements of figures were created with BioRender.com. This project has received funding from the European Research Council (ERC) under the European Union's Horizon 2020

research and innovation programme (grant agreement no. 951475—ERC Synergy Award FungalTolerance to J.B. and M.R.; PhenomeNal ERC Consolidator grant 772505 to J.S.); the Ministry of Education and Research (BMBF) as part of the National Research Node 'Mass spectrometry in Systems Medicine' (MSCoreSys) under grant agreements 031L0220 (to M.R.) and 161L0221 (to V.D.); the Wellcome Trust (Investigator Award IA 200829/Z/16/Z to M.R.); and the Fondation pour la Recherche Médicale (EQU202003010413) and Agence Nationale de la Recherche (ANR-15-IDEX-01) (to G.L.). P.T. is supported by funding from the European Union's Horizon 2020 research and innovation programme under the Marie Skłodowska-Curie grant agreement no. 101026830. C.B.M. is supported by the Precision Proteomics Center Davos, which receives funding through the Swiss canton of Grisons. N.B. is supported by the Deutsche Forschungsgemeinschaft (DFG) funded Sonderforschungsbereich (SFB) TRR 186. This work was further supported by the Francis Crick Institute, which receives its core funding from Cancer Research UK (FC001134), the UK Medical Research Council (FC001134) and the Wellcome Trust (FC001134).

**Author contributions** J.M. analysed and interpreted all data, and designed, performed and supervised proteomic, metabolomic, ubiquitinomic and protein turnover experiments. P.T. developed the raw mass spectrometry data-processing pipelines, analysed the raw proteomic data and provided scientific input. F. Agostini performed proteomic experiments and provided scientific input. H.Z. designed and performed dynamic SILAC turnover experiments. C.B.M. designed the high-throughput yeast proteomics sample cultivation and preparation pipeline and provided scientific input. M. Steger performed ubiquitinomics measurements and provided scientific input. A.L. and N.B. performed ubiquitinomics and amino acid uptake experiments, respectively. C.K. and K.L. grew and sampled yeast for turnover experiments. E.C., M.D.C., T.I.G., G.L. and J.S. provided scientific input. K.T.-T., A.-S.E. and F. Amari performed mass spectrometry measurements. V.D. contributed to the development of the data-processing pipeline for the quantitative (that is, non-turnover) proteomics part of the natural isolate proteomes. M.M. supervised mass spectrometry measurements and provided scientific input. M. Selbach designed and supervised dynamic SILAC measurements and provided scientific input. J.B. and M.R. acquired dedicated funding, supervised the project and interpreted data. J.M., J.B. and M.R. wrote the manuscript. All authors contributed to finalizing the text and proofreading.

**Funding** Open access funding provided by Max Planck Society.

**Competing interests** M. Steger was an employee of Evotec München. V.D. holds share options of NEOsphere biotechnologies. M.R. is a founder and shareholder of Eliptica. M.M. is a consultant and shareholder of Eliptica. The remaining authors declare no competing interests.

**Additional information**
**Correspondence and requests for materials** should be addressed to Judith Berman or Markus Ralser.

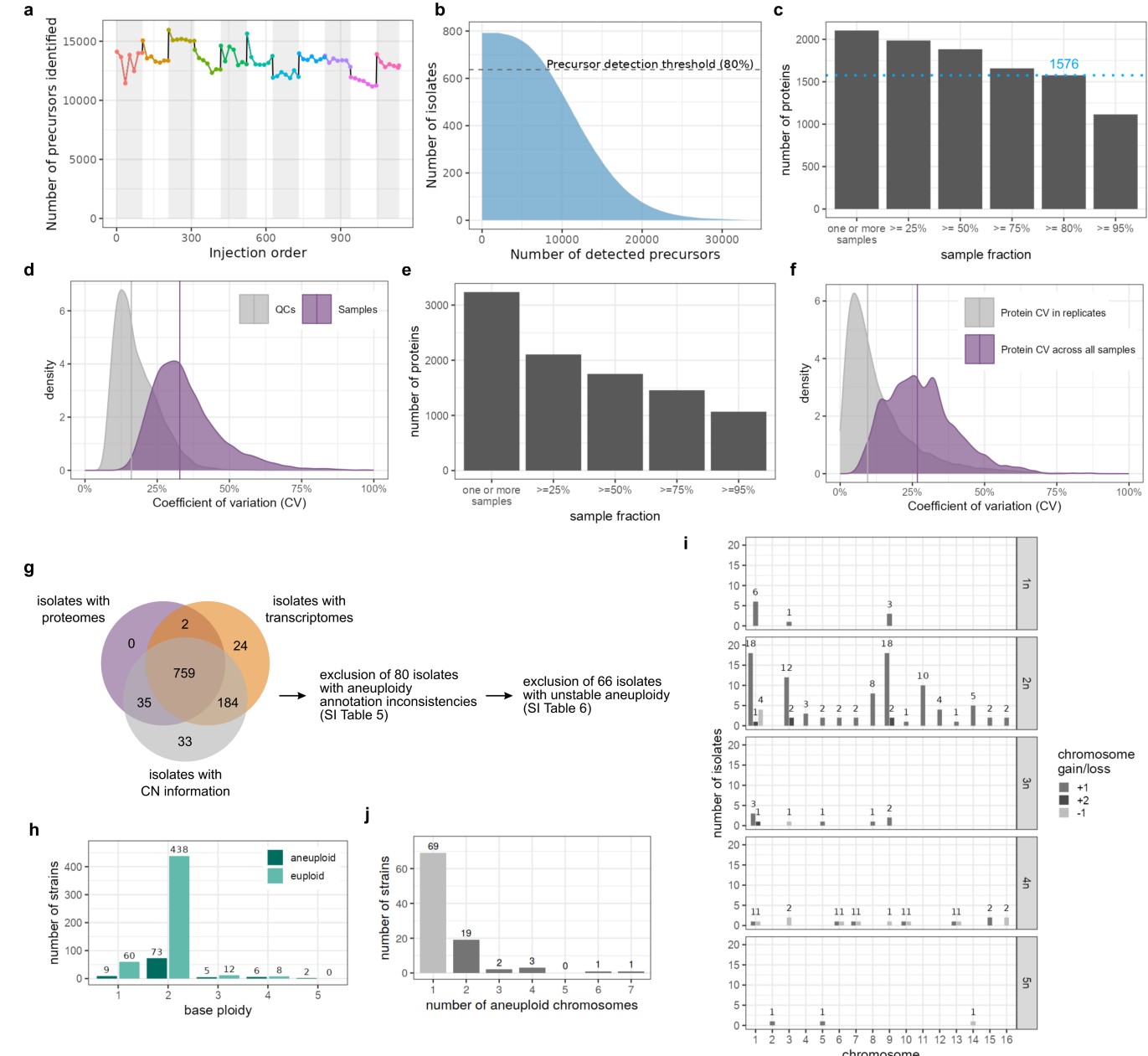

**Extended Data Fig. 1 | Quality assessment plots for proteomics data processing and integrated dataset assembly. a**, Number of precursors identified by DIA-NN in the QC samples (n = 77) of the natural isolate collection, before further processing, ordered by injection number, demonstrating a stable performance over the acquisition of the data. Colours and shaded backgrounds highlight separate batches. **b**, Precursor detection rate in the natural isolates. A strict threshold was set to retain precursors that were well detected in at least 80% of the isolates. The precursors retained were then used for protein quantification. **c**, Number of proteins quantified across samples. The blue dotted line highlights the 80% cut-off used for the proteomic dataset, leading to 1,576 proteins consistently quantified across the natural isolates after preprocessing. **d**, Coefficients of variation (CV) of the precursor quantities in the technical quality controls (QCs, technical variability, n = 77) compared to the biological samples (yeast isolates, biological signal, n = 796). The solid purple line indicates the median CV across samples (32.8 %). **e**, Number of proteins

quantified across samples for different sample fraction thresholds in the disomic dataset. **f**, Coefficients of variation (CV) of protein abundance within replicates or across all samples in the disomic dataset. All disomic strains were measured in triplicates, except for disome 8, which was measured in duplicates due to one replicate not passing preprocessing quality control thresholds (Methods). The solid purple line indicates the median CV across all samples (26.7%). The comparison of the CVs demonstrates low technical variability and a well-detected biological signal in the disome dataset. **g**, Overlap between genomic, transcriptomic, and proteomic datasets and number of isolates excluded due to inconsistencies at or between genome and transcriptome layer. **h**, Number of aneuploid and euploid isolates per ploidy. **i**, Chromosome gains (+1, +2) and losses (−1) across the isolates of the integrated dataset by ploidy. For isolates with complex aneuploidies, each aneuploid chromosome was counted separately. **j**, Distribution of chromosome copy number changes per aneuploid isolate across the 613 isolates in the integrated dataset.

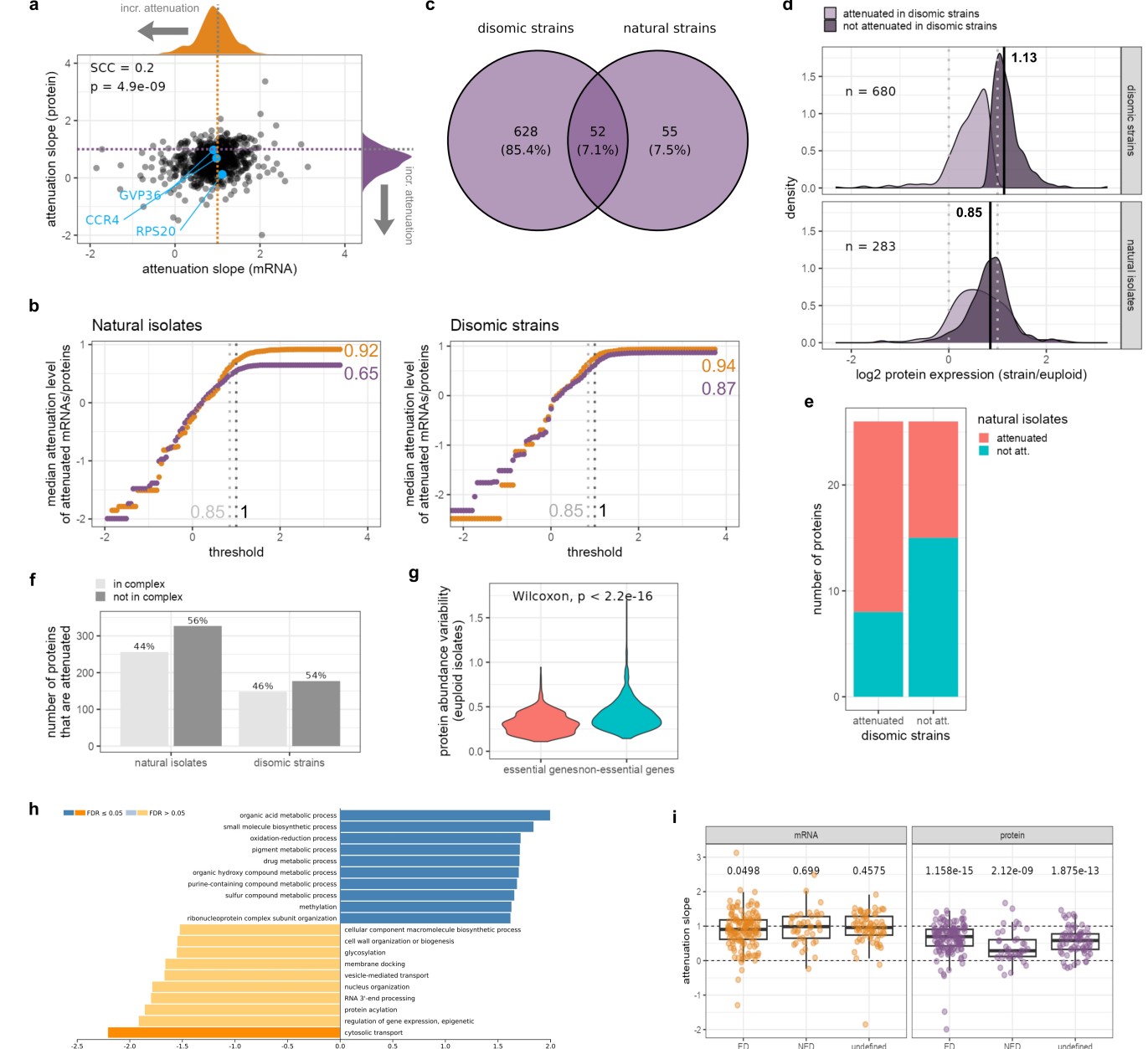

**Extended Data Fig. 2** | See next page for caption.

**Extended Data Fig. 2 | Quantification of dosage compensation 1. a**, mRNA and protein attenuation slopes for the 827 genes (black points) evaluated in the linear regression in natural isolates. Specific genes mentioned and named in Fig. 2a are highlighted in blue. The orange and purple dotted lines depict the expected slope for genes with no attenuation at either the mRNA or protein level, respectively. The distributions of the mRNA and protein regression slopes are shown at the top and right of the scatter plot, respectively. SCC: Spearman correlation coefficient, $p = 4.9 * 10^{-9}$ (two-sided). **b**, "Rolling threshold" (cumulative distribution) analysis for natural isolates and disomic strains comparing the effect size of attenuation at the mRNA (orange) or protein level (purple) based on the distribution slopes. The median attenuation level of all attenuated mRNAs or proteins at a given threshold was calculated. The vertical dotted light grey line denotes a threshold of 0.85 that was used for subsequent analyses, the darker grey dotted line denotes 1 (slope if no attenuation occurs). **c**, Venn diagram showing numbers of proteins for which a regression analysis to determine the extent of buffering at the protein level could be performed and that were expressed from genes located on duplicated chromosomes in disomic strains (680/680 proteins on which regression analysis was performed, from 9 strains), or on duplicated chromosomes in haploid as well as diploid natural isolates (107/827 proteins on which regression analysis was performed, from 13 isolates). **d**, Disomic and natural isolate log2 protein expression distributions for proteins from **c**, with those proteins that were determined to be attenuated in disomic strains shown in light purple, and those that were not attenuated in disomic strains shown in dark purple. The dotted lines mark 0 and 1 (expected euploid and duplicated log2 expression value, respectively), and the black line and number mark the respective median of the distribution of those proteins that are not attenuated in disomic strains. For the disomic strains, all 680 proteins were used to draw the protein expression distributions shown in **d**; for natural isolates, the 52 overlapping proteins from **c** were used. Because these 680 proteins appear exactly once on a duplicated chromosome in the disomic strain (only one disomic isolate per chromosome in the collection), and the 52 overlapping proteins appear on duplicated aneuploid chromosomes in multiple natural isolates, the total number of data points used to draw the distributions is n = 680, and n = 283 ( > 52), respectively. The number of attenuated versus not attenuated values in the distributions shown for the disomic strains was 325 and 355, respectively; and for the natural isolates, 145 and 138, respectively. Proteins that are not attenuated in the synthetic aneuploids (median log2 protein expression of 1.13), are, on average, nonetheless attenuated in the natural disomes (median log2 protein expression of 0.85). **e**, Bar plot showing whether proteins that are attenuated or not attenuated in disomic strains are attenuated or not in natural isolates as well. **f**, Proportion of genes attenuated at the protein level that are in a protein complex (light grey) or not part of a complex (dark grey). **g**, Variability (standard deviation) of protein abundance levels across euploid isolates for essential versus non-essential genes. Distributions medians were compared using a two-sample, two-sided Wilcoxon test ($p < 2.2 * 10^{-16}$). **h**, GSEA of the differences between mRNA and protein abundance variability (measured as standard deviation) across euploid isolates. Positive normalized enrichment scores indicate higher variability at the mRNA versus the protein abundance level, negative normalized enrichment scores indicate higher variability at the protein versus the mRNA abundance. **i**, Comparison of mRNA and protein attenuation of yeast proteins predicted to be exponentially degraded (ED, n = 138), non-exponentially degraded (NED, n = 39), and undefined (n = 69) based on work in a human aneuploid cell line[25]. "Attenuation slope" refers to the slope determined in the attenuation regression analysis (compare Fig. 2 and Methods), with slopes close to 1 (or >1) marking no attenuation. Each dot represents the attenuation slope of a single gene at the mRNA or protein level. The adjusted p-values (Benjamini–Hochberg) of one-sample, two-sided t-tests comparing the mean of each group to the expected value if no attenuation occurs ($\mu = 1$) are shown above the box plots. Box plot hinges mark the 25th and 75th percentiles and whiskers show all values that, at maximum, fall within 1.5 times the interquartile range.

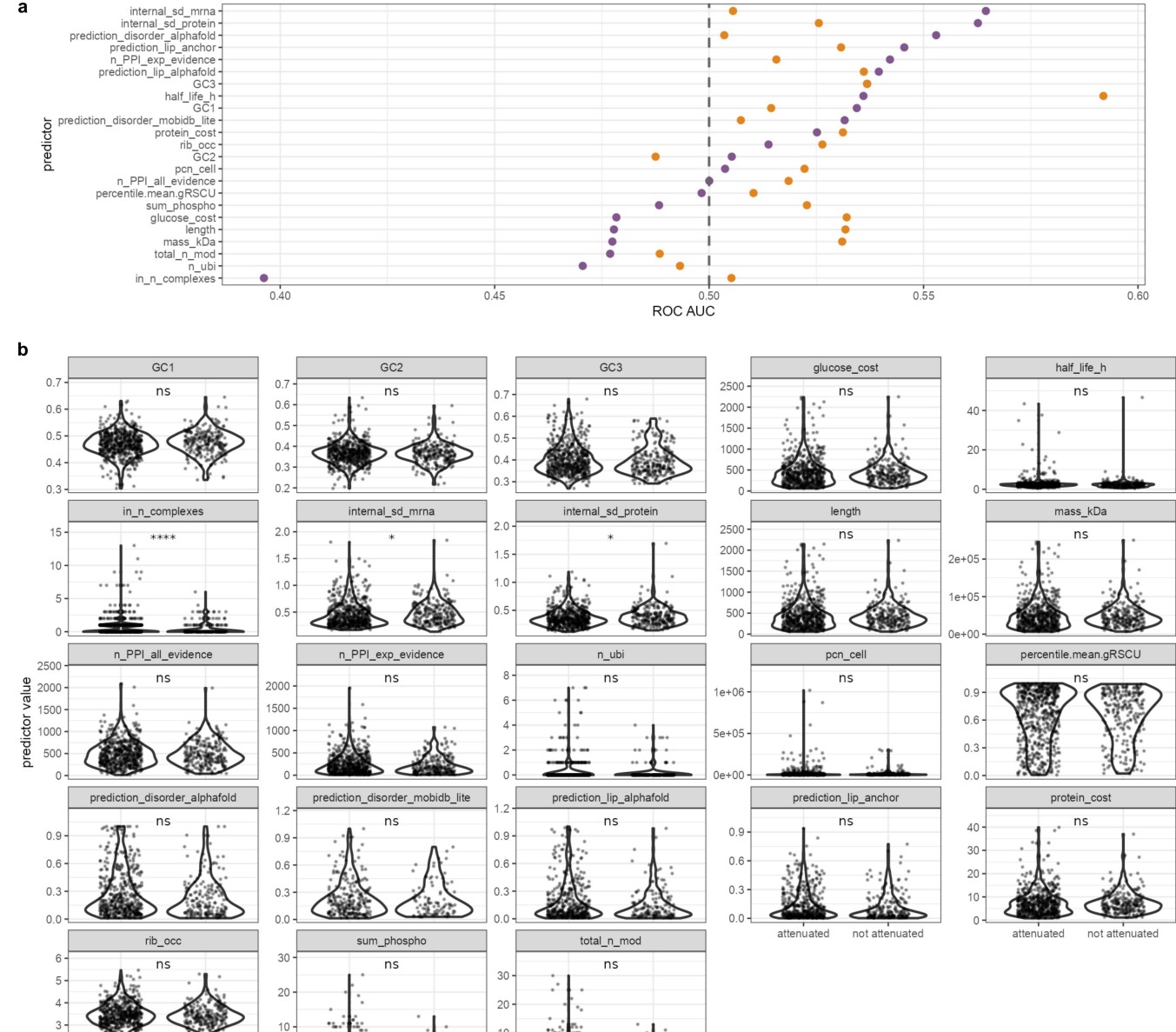

**Extended Data Fig. 3 | Quantification of dosage compensation 2.**
**a**, Comparison of the area under curve (AUC) of the receiver operator characteristic (ROC) for different features that could influence attenuation. ROC AUC values are shown for both attenuation prediction at the mRNA (orange dots) and at the protein (purple dots) level. **b**, Comparison of features (panels) between proteins (dots) that are attenuated (n = 583) and not attenuated (n = 244) in natural aneuploid yeast isolates. The Benjamini–Hochberg-adjusted p-value significance levels of two-sample, two-sided Wilcoxon tests (attenuated versus not attenuated for each feature) are shown above the violin plots (internal_sd_mrna: p = 0.034; internal_sd_protein: p = 0.034; in_n_complexes: p = 9.3*10$^{-7}$; ns: p > 0.05 after multiple hypothesis correction). In **a** and **b**, the following features are displayed: internal_sd_protein = standard deviation of protein abundance across all euploid isolates of the collection; n_PPI_all_evidence and n_PPI_exp_evidence = number of protein:protein interactions (PPIs) the protein is taking part in, both on the level of overall determined PPIs, as well as for only experimentally confirmed PPIs (STRING db);

internal_sd_mrna = standard deviation of mRNA abundance across all euploid isolates of the collection; prediction_disorder_alphafold and prediction_disorder_mobidb_lite = protein disorder as predicted by alphafold and MobiDBi, respectively; prediction_lip_anchor and prediction_lip_alphafold = occurrence of linear interacting peptides (short linear motifs in disordered protein regions) in gene sequence as predicted by ANCHOR and AlphaFold, respectively; GC1, GC2, GC3 = GC content at first, second and third codon position, respectively; rib_occ = ribosome occupancy; percentile_mean_gRSCU = codon optimization of the gene; protein_cost and glucose_cost = amino acid and glucose synthesis cost used to built each protein; pcn_cell = absolute protein copy number per cell; length and mass_kDa = length in amino acids and mass of protein in kDa, respectively; half_life_h = protein half-life in h; sum_phospho and n_ubi = number of phosphorylation and ubiquitination sites per protein; total_n_mod = total number of modification sites per protein; in_n_complexes = total number of complexes a protein is part of.

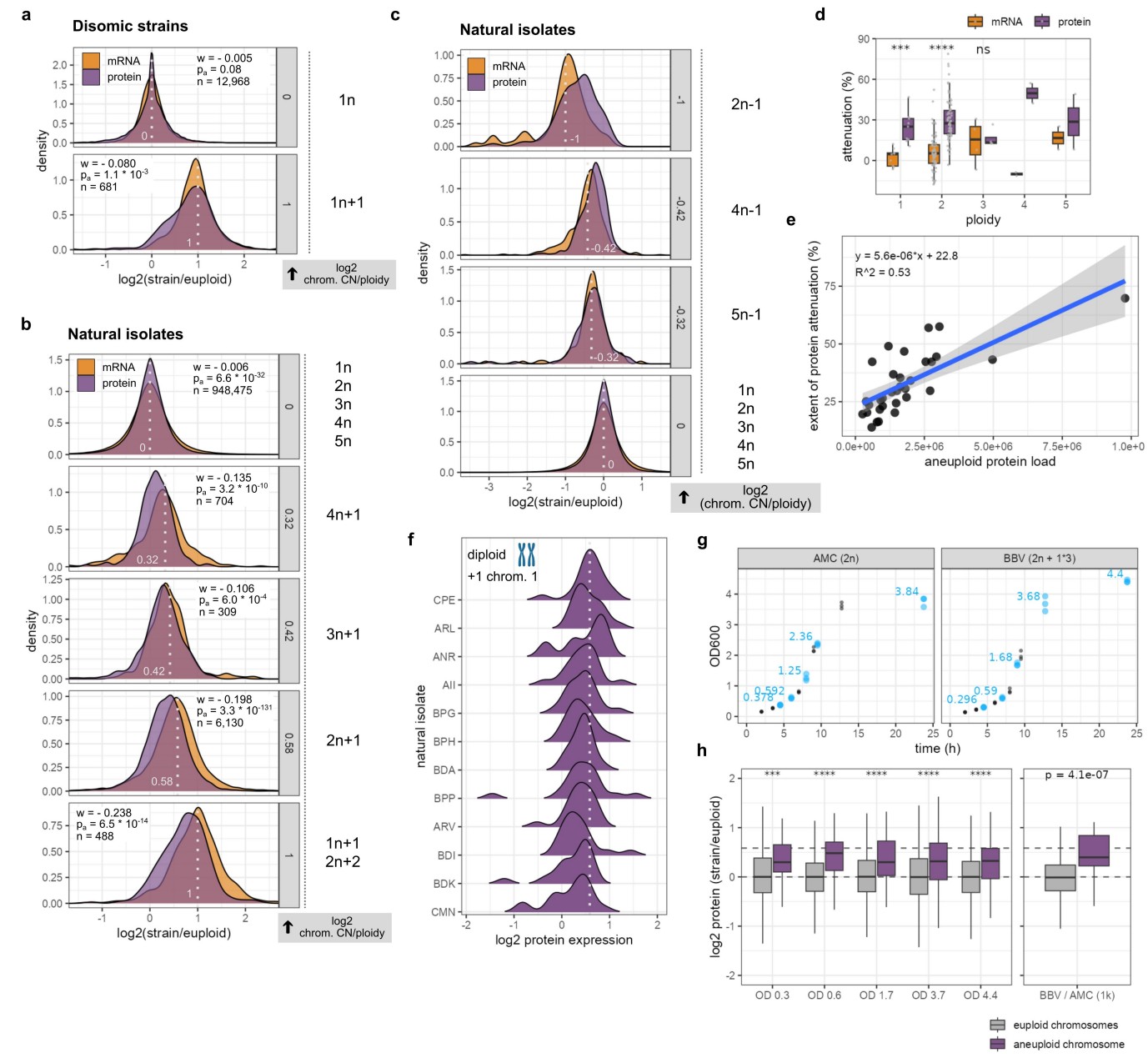

**Extended Data Fig. 4** | See next page for caption.

**Extended Data Fig. 4 | Quantification of dosage compensation 3.**
**a**–**c**, Distributions of $\log_2$ mRNA (orange) and protein (purple) ratios across all euploid or aneuploid chromosomes of disomic strains (**a**); natural isolates (**b**); of all euploid natural isolates as well as natural isolates with chromosome losses (**c**). Genes are binned according to the relative copy number (CN) change of the chromosome encoding them ($\log_2$ chrom. CN/ploidy, grey). $\log_2$ ratios of all genes encoded on euploid chromosomes are summarized in panels "0" (0: chromosome CN equal to ploidy), $\log_2$ ratios of genes encoded on aneuploid chromosomes are summarized in the other panels (−1–1: aneuploid chromosome gains (**a**,**b**) or losses (**c**) in haploid, diploid, triploid, or tetraploid isolates). Light grey dotted lines and numbers – relative chromosome CN change ($\log_2$ chrom. CN/ploidy) in the density plots. Test statistic (w), adjusted p-value ($p_a$, Benjamini–Hochberg), and observations (n) for two-sample Wilcoxon tests conducted to compare mRNA and protein distributions for each relative chromosome CN change in **a** and **b** are shown. For all distributions, relative expression levels are shown between −1.5 and 2.5. **d**, Comparison of mRNA- and protein-level buffering of aneuploid isolates (grey dots) across base ploidy. Statistical significance of the difference between mRNA and protein distributions per ploidy is indicated (two-sample, two-sided t-test, for ploidy = 1: n = 9, p = 0.00026; ploidy = 2: n = 68, p = <$2*10^{-16}$; ploidy = 3: n = 4, p = 0.78; no tests performed for tetra- and pentaploid isolates due to low number of isolates). We note that attenuation at the mRNA level in triploid aneuploid isolates appears stronger than in diploid or tetraploid isolates. However, there that are only four aneuploid triploid isolates with a single relative chromosome copy number change across all aneuploid chromosomes present in the dataset (in contrast to a higher number of haploid and diploid isolates), and we thus suspect that this observation might be an outlier rather than a true biological difference. **e**, Relationship between the median extent of protein-level buffering across isolates with the same degree of aneuploidy (black dots) and the degree of aneuploidy, measured as the sum of copies of all proteins encoded on all aneuploid chromosomes per isolate. Blue line: linear model, grey band: 95% confidence interval, R^2 = adjusted $R^2$ of linear regression, p = $1.6*10^{-6}$ (two-sided). **f**, Relative protein expression distributions of genes encoded on the aneuploid chromosome of diploid natural isolates that gained one copy of chromosome 1 ($2n + 1*1$). The dashed grey line indicates the expected median of the $\log_2$ distribution in case no attenuation occurred. **g**, 24 h growth curves of a euploid isolate (AMC) and an aneuploid isolate (BBV) after dilution from a pre-culture (t = 0 h, OD600 = 0.1). The OD was regularly monitored, and samples for proteomics were taken at five time points (highlighted in blue). The experiment was performed in biological triplicates. The median OD600 at the time points when samples were collected is shown. **h**, Left, box plots showing the distributions of $\log_2$ protein ratios of all genes encoded on the euploid chromosomes (grey) or the singly gained aneuploid chromosome (purple) at five different ODs across the growth curve from **g**. The median of the distributions is marked with a solid black line within the boxes. The displayed p-value significance levels are derived from two-sample, two-sided t-tests performed per OD between euploid and aneuploid data points (OD 0.3: n (euploid) = 2163, n (aneuploid) = 55, p = 0.00031; OD 0.6: n (euploid) = 2403, n (aneuploid) = 62, p = $1.4*10^{-8}$; OD 1.7: n (euploid) = 2701, n (aneuploid) = 71, p = $1.3*10^{-7}$; OD 3.7: n (euploid) = 2940, n (aneuploid) = 72, p = $8.4*10^{-5}$; OD 4.4: n (euploid) = 3089, n (aneuploid) = 80, p = $6.4*10^{-8}$). Only $\log_2$ ratios between −2 and 2 are shown to improve readability, and outliers are truncated. Right, relative protein expression levels between the aneuploid isolate BBV and the euploid AMC in the main dataset (795 isolates x 1,653 proteins). In box plots, the centre marks the median, hinges mark the 25th and 75th percentiles and whiskers show all values that, at maximum, fall within 1.5 times the interquartile range.

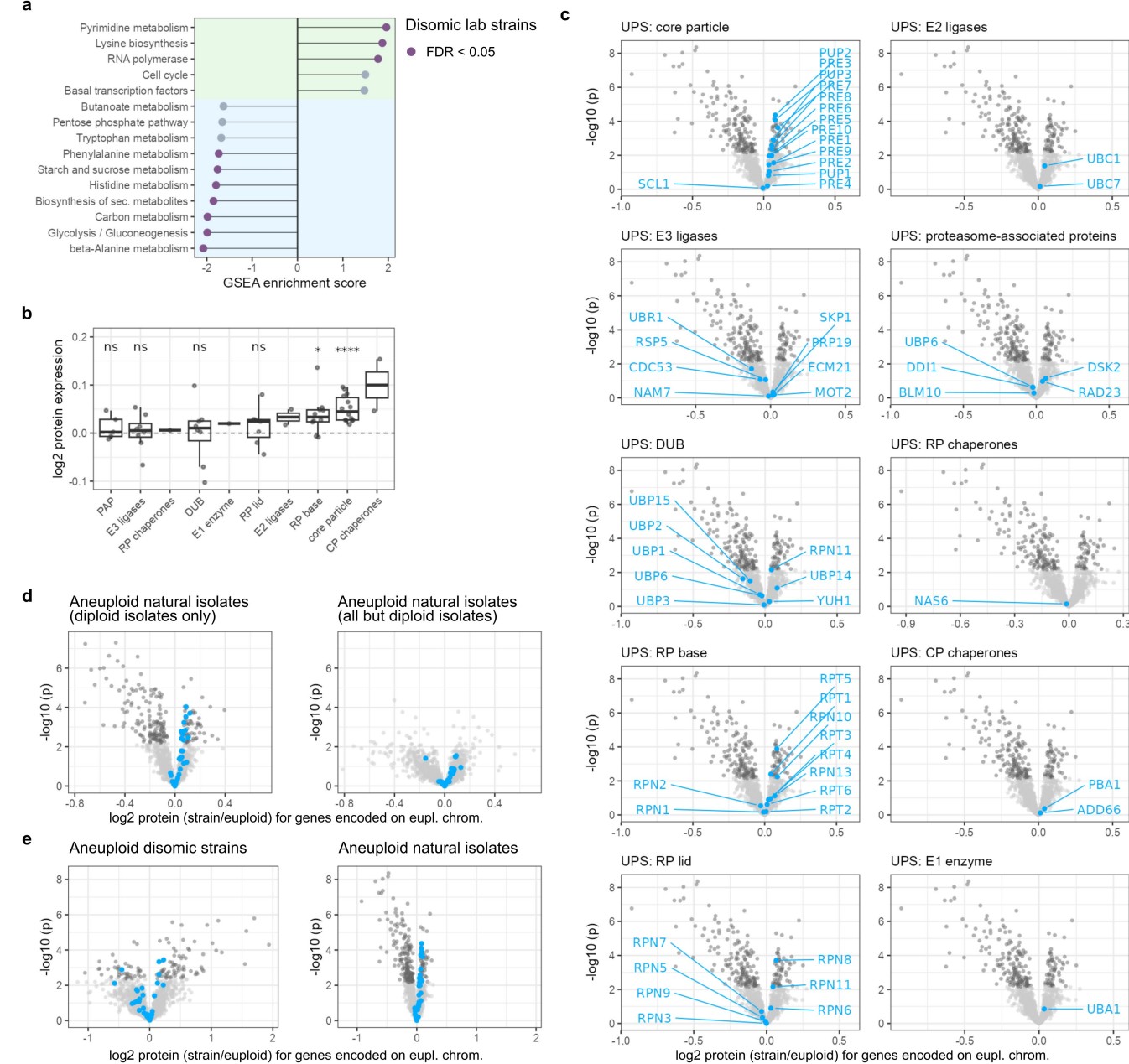

**Extended Data Fig. 5 | Enrichment of structural components of the proteasome in aneuploid natural isolates. a**, GSEA of median log2 protein expression ratios (strain/euploid) of genes encoded on euploid chromosomes across all disomic strains. Statistically significant enrichment scores (FDR < 0.05) are coloured in purple (protein). **b**, Median relative expression of UPS components on euploid chromosomes in aneuploid natural *S. cerevisiae* isolates. The adjusted p-values (Benjamini–Hochberg-corrected) of one-sample, two-sided t-tests comparing the mean of each group to the expected mean expression of UPS components in euploid isolates (μ = 0) are shown above the box plots (core particle: p = 0.000053; RP base: p = 0.044; ns: p > 0.05). In box plots, the centre marks the median, hinges mark the 25th and 75th percentiles and whiskers show all values that, at maximum, fall within 1.5 times the

interquartile range. **c**, Components of the UPS and their regulation in aneuploid isolates (*trans* expression). Volcano plots show the results of one-sample, two-sided t-tests comparing the mean log2 protein ratios of proteins expressed on euploid chromosomes of 95 aneuploid isolates to μ = 0. Proteins with statistically significant differential expression after multiple hypothesis correction (Benjamini–Hochberg) are coloured in dark grey. Components of the UPS are highlighted and labelled as blue dots in separate panels. **d**, Volcano plots showing results of one-sample, two-sided t-tests as in Fig. 4c, but for all diploid natural isolates (n = 73, left panel), or all isolates but diploid isolates (n = 22, right panel). **e**, Volcano plots as in Fig. 4c, but with the *x* and *y* axes scaled to match.

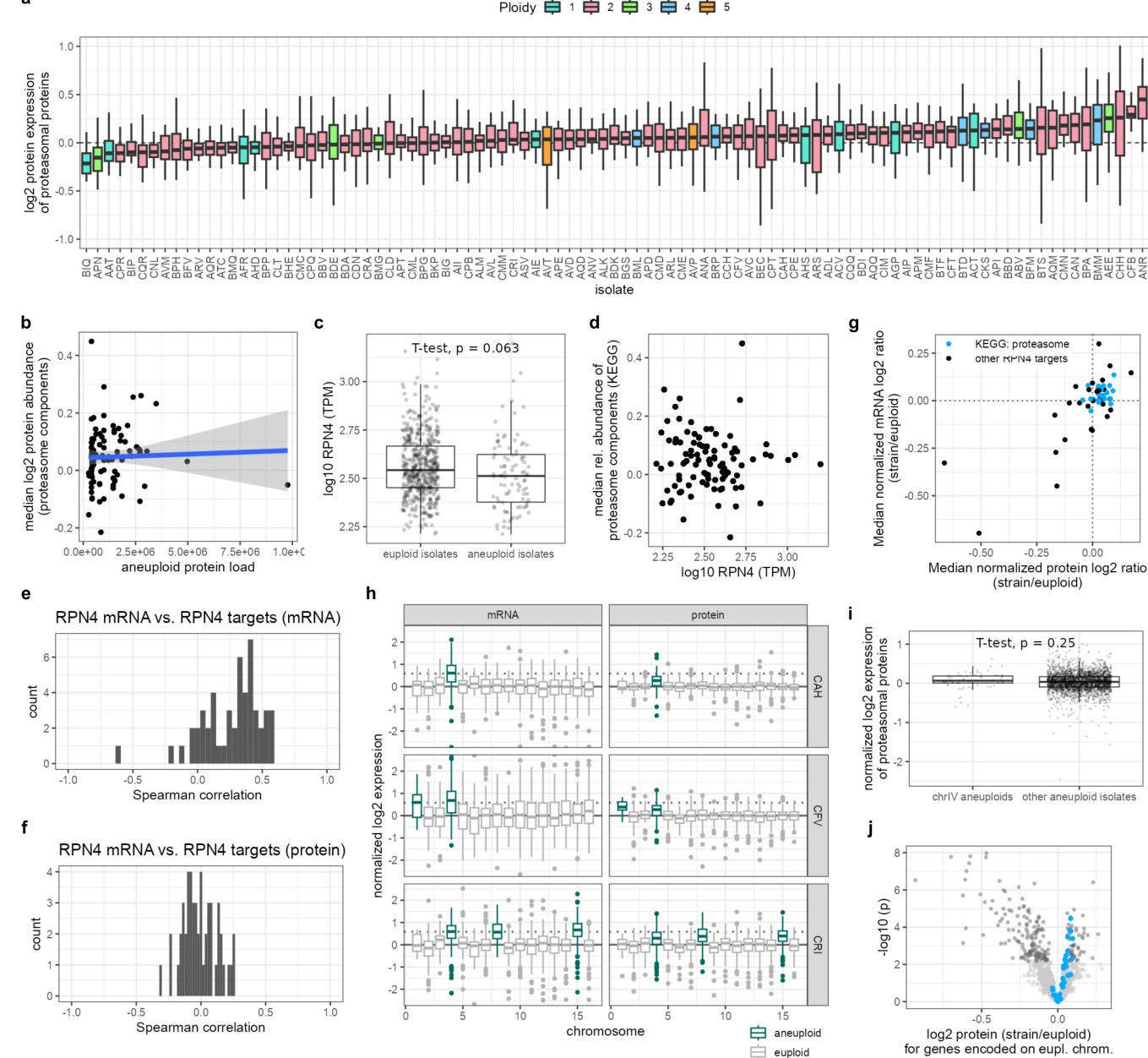

**Extended Data Fig. 6** | See next page for caption.

**Extended Data Fig. 6 | Regulation of proteasome abundance. a**, Relative expression of proteasomal proteins (KEGG term "Proteasome") on chromosomes *trans* to aneuploid chromosomes in natural *S. cerevisiae* isolates (BMM: n = 16; CRI: n = 19; BMQ: n = 20; BML, CKS: n = 24; AVC: n = 25; AEE: n = 26; ADL, CAH, CFV: n = 27; AQD, AVT, BFM, CFT, CHH: n = 28; ALK, API, AQM: n = 29; ANV, BBD, BIP, BKP, BPA, CCH, CPR: n = 30; ABV, AGP, AHS, AIE, AIP, ALM, ANA, APD, APE, APM, AQQ, AQR, ASV, ATC, AVD, AVM, BEC, BFV, BGS, BTD, BTF, BTS, CAN, CFB, CIM, CLT, CMD, CME, CPQ, CPT: n = 31; all other isolates: n = 32 proteins). The box plots are coloured by the basal ploidy of the isolates. The y-axis is shown for log2 protein ratios between −1 and 1 to improve readability. **b**, Relationship between relative abundance of proteasomal proteins (KEGG term "Proteasome") in natural yeast isolates and the aneuploid protein load (blue line: linear model, grey bands: 95% confidence interval). **c**, Comparison of RPN4 mRNA levels in aneuploid (n = 95) versus euploid isolates (n = 518) of the integrated dataset. The displayed p-value (p = 0.063) is derived from a two-sample, two-sided t-test between euploid and aneuploid data points. **d**, Relationship between the median abundance of proteasome components (KEGG term "Proteasome") in natural aneuploid isolates (black dots) and RPN4 mRNA levels (TPM: transcripts per million). No correlation is observed. **e,f**, Distribution of Spearman correlation coefficients for the correlation between RPN4 mRNA levels and either RPN4 target mRNA levels (**e**) or RPN4 target protein levels (**f**) expressed in *trans*, so on euploid chromosomes in natural aneuploid isolates (n = 95). Correlation coefficients are higher for RPN4 target mRNA levels than for RPN4 target protein levels. **g**, Regulation of RPN4 targets at the protein and mRNA level in natural aneuploid isolates. The median expression level per gene across all isolates that express the gene in *trans* to aneuploid chromosomes is shown. Structural components of the proteasome are highlighted in blue, other RPN4 targets are black. **h–j**, RPN4 is located on chromosome 4. We assessed whether natural isolates carrying a chromosome 4 aneuploidy show particularly prominent induction of the proteasome due to increased gene dosage of RPN4. **h**, Box plots showing log2 mRNA and protein expression sorted by chromosome (chr. 1: n = 18; chr. 2: n = 106; chr. 3: n = 41; chr. 4: n = 186; chr. 5: n = 89; chr. 6: n = 30; chr. 7: n = 148; chr. 8: n = 70; chr. 9: n = 48; chr. 10: n = 94; chr. 11: n = 96; chr. 12: n = 140; chr. 13: n = 134; chr. 14: n = 109; chr. 15: n = 140; chr. 16: n = 114 genes per chromosome) for the three isolates (CAH, CFV, CRI; all diploid) of the integrated dataset that carry an aneuploidy of chromosome 4. The aneuploid chromosomes are highlighted for each isolate. The grey dotted line indicates 0.58 – the expected attenuated relative expression level for mRNAs or proteins encoded on trisomic chromosomes in diploid isolates. Relative expression values are shown between −2.5 and 2.5. **i**, Comparison of upregulation of proteasomal proteins (KEGG term "Proteasome", two-sample, two-sided T-test, chromosome 4: n = 73 proteins; other aneuploid isolates: n = 2809, p = 0.25) that are expressed in trans to aneuploid chromosomes in natural isolates that carry an aneuploidy of chromosome 4 (3 isolates) versus all other natural aneuploids (92 isolates). **j**, Volcano plot for all natural aneuploid isolates except for the three isolates carrying an aneuploidy of chromosome 4 (n = 92) showing results of one-sample, two-sided t-tests comparing the mean log2 protein ratios to μ = 0. Proteins with statistically significant differential expression after multiple hypothesis correction (Benjamini–Hochberg) are coloured in dark grey. Structural components of the proteasome are highlighted in blue. In box plots, the centre marks the median, hinges mark the 25th and 75th percentiles and whiskers show all values that, at maximum, fall within 1.5 times the interquartile range.

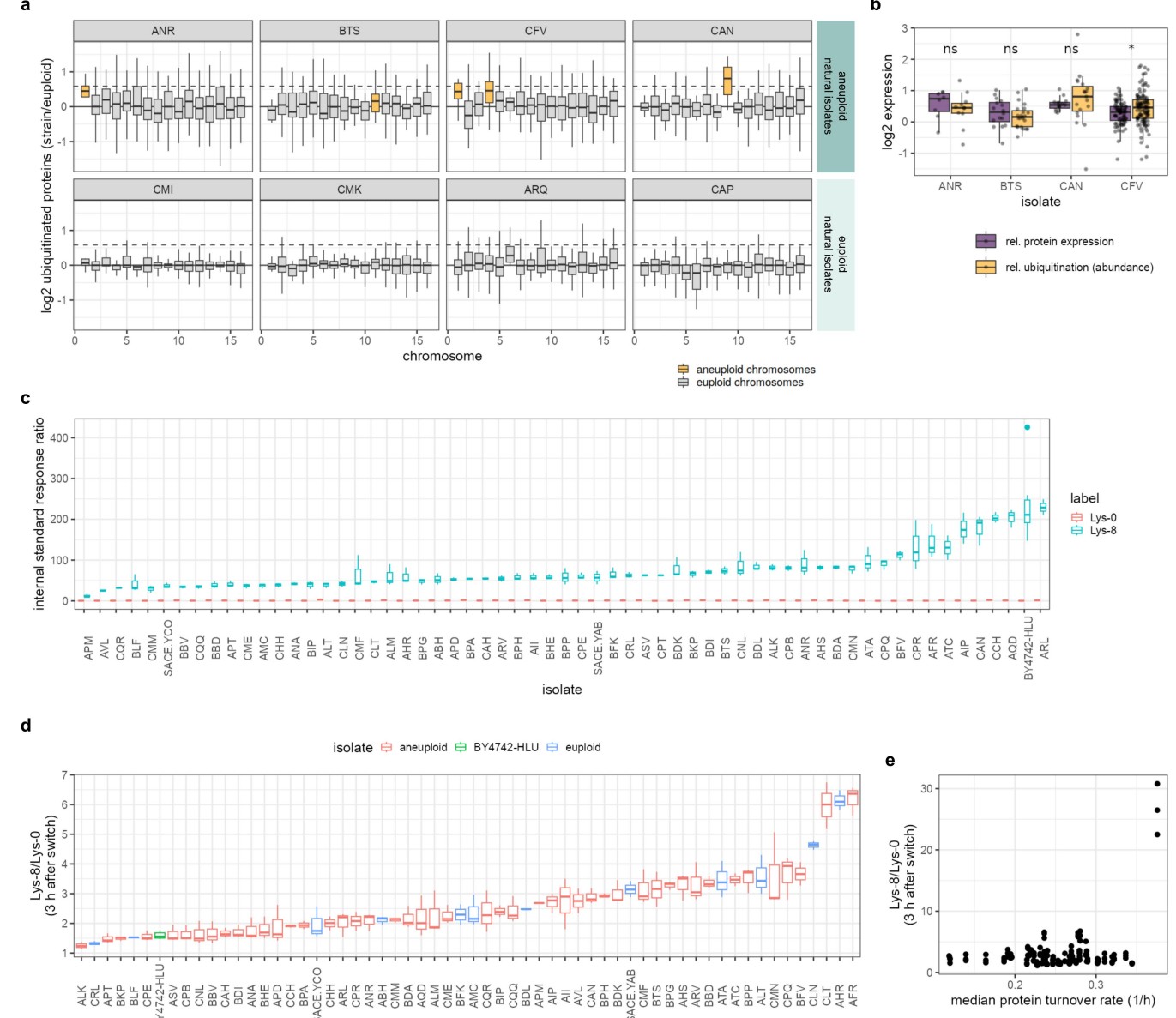

**Extended Data Fig. 7 | Ubiquitinomics and lysine uptake of natural isolates.**
**a**, Chromosome-wide distributions of the relative abundance of ubiquitinated proteins quantified by diglycine remnant profiling in aneuploid and euploid natural isolates. Proteins carrying ubiquitin side chains were used to determine normalized log2 protein abundance ratios (comparing the expression of each gene per isolate with the median expression of the respective gene across the four euploid isolates). Aneuploid chromosomes are highlighted in yellow, euploid chromosomes are coloured grey (chr. 1: 7 <= n <= 11; chr. 2: 42 <= n <= 72; chr. 1: 13 <= n <= 22; chr. 4: 62 <= n <= 112; chr. 4: 36 <= n <= 58; chr. 6: 11 <= n <= 18; chr. 7: 62 <= n <= 88; chr. 8: 24 <= n <= 44; chr. 9: 12 <= n <= 25; chr. 10: 29 <= n <= 52; chr. 11: 23 <= n <= 43; chr. 12: 56 <= n <= 81; chr. 13: 44 <= n <= 76; chr. 14: 32 <= n <= 64; chr. 15: 53 <= n <= 77; chr. 16: 40 <= n <= 64 proteins per chromosome). Outliers are not shown to improve readability of the distributions. The solid and dotted horizontal lines mark 0 and 0.58, respectively. **b**, Comparison of relative protein expression (purple) and ubiquitination (as in **a**, yellow) levels of proteins encoded on aneuploid chromosomes in four natural isolates. Statistical significance of two-sample, two-sided t-test between relative

protein expression and relative ubiquitination per isolate is indicated (ns: p > 0.05, *: p = 0.018). **c**, Ratio (internal standard response ratio) between unlabelled (Lys-0, red) or labelled (Lys-8, cyan) intracellular lysine and an internal quantification standard (Lys-4, Methods) in prototroph natural isolates and a lysine-auxotroph laboratory strain (BY4742-HLU) after continuous growth in minimal medium (SM) supplemented with 80 mg/L labelled lysine (Lys-8). **d**, Ratio between labelled and unlabelled intracellular lysine (Lys-8/Lys-0) in prototroph natural isolates and a lysine-auxotroph laboratory strain (BY4742-HLU) three hours after switching from growth in unlabelled (SM + 80 mg/L Lys-0) to labelled (SM + 80 mg/L Lys-8) medium. Aneuploid (red) and euploid (blue) natural isolates, as well as the lab strain (green) are highlighted. For **c** and **d**, n = 2 for ATC, n = 6 for BY4742-HLU, n = 3 biological replicates for all other isolates. **e**, The relationship between Lys-8/Lys-0 ratios from **d** and the median protein turnover rate per isolate (black dots) shows no correlation. In box plots, the centre marks the median, hinges mark the 25th and 75th percentiles and whiskers show all values that, at maximum, fall within 1.5 times the interquartile range.

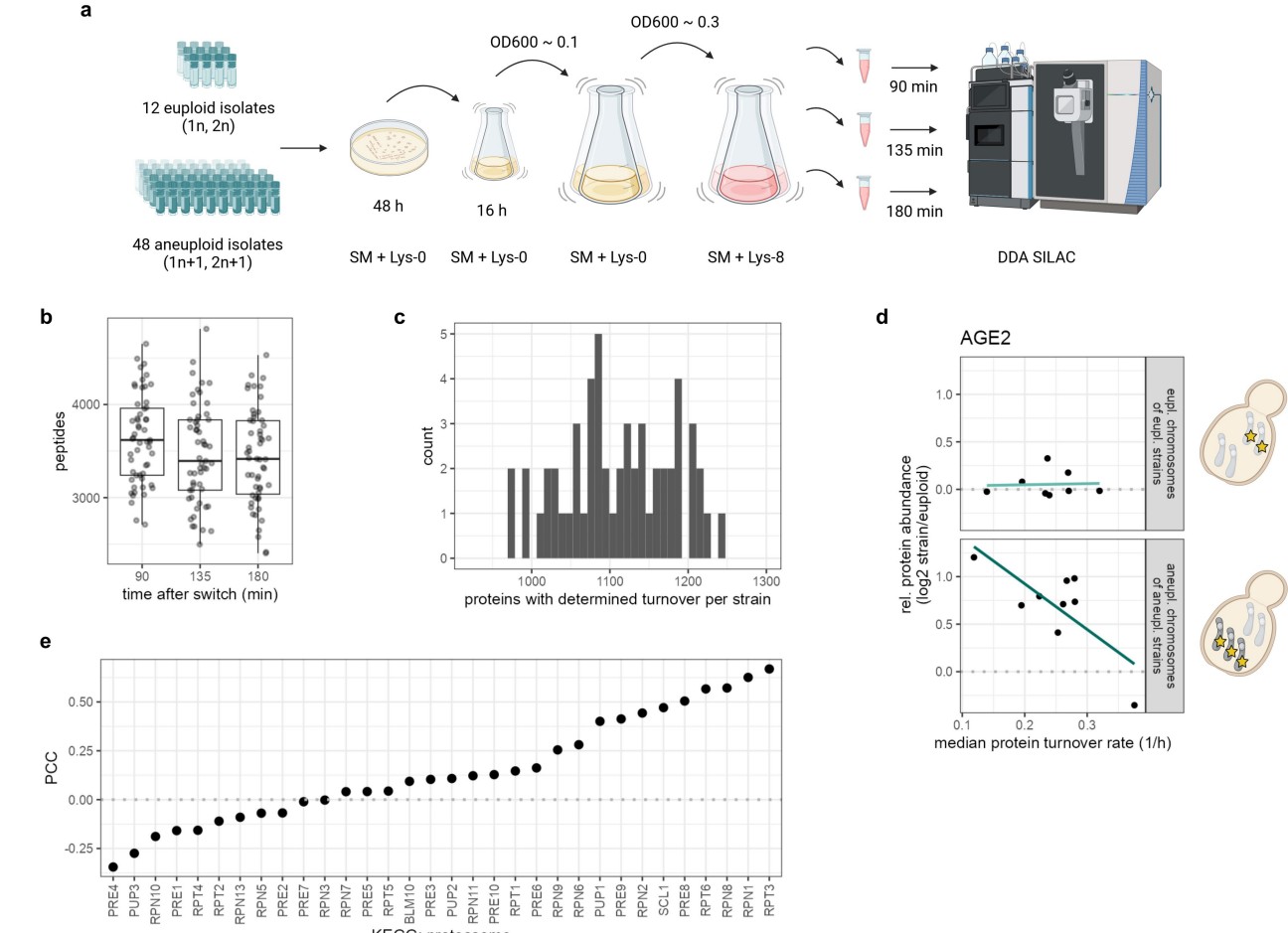

**Extended Data Fig. 8 | Protein turnover quality control and relationship between protein abundance and turnover. a**, Experimental design of dynamic SILAC turnover experiments. Aneuploid or euploid isolates (n = 60) were grown on minimal medium agar supplemented with unlabelled lysine (SM + Lys-0), pre-cultured for 16 h, and then diluted to a low OD. After reaching early mid-log phase (~OD 0.3), cells were transferred into minimal medium supplemented with heavy-isotope labelled lysine (SM + Lys-8) and samples were collected and prepared for proteomics after 90, 135, and 180 min. **b**, Box plots depicting the number of peptides per isolate (dots, n = 56) with valid SILAC ratios per time point after the switch from unlabelled to Lys-8-labelled SILAC medium. Box plot hinges mark the 25th and 75th percentiles and whiskers show all values that, at maximum, fall within 1.5 times the

interquartile range. **c**, Distribution of the number of proteins per isolate (n = 55) for which turnover rates were determined. **d**, Pearson correlation between relative Age2 protein expression and isolate turnover rate across euploid isolates (top, light teal line) as well as aneuploid chromosomes of aneuploid isolates (bottom, dark teal line). The dotted grey line denotes 0. **e**, Pearson correlation coefficients (PCC) between relative protein abundances and the overall protein turnover rate across euploid isolates for genes annotated as structural components of the proteasome (KEGG, identified via gene set enrichment analysis of ranked PCCs, p = 0.0019/FDR: 0.16). PCC were calculated across all euploid isolates for which isolate-wise turnover rates could be determined (n = 8).

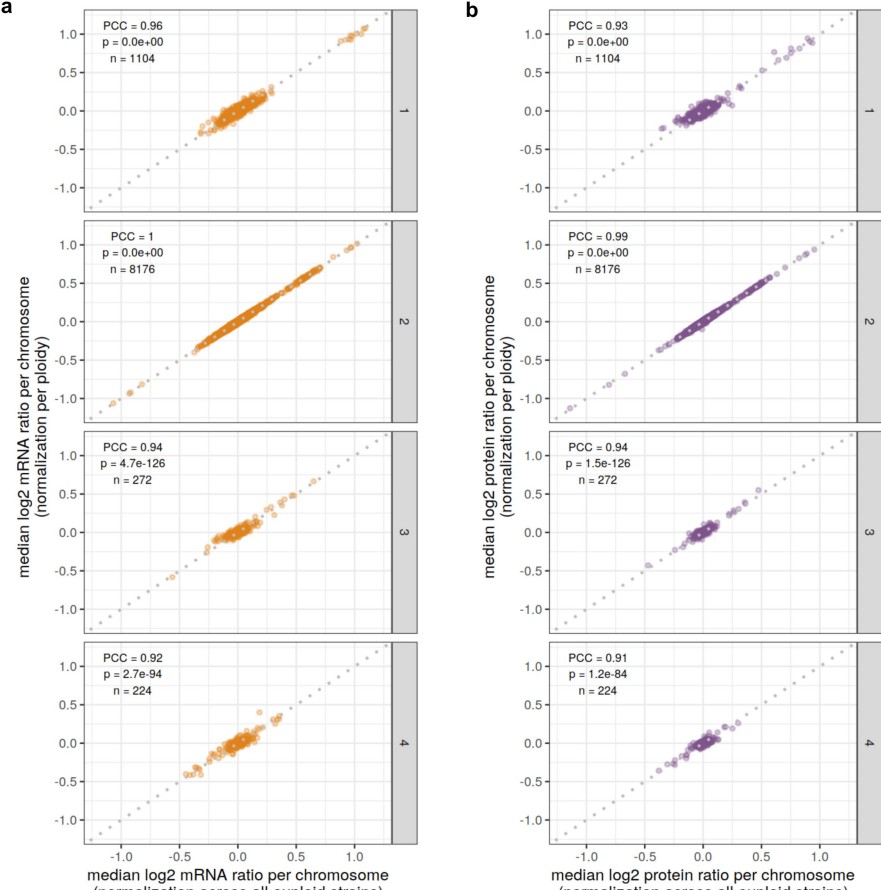

**Extended Data Fig. 9 | Comparison of all-euploid isolate versus ploidy-wise normalization procedures. a**,**b**, Pearson correlation coefficients (PCC) between the different normalization procedures were determined per ploidy (haploid to tetraploid, vertical panels) for median chromosomal log2 mRNA (**a**, orange dots) or log2 protein (**b**, purple dots) expression values. Euploid and aneuploid isolates were included in the analysis. The number of chromosomes per ploidy, so the number of data points included in the correlation analysis, is denoted by n, and p refers to the p-value of the two-sided correlation test. The grey dotted line indicates a correlation of 1 ($y = x$).

**a**

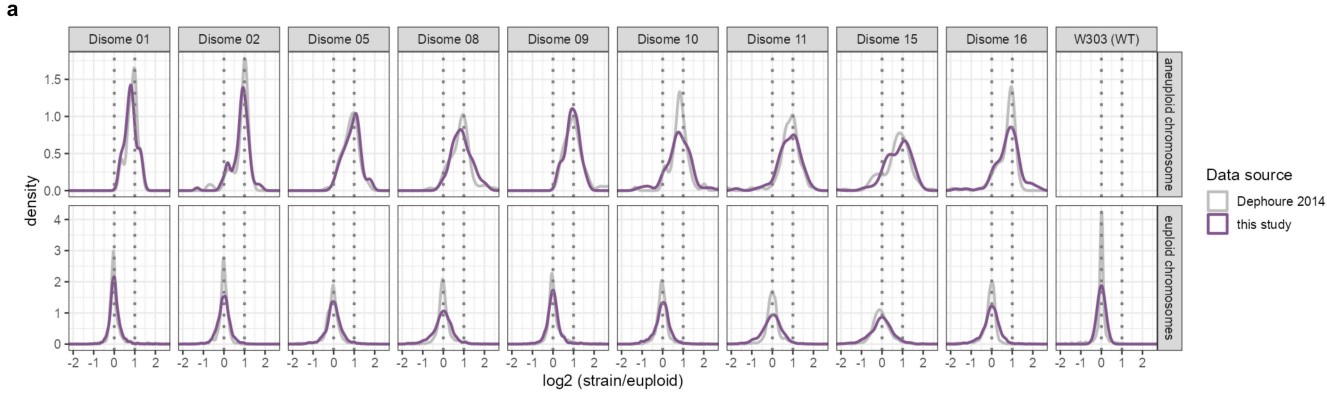

**b**

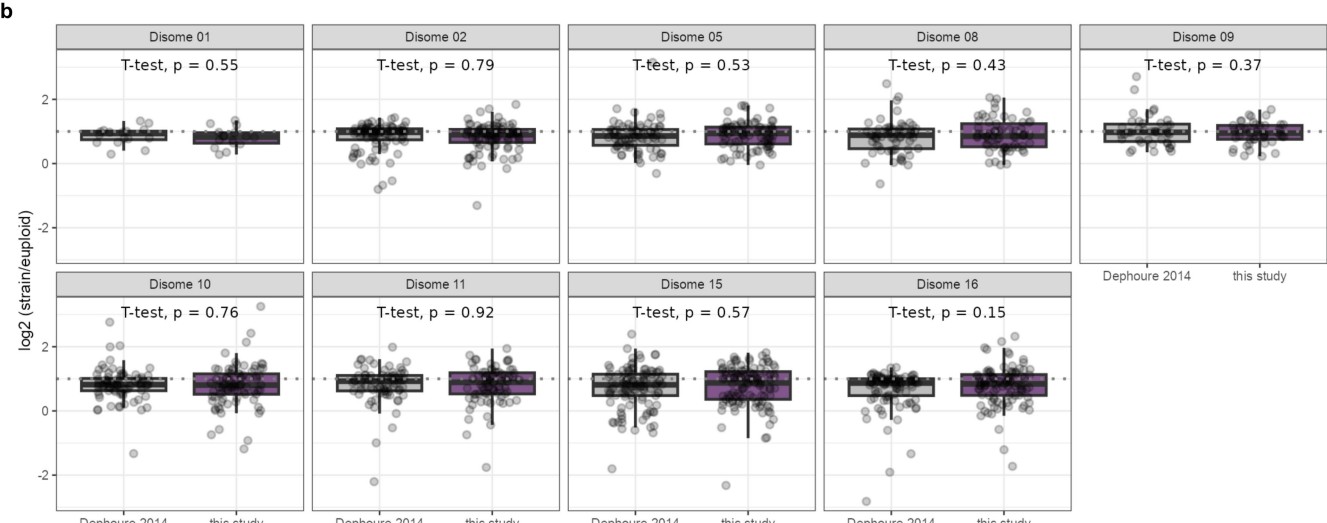

**Extended Data Fig. 10 | Comparison of the attenuation of proteins located on aneuploid chromosomes between a previously published SILAC proteomic dataset on lab-engineered synthetic disomes and this study.**
**a**, Distribution of relative protein expression levels (log2 strain/euploid) on aneuploid (top) and euploid (bottom) chromosomes across strains of the disomic strain collection. Distributions based on measurements by Dephoure et al.[10] are displayed in grey, the ones based on this study in purple. The dotted lines mark expected log2 protein expression levels for proteins located on euploid chromosomes (0) and ones located on duplicated chromosomes (1).

**b**, Comparison of the median log2 protein expression of proteins located on aneuploid chromosomes of disomic strains between the data published by Dephoure et al.[10] and this study (two-sample, two-sided T-test, disome 01: n = 16; disome 02: n = 96; disome 05: n = 81; disome 08: n = 65; disome 09: n = 45; disome 10: n = 86; disome 11: n = 79; disome 15: n = 114; disome 16: n = 99 proteins). In the box plots, the centre marks the median, hinges mark the 25th and 75th percentiles and whiskers show all values that, at maximum, fall within 1.5 times the interquartile range.

# Reporting Summary

## Statistics

For all statistical analyses, confirm that the following items are present in the figure legend, table legend, main text, or Methods section.

| n/a | Confirmed | |
|---|---|---|
| ☐ | ☒ | The exact sample size ($n$) for each experimental group/condition, given as a discrete number and unit of measurement |
| ☐ | ☒ | A statement on whether measurements were taken from distinct samples or whether the same sample was measured repeatedly |
| ☐ | ☒ | The statistical test(s) used AND whether they are one- or two-sided *Only common tests should be described solely by name; describe more complex techniques in the Methods section.* |
| ☒ | ☐ | A description of all covariates tested |
| ☐ | ☒ | A description of any assumptions or corrections, such as tests of normality and adjustment for multiple comparisons |
| ☐ | ☒ | A full description of the statistical parameters including central tendency (e.g. means) or other basic estimates (e.g. regression coefficient) AND variation (e.g. standard deviation) or associated estimates of uncertainty (e.g. confidence intervals) |
| ☐ | ☒ | For null hypothesis testing, the test statistic (e.g. $F$, $t$, $r$) with confidence intervals, effect sizes, degrees of freedom and $P$ value noted *Give P values as exact values whenever suitable.* |
| ☒ | ☐ | For Bayesian analysis, information on the choice of priors and Markov chain Monte Carlo settings |
| ☒ | ☐ | For hierarchical and complex designs, identification of the appropriate level for tests and full reporting of outcomes |
| ☒ | ☐ | Estimates of effect sizes (e.g. Cohen's $d$, Pearson's $r$), indicating how they were calculated |

*Our web collection on statistics for biologists contains articles on many of the points above.*

## Software and code

Policy information about availability of computer code

| Data collection | Proteomic raw data were collected using Analyst 1.8.1. Metabolomic raw data were collected using Mass Hunter B.07.01. |
|---|---|
| Data analysis | Raw mass spectrometric data for natural isolates and disomic strains were processed using DIA-NN version 1.7.12. Ubiquitinomics data and time course proteomics data were processed using DIA-NN version 1.8. Raw mass spectrometric data of turnover samples were processed using MaxQuant version 1.6.7.0. Pre-processing and statistical analyses were performed using R 3.6. The following dedicated packages were used during analysis: diann 1.0.1, rstatix 0.7.2, data.table 1.14.2, org.Sc.sgd.db 3.18.0, ComplexHeatmap 2.18.0, pROC 1.18.5, edgeR 3.38.4. |

For manuscripts utilizing custom algorithms or software that are central to the research but not yet described in published literature, software must be made available to editors and reviewers. We strongly encourage code deposition in a community repository (e.g. GitHub). See the Nature Portfolio guidelines for submitting code & software for further information.

## Data

Policy information about availability of data

All manuscripts must include a data availability statement. This statement should provide the following information, where applicable:
- Accession codes, unique identifiers, or web links for publicly available datasets
- A description of any restrictions on data availability
- For clinical datasets or third party data, please ensure that the statement adheres to our policy

Raw and processed mass spectrometry data of the natural collection are deposited in Massive under ID MSV000090435. Data associated with the disomes

# Research involving human participants, their data, or biological material

Policy information about studies with human participants or human data. See also policy information about sex, gender (identity/presentation), and sexual orientation and race, ethnicity and racism.

| | |
|---|---|
| Reporting on sex and gender | -NA |
| Reporting on race, ethnicity, or other socially relevant groupings | -NA |
| Population characteristics | -NA |
| Recruitment | -NA |
| Ethics oversight | -NA |

Note that full information on the approval of the study protocol must also be provided in the manuscript.

# Field-specific reporting

Please select the one below that is the best fit for your research. If you are not sure, read the appropriate sections before making your selection.

☒ Life sciences ☐ Behavioural & social sciences ☐ Ecological, evolutionary & environmental sciences

For a reference copy of the document with all sections, see nature.com/documents/nr-reporting-summary-flat.pdf

# Life sciences study design

All studies must disclose on these points even when the disclosure is negative.

| | |
|---|---|
| Sample size | We measured the entire isolate collection of the 1011 Saccharomyces cerevisiae genomes project. |
| Data exclusions | Gene copy numbers for natural isolates were downloaded from the 1002 Yeast Genome website, and the following loci were excluded: ribosomal DNA, Ty elements, RTM loci, ORFs located on the 2-micron plasmid, mitochondrial ORFs, and non-reference material. Furthermore, the table was filtered to only retain genes with non-zero and non-missing values for further analyses. Chromosome copy number status for all engineered disomic strains was confirmed by Torres et al. 2007, meaning all disomic strains used in our study were haploid with indicated "disomic" chromosomes duplicated. One exception was Disome 13: despite published mRNA expression values being available and proteomics data having been measured in our experiments, Disome 13 was excluded from all analyses due to having undergone whole-genome duplication when reaching our laboratory (private communication, J. Zhu). For 761 isolates, both proteomes (this study) and transcriptomes were available, and for 759 isolates, gene copy number information from Peter et al. 2018 was available. Data for gene copy number, mRNA expression, and protein abundances were matched by strain name and systematic ORF identifier for both the natural isolate collection and the disomic strain collection. Only genes for which values for gene copy number, transcript, and protein levels were available were used for analyses. We noticed that a number of strains in the natural isolate collection exhibited a mismatch between the median gene copy number per chromosome and the assigned aneuploidy, most likely attributable to segmental aneuploidies, shorter gene copy number variations, or algorithm-specific thresholds used for aneuploidy determination. We excluded all strains (n = 80) containing one or more of those "mismatched" chromosomes from our analysis. |
| Replication | The proteomic dataset for the small strain library (disomic lab-engineered strains) was acquired in triplicates, while the large dataset (> 1000 isolates) was acquired once. Here we determined reproducibility and technical precision through replicate injections of a reference sample (77 injections), similar to previous studies (e.g. Messner et al., Cell, 2023). |
| Randomization | The strains were arrayed in a randomized order. |
| Blinding | Blinding was not applicable since data acquisition was performed in an automated and randomized manner. |

# Reporting for specific materials, systems and methods

We require information from authors about some types of materials, experimental systems and methods used in many studies. Here, indicate whether each material, system or method listed is relevant to your study. If you are not sure if a list item applies to your research, read the appropriate section before selecting a response.

## Materials & experimental systems

| n/a | Involved in the study |
|-----|----------------------|
| ☒ | ☐ Antibodies |
| ☒ | ☐ Eukaryotic cell lines |
| ☒ | ☐ Palaeontology and archaeology |
| ☒ | ☐ Animals and other organisms |
| ☒ | ☐ Clinical data |
| ☒ | ☐ Dual use research of concern |
| ☒ | ☐ Plants |

## Methods

| n/a | Involved in the study |
|-----|----------------------|
| ☒ | ☐ ChIP-seq |
| ☒ | ☐ Flow cytometry |
| ☒ | ☐ MRI-based neuroimaging |

