## [Peer Review File · Nature]

Manuscript Title: Natural proteome diversity links aneuploidy tolerance to protein turnover

Reviewer Comments & Author Rebuttals

Reviewer Reports on the Initial Version:

Referees' comments:

Referee #1 (Remarks to the Author):

The review of manuscript “The natural diversity of the yeast proteome reveals chromosome-wide dosage compensation in aneuploids” by Muenzner et al.

The presented manuscript focuses on understanding the consequences of abnormal chromosomal numbers in naturally occurring yeast aneuploid strains. The author team performs an impressive tour-de-force analyzing the transcriptomes and proteomes of several hundreds of aneuploid natural strains. The elegant evaluation and visualization make the data accessible, which is indeed a difficult task. The authors demonstrate that the effects of aneuploidy on cis-gene expression is largely dosage compensated on proteome level, while the transcriptome generally scales with the chromosome copy numbers. By elegant exploitation of proteomic analysis of ubiquitinated lysine residues they show that there is more ubiquitinated peptides in four natural aneuploid isolates and these are enriched for proteins encoded on the aneuploid chromosomes. This is further complemented by findings that aneuploid yeast degrade model proteasome substrate faster than euploid strains.

This is an interesting and comprehensive study. However, there are several points, which need to be clarified and which bring some interpretations into questions. Also, the way how the data is interpreted and the entire tone of the manuscript is somewhat troublesome. The authors repeatedly state that this is the largest data set, and the first time that dosage compensation has been demonstrated, that they solve an ongoing conundrum, and that datasets from other species were too small. They propose in the discussion that possible further studies may reveal that “chromosome-wide dosage compensation could be a mechanism conserved across species”. The problem here is that the authors ignore a significant number of publications on this topic that clearly show that human aneuploidy is dosage compensated on proteome level in engineered aneuploids as well as in cancer samples (see below). Additionally, the authors could have done more analysis with their rich data, which would bring further insight into the observed phenomenon.

Below are the major aspects which should be addressed.

1. Throughout the manuscript, the authors present the previous literature as if the dosage compensation on proteome level has never been demonstrated. This is likely originating from the confusing and to a certain extent inconclusive previous works on aneuploidy in budding yeast. It is indeed an honorable task to bring more light into this difficult question that has been obscured by disagreements of several publications on this topic. However, it should be noted that the laboratory of Angelika Amon/Eduardo Torres repeatedly showed for the very same set of disomic lab strains that there is dosage compensation on proteome level, but not on transcriptome level. This is suggested first

by Torres et al (2007, Science, 2010, Cell) and clearly stated in Dephoure et al (2014, Elife, see figures 2B, 3B). Other laboratories did not reach the same conclusion, which is indeed interesting. The discrepancy among other publications might be for example due to different pre-processing/normalization procedures, and it would be important for the field if the authors would clearly identify the source of confusion on the topic of dosage compensation in budding yeast.

2. The authors focus on yeast strains. However, it is incorrect to claim that the dosage compensation could not be clearly shown in other species. Paper by Stinglele et al (MSB, 2012) shows dosage compensation on proteome level (enriched, among others, for subunits of macromolecular complexes) in six different engineered human cell lines with extra chromosomes. Donnelly et al (EMBO J, 2014) shows that the proteasome activity is increased in human aneuploid cells, and McShane et al (Cell, 2017) shows that proteasomal degradation is responsible for a significant fraction of dosage compensation in aneuploid human cells. Dosage compensation on protein levels was also recently demonstrated in human monosomic cell lines by Chunduri et al (Nat Comms 2021) and analysis of transcriptome and proteome from 375 cancer cell lines confirmed genome-wide dosage compensation on proteome level (). There are several other papers showing various aspects of dosage compensation on protein level in cancer cell lines. None of these papers is cited. From all the work on human cell lines, only the paper by Liu et al focusing on trisomy 21 is mentioned (Nat Comms, 2017). The authors of course can choose to focus on yeast strains, but then they should not make these generalized claims.

3. A key issue with analysis of data from aneuploid strains is the normalization. The authors normalize the data to an “ideal euploid”, which is a median of all euploid strains regardless their ploidy. This requires the assumption that proteome scales linearly with ploidy, which has never been demonstrated. In contrary, there is evidence that with increasing ploidy the proteome is sub-scaling (Gemble et al, Nature 2022; <https://www.biorxiv.org/content/10.1101/2021.05.06.442919v2>). Thus, the authors should normalize haploid aneuploids to haploid euploids, diploid to diploid euploids etc. Did they employ the normalization to the median euploid wild type for all their analysis? Did they perform forced normalization to 0? Was the forced normalization performed including the data from aneuploid chromosomes, or excluding them, or euploid and aneuploid chromosomes separately?

4. The dosage compensation might be different in cells with higher ploidy than in haploids – one can imagine that $4N+1$ results in different response than $1N+1$ (this is also suggested by data from Dephoure et al, eLife 2014). This is also suggested here in the plot 3a – the dosage compensation in $3N+1$ seems smaller than in other datasets. The authors need to rigorously test this.

5. The dataset allows the authors to address some further important questions. For example, could it be that the increased proteasomal degradation and increased dosage compensation correlates with the degree of aneuploidy (=number of imbalanced chromosomes), or type of aneuploidy (gains vs. loss)?

6. Comparison of engineered and naturally occurring disomic strains revealed increased dosage compensation on protein level in naturally occurring strains. What are the additional proteins that become dosage compensated? In engineered strains typically subunits of macromolecular complexes are dosage compensated. Is there any subgroup of proteins enriched among the dosage compensated proteins in naturally occurring aneuploids?

7. In human monosomic cells as well as in *Drosophila* lacking substantial parts of a chromosome, some dosage compensation has been observed also on a transcriptome level. Is this the case also in the naturally occurring aneuploid yeast lacking chromosomes? Related to this - in Fig. 2a, there are very

clearly some strains which show an increased genome signal, but reduced mRNA signal (e.g. the first aneuploid on the top, chromosome 1, generally chromosome 1 in most strains). Yet, the global quantification shows no dosage compensation on mRNA level. How comes?

8. Figure 1 is not informative. Panels a and b are well described in the text and could be in supplementary (or left out). It is not clear what is the purpose of panel c – it does not illustrate anything relevant for the manuscript. Panels d and e would be useful, but are too small. In d, it would be better to express the fraction of aneuploids in % (and state the total number of strains in each category). Panel e – is the distribution of aneuploid chromosomes the same for all different ploidy levels?

9. The K—GG-modified peptides should be compared with the dataset from McShane et al (Cell, 2017) to test whether similar categories of proteins are degraded by proteasome in aneuploid yeast strains as in aneuploid human cell lines.

10. Fig. 3c is a nice summary, but a fair comparison would be natural disomic strains vs laboratory disomic strains (unless the authors demonstrate that the dosage compensation is independent of basal ploidy).

11. Fig. 4a, b – the enrichments scores and log₂ fold changes depicted are very small (<0.1 in 4b!). The scale on X and Y axes should be identical on the left and right plot in 4b to allow proper comparison. For fair comparison, only disomic natural strains should be compared with disomic engineered strains. In human aneuploid cells lines, the increased expression of proteasomeal subunits correlates with the amount of extra DNA (Donnelly et al, EMBO J 2014). Is it the case here as well?

12. Discussion, first paragraph - the authors claim that their datasets is unique, advances proteome technology, and such studies have not been previously possible due to the technical limitations of proteomic platforms. This is a false statements. Comparable datasets have been obtained, e.g. the transcriptomes and proteomes of hundreds of cancer cell lines from the CCLE. Their claim of highly complete proteome for 796 strains cannot be validated, because the full data set is not provided.

13. The discussion is largely overinterpreting the data and making sweeping generalized statements which are not justified. Besides the already above mentioned examples the authors claim that “the increased survival of aneuploid cells under stress conditions is often dependent on one or more key stress response genes . . . “. For this they cite exactly one original papers. This does not sound like a general phenomenon.

Technical comments

1. The authors measure 933 strains, from which only data from 613 were used where the transcriptome and proteome could be matched. What exactly is the problem with the other data? Why could not be transcriptomes and proteomes of 1/3 of the samples matched? The authors state “Importantly, the exclusion of these strains did not bias the dataset for the study of aneuploidy” (Line 178, Page 5). How do they know? No data is shown to support this claim.

2. What is “biological signal”? (line 164, page 5)

3. “1576 protein quantities” seems rather low (~ 25 % of the expressed yeast genome). The CCLE proteome allowed identification of significantly higher fraction of the expressed genome (Nusinow, Cell 2020). What is the reason?

Minor comments

1. The figures are largely visually appealing and clear. However, the font is often very small and difficult to read in a print. Also, white text on a light green backgrounds is difficult to read. It would be great if

the authors would adapt the figures for better legibility. Fig. 1 should be changed.

2. Were the disomic strains from Amon lab newly re-sequenced? In Methods it is stated that the status was confirmed by Torres et al, but this has been several years ago and aneuploid strains are genomically unstable and euploids regularly outgrow the aneuploids.

3. In the case of 95 aneuploid strains, was it always whole chromosome aneuploidy? Or were also segmental aneuploids considered?

4. Fig. 2a, the euploid natural strains – are these all haploids, or all diploids? Could authors eventually also mark the ploidy? The same for the aneuploids – it would be useful to distinguish the basal ploidy.

5. Figure 3a – the graphical depiction of the karyotypes is confusing. The best would be just to write it next to the plots. It would also save space. Haploids and diploids should be separated (bottom panel).

6. Line 333, page 11 : “. . . or related to transcription, which is altered in aneuploid to the aberrant chromosome number”. What do the authors mean? Do they now talk about the general transcriptional response to aneuploidy or the transcriptional changes of the genes encoded on the numerically aberrant chromosomes? They should change the sentence to avoid misinterpretations.

7. Fig. S4 – there are no dark grey points visible.

Referee #2 (Remarks to the Author):

The question of how aneuploidy affects cell physiology and promotes disease has a long history going back to the incredible work of Boveri over 100 years ago. Over the last 15 years, yeast has been the setting for many of the major advances and vigorous debates on this issue. It is a very interesting and also fundamental problem. But is there anything new to say about it? This paper makes the pretty interesting claims that wild strains of yeast follow different rules of aneuploidy tolerance than laboratory strains (i.e., superior dosage compensation), and that their more robust capacity in this relates to a physiological adaptation involving the proteasome (i.e, higher proteasome levels). Interesting that the effect maps to the proteasome and not to the chaperone pathway. The interesting panels 4B and 4C are key to the argument. Fairly simple argument, which is good, and I think that if one can make it stick it can be a major paper, with good novelty and generality. The issue of how domesticated laboratory strains differ from wild strains is highly interesting and clearly germane not only to *S. cerevisiae*.

I will focus on a key problem with the paper, maybe the key problem: validation of the proteasome effect. In general the level of sophistication of the analysis tends to fall off when the argument reaches this point. Here are some issues:

1. Why do the wild strains show a perturbation of proteasome levels? This is essential to resolve and should not be terribly difficult I think. The proteasome is under complex control but as far as general controls over the level of the proteasome, it is at least thought to be pretty simple. The main control is via the transcription factor Rpn4, and a secondary control is via proteaphagy. Rpn4 has targets other than the proteasome however, so it should already be clear from the mRNA and protein data whether

an Rpn4-driven compensatory response is operating. The other possible Rpn4-related hits may not assort into cookie-cutter GO terms, it has to be examined gene-by-gene. Also, if via Rpn4, proteasome upregulation would be readily evident at the RNA level. I may have missed this, but I didn't find any comment on this one way or another in the main text.

So to be clear, it is critical to resolve whether Rpn4 has a role in this phenomenon. If not the authors might examine proteaphagy or whether a novel mechanism could be in play. One might look at Rpn4 itself, but that can be tricky as it is expressed at low levels and may have therefore fallen out of many of the datasets. Also one may see an effect on Rpn4 mRNA levels (a bit easier) but much of the regulation of Rpn4 is post-translational. If there is any suggestion of Rpn4 involvement, it raises the opportunity of genetic manipulation of Rpn4 in these strains to test specific hypotheses. For example, knocking out the RPN4 gene may normalize proteasome levels—easy experiment (also reducing state-state levels a bit in euploids) that could go right to the heart of things.

2. Fig 4B (left) is interesting and important and looks good to the eye, but I don't feel comfortable in my understanding of it. How many euploid strains are involved here, more importantly how many aneuploid? Was only some fraction of the aneuploid strains used for this? Only the haploids? All of the haploids? Or only haploids that are aneuploid for certain chromosomes (would not be surprising...)? Basically I read the legends as vague. More generally can one say that the aneuploid strains that were selected for presentation were not cherry-picked on the basis of a property under examination?---I have no reason to believe that but at the same time the text did not seem to clearly exclude that scenario. How many technical replicates are involved (4B left)?

3. Rpn4 is present on chromosome 4. Did chromosome IV aneuploids show any perturbation of proteasome levels?—I suspect so. Note that there is chromosome IV aneuploidy in the experimental (wild) aneuploid strain CFV. How does Fig 4B (left) look when chromosome IV aneuploids are removed from the pool? Are wild chromosome IV aneuploids better adapted to the aneuploid state?

4. I would want to know the identity of each dot, each protein, in 4B (left). That may not fit into the main text of course but would be a good supplemental figure; same volcano plot, every blue dot labeled. For example, it could be that the positive hits are all from some specific part of the proteasome—there are 33 distinct gene products, counting only integral subunits. Are all 33 in there? I can't tell. And I would have a supplementary table showing those data as well as those for the many proteins that work hand in glove with the proteasome (I would be happy to prepare a list in that would help). I assume that no ubiquitin ligases or deubiquitinating enzymes seem to be affected—correct?

5. 4B and 4C are both interesting, very connected in spirit, but I'm not sure how they correlate. The questions are related to those directly above, how and why individual strains were chosen for analysis although the uncertainty relates mostly to 4B—without knowing that we can't make a confident comparison. Perhaps some of the information that I'm concerned with really is in the text, but if it is then at least I would think that these points could be explained more clearly to the reader.

6. SI Fig 3c is another important component of the argument although put in the supplement. Not a hard experiment but it needs work. On the one hand it is not good technically, but also, even apart from that it would not be convincing. As above, it is not clear to me why we are looking at these four strains and whether they are as representative as one would hope—perhaps they are, I just can't tell. The third set of plots (ARQ-CFV) is of no use, there simply have to be more replicates and it is not hard to see that the one determination there for the euploid control is likely to be inaccurate. And for plots like this the y axis should extend to zero. Actually better to have more than three replicates for each set of samples, as the effect size is small and the sample to sample replicate variation not small. The second set is not showing enough of a difference to be convincing. Why do this experiment so casually? There are no error bars and the legend speaks of a mean but no mean is shown. It is fine to measure proteasome activity using the suc-LLVY-AMC assay. But it is essential to perform the assay +/- proteasome inhibitor and to subtract the +inhibitor values from the -inhibitor values. Also it is recommended to perform the assay in the presence and absence of 0.02% SDS; in the presence of SDS your readout is essentially of core particle levels as the proteasome gate will be wide open.

7. I felt uncomfortable with the pairing of wild strains according to their genetic closeness. Again this is likely more a matter of presentation than of a methodological deficiency, it is just that I didn't find an objective measure of that shown, making it hard to evaluate the utility of these pairs. Also it would be reassuring to know that these critical pairs of comparable strains are in each case well matched in their growth properties, as well as can be expected.

8. The difference between euploid and aneuploid strains highlighted in this work are subtle and I would recommend care as to whether they could have a metabolic basis. I would highly recommend a set of control experiments in which determinations of choice (eg that of SI Fig 3C or some proteomic analysis, which may give more precise values) are read across the span of a growth curve, say from OD 0.4 to OD 3.0 (before and after the diauxic shift). As it is I had a hard time figuring out at what growth stage most of the experiments were done at and whether this was adequately controlled.

9. Aggregates in yeast are often visualized using Hsp104-GFP fusion proteins. It may be good to give that a try in some of the strains. It is easy and has the potential to validate the idea that these strains have impaired proteostasis.

Referee #3 (Remarks to the Author):

In this manuscript, Muenzner et al. characterize the proteome of a large collection of wild and industrial yeast strains, and examine how the chromosome-wide dosage is compensated in aneuploids. For this study, the authors established a platform to quantify the proteome of nearly 800 strains. Several innovative steps were implemented to complete this tour de force that led the quantitation of ~1500 proteins across the examined strains. Notably, among the over 600 strains that could be matched to

previously sequenced genomes (Peter J et al Nature 2018), nearly 100 strains were aneuploid, which is surprising given the fact lower cell fitness was observed in aneuploid yeast lab strains (Pavelka et al Nature 2010). The authors then examined in more details how the proteome is buffered in aneuploids, meaning how protein levels expressed from sur-numerous chromosomes are dampened. Similar to previous work (Dephoure N et al eLife, 2014), the authors found that this phenomena is controlled post-translationally and not at the mRNA level. They then showed that this buffering correlates with higher activity of the ubiquitin proteasome system (UPS). This study is interesting and unique. However, the main observations and conclusions are not entirely novel. While the authors show that the buffering may be greater in natural isolates in comparison to lab strains, the notion that many sur-numerous proteins are degraded or aggregate, especially if they are part of stable protein complexes is not new. Maybe the conclusions of this study would be more distinctive if the focus was on the overall proteome tuning, and how some proteins may be differentially expressed in natural isolates.

Dephoure et al eLife, 2014 previously showed that a significant proportion of proteins are expressed at lower levels than expected in disomic lab strains. This is notably shown in Figure 2B of the eLife publication, in which there is a clear binomial distribution. Similarly, the concept that the UPS is involved is not new, as Torres et al., Cell 2010 showed that aneuploid cells have an increased reliance on the UPS, and levels of proteasome subunits were shown to be more elevated in disomic lab strains (Figure 6 of Dephoure N. et al., eLife 2014).

Figure 2 and 3 could be combined and better quantitation should be provided. One issue is that, in Figure 2A, it appears that a large portion of strains have approximately equal mRNA and protein enrichment. Including additional supplemental data with quantitative data could address this issue. Interestingly the data in Fig 2B shows a large variability at the mRNA levels in AHS strains, suggesting a potential larger buffering for other proteins; besides AHS, more examples should be provided. Fig 3C may be more appropriate for a supplemental material.

Curiously, disomic lab strains analyzed in this manuscript seem to display a lower buffering capacity, in comparison to the previously published results (Dephoure N et al eLife 2014). For instance, many proteins located on disomic chromosome 9 display levels similar to control cells in Figure 1D of the 2014 publication, which is not the case in the data presented in Figure 2B of the current manuscript. One potential issue is that natural and disomic lab strains in this study were grown in different media (synthetic minimum without amino acids vs. SD-HIS+G418), which is somewhat problematic.

Enrichment of K-GG on proteins encoded on aneuploid chromosome may not be a true enrichment. The increase of ubiquitination may be simply due to higher protein levels. The authors need to normalize to the abundance of the proteins from which the K-GG peptides originate, to account for the increased abundance of proteins. They should also try to better correlate observed reduced levels at the proteome level in comparison to increased ubiquitination. Significance of this potential enrichment (Figure 4C) and of the elevated proteasome activity (Figure S3C) should be examined with statistical tests. Perhaps, one potential unique future avenue for this study would be to delineate how elevated proteasomes are regulated in aneuploid cells, and how come chaperones and other elements of the protein homeostasis

network are not solicited.

It should be noted that many of the examined strains are not necessarily “natural” but have been used by humans for many different processes (e.g. fermentation), in which aneuploidy may have been selected to cope with the additional stresses. Have the authors examined whether aneuploidy is more frequent in isolates associated to industrial activities vs. natural habitats?

Unless I missed it, growth rate of aneuploid cells in the 1011 collection is only discussed in the introduction (line 106) and the results presented in Figure S3A is not discussed anywhere else. It should be noted that in some cases, growth differences between lab disomic strains can be very subtle, and only reliably quantified when both strains are mixed together in a competition growth assay. Therefore, one should be cautious when drawing conclusions from growth rate data.

Minor:

Line 143: redundant statement “Indeed, to our knowledge, DIA-NN is, to the best of our knowledge...”

Line 330: typo? “The proteasome was further the only enriched term...”

Lines 808 – 809: Only uses ORFs with systematic names, so there is the potential that the effect is due to non-systematically named ORFs which we know are present in these strains.

Author Rebuttals to Initial Comments:

General remarks to all three reviewers and the editor

We thank the reviewers and the editor for taking the time to work on this manuscript, and for the detailed and constructive feedback that we received. We have addressed all individual reviewer comments point-by-point in the rebuttal below. We preface this response with a summary of three major points addressed in the revision.

I. Impact of the large-scale proteomic technology and the ability of the species-wide approach to reveal generalizable insights that are not possible with studies of laboratory isolates

- All three reviewers were experts in the field of aneuploidy and provided detailed feedback that helped improve the manuscript in several key areas: addressing open questions specific to the aneuploidy field; enhancing the clarity of the message; and improving the language and presentation. We have addressed these issues as detailed in the point-by-point responses.
- Our study uniquely highlights the potential of a proteome-centric, multi-omic approach to natural isolate libraries, thereby expanding the scope of this research beyond aneuploidy alone. Recognizing that the novelty and significance of this aspect may have been insufficiently appreciated in the review process, we now better underline the value of moving beyond classic laboratory strains towards natural strain libraries for addressing conflicting laboratory findings, improving reproducibility, and discovering unknown protein functions. We also note that there are many technical and scientific obstacles to obtaining and analyzing high-quality, precise large-scale proteomes, especially when doing so in the context of the genetic and physiological heterogeneity of many hundreds of isolates across a diverse microbial species. We think that both these achievements, and the resource character of our study, are of broad general interest.

II. Difference in dosage compensation between natural isolates and previous studies using lab-generated aneuploids strains

The nature of dosage compensation in lab-generated, synthetic aneuploids has remained a highly debated subject. For instance, while synthetic aneuploids of the disome collection (Torres et al. 2007; Dephoure et al. 2014) attenuate subsets of proteins, in particular surplus subunits of macromolecular complexes, other lab strain models did not show such attenuation (Pavelka et al. 2010). In the search for an explanation as to why the lab strain collections differ in dosage compensation, we obtain a third answer in analyzing the natural strains: Our manuscript highlights significant differences between natural aneuploids and lab-generated aneuploids, necessitating a reassessment of the mechanisms underlying dosage compensation.

We admit that our initial cautious writing may have obscured key points; thus, we now more clearly distinguish between natural and lab-engineered aneuploids, and the mechanistic implications that derive from this distinction.

- While both synthetic disomes and natural aneuploids attenuate specific proteins, i.e., protein complex members (see Reviewer #1, points #0.2 and #6 as well as Reviewer #3, point #1; see also the newly added Fig. 2 of the manuscript), they differ substantially in the extent of attenuation

respective to the chromosome-wide effects. We apologize that the differences between studying
attenuation on a gene-by-gene level and on the chromosome-wide level created some confusion
around the novelty of our findings. Indeed, attenuation of the levels of protein complex components
was reported by Dephoure et al. and others (Brennan et al. 2019; Stingle et al. 2012; Dephoure
et al. 2014), and results in a bimodal distribution of dosage compensated and non-dosage
compensated proteins in lab-generated aneuploids. We confirm these original observations by
generating new proteomic data for the disomic strains as part of this study. Critically, both the
original data from Dephoure et al. and our data from these disomic strains show that while some
genes are attenuated, many attenuations have a moderate effect size, others are upregulated, and
many are unchanged, so that the average gene encoded on a duplicated chromosome is still
expressed 2-fold higher at both mRNA and protein level. Thus, “chromosome-wide” dosage
compensation is not significant in these synthetic aneuploids. By contrast, dosage compensation
affects ~70% of the proteins encoded on aneuploid chromosomes on natural isolates. In these
strains, on average, 25% of the relative gene dosage added by gained chromosomes is removed
at the proteome level. Individual isolates vary greatly, with some showing chromosome-wide
attenuation levels of 40% and more. Thus, there is extensive chromosome-wide dosage
compensation in natural isolates. This striking difference between aneuploid lab strains and
aneuploid natural strains is important to consider and suggests that long-term selection in natural
environments can select for the ability to tolerate aneuploidy. Further, this situation demands for
new mechanistic explanations (See point III) as dosage compensation involves many proteins that
are not known members of macromolecular complexes.

- - Natural aneuploids show few, if any, signs of proteotoxic stress, which has been considered a
source of the fitness costs in lab-generated aneuploids.
- - An important new aspect of our revision is that we broadened and conducted an in depth analysis
of the gene expression responses on euploid chromosomes in aneuploid strains (‘trans
signatures’). Laboratory-generated aneuploids show transcriptional signatures of the
environmental stress response (ESR) (Torres et al. 2007), the aneuploidy-associated protein
signature (APS) (Dephoure et al. 2014), and the common aneuploidy gene expression (CAGE)
signature (H. J. Tsai et al. 2019). These signatures were fiercely debated in the literature, as they
could indicate a common response to aneuploidy. However, our results show that transcriptional
signatures common to aneuploidy in laboratory strains are attenuated at the protein level in natural
isolates, and thus unlikely adaptive.
- - Natural strains, but not the lab-generated disomes, show an induction of structural components of
the ubiquitin proteasome system. In the revision, we expand this direction, as it provides
mechanistic insights into the mechanisms of dosage compensation acting on the genome-wide
level (see general comment #3).

**III. Importance of addressing mechanisms mediating dosage compensation in natural isolates**

The reviewers appreciated that this study helps to resolve disagreements between highly debated studies
in the field. They also encouraged us to leverage this dataset to contribute to a more detailed mechanistic
understanding of chromosome-wide dosage compensation.

In contrast to the attenuation of surplus protein complex subunits, chromosome-wide dosage
compensation is not understood mechanistically. As aforementioned, despite general buffering of gene
expression changes acting on euploid chromosomes *in trans*, structural components of the proteasome
are increased across natural aneuploids - and not in laboratory generated disomes. Interestingly, a deeper
analysis of our ubiquitinome data did show an increased ubiquitination of proteins encoded on aneuploid
chromosomes but demonstrated that they are not ubiquitinated at super-stoichiometric levels. This result
argued against specific degradation of proteins encoded on aneuploid chromosomes, presenting a
substantial mechanistic novelty compared to the dosage compensation of protein complex subunits that
are degraded when in excess (McShane et al. 2016).

From the reviewer comments, and from publicly available reviewer comments on previous papers in the
field (Dephoure et al. 2014), we realized that protein turnover data could provide crucial insights into a
mechanistic understanding of dosage compensation. However, no large-scale *S. cerevisiae* protein
turnover data was available, and certainly there was no data on protein turnover in natural isolates. To
105 address this gap and to reach deeper mechanistic understandings, we developed a strategy for measuring
protein turnover across the natural isolates in collaboration with Matthias Selbach's lab, experts in studying
protein turnover. We now present experiments measuring protein turnover in 55 natural isolates.
Importantly, we discovered that the total protein turnover rate correlates with the degree of chromosome-
wide dosage compensation.

This revised paper thus represents the first proteome-centric multi-omic study of a diverse collection of
natural yeast isolates. It includes not only matched genomes, mRNA and protein abundance data for ~800
genetically diverse natural isolate strains, but also ubiquitinomics and protein turnover data for subsets of
the strains. Next to the study of aneuploidy, our work has hence resource character. It provides the
115 community with publicly available data sets of natural isolates, allowing for both testing the generalizability
of lab-strain findings, and for identification of novel mechanisms.

Referee #1 (Remarks to the Author):

0.1 The review of manuscript “The natural diversity of the yeast proteome reveals chromosome-wide dosage compensation in aneuploids” by Muenzner et al. The presented manuscript focuses on understanding the consequences of abnormal chromosomal numbers in naturally occurring yeast aneuploid strains. The author team performs an impressive tour-de-force analyzing the transcriptomes and proteomes of several hundreds of aneuploid natural strains. The elegant evaluation and visualization make the data accessible, which is indeed a difficult task. The authors demonstrate that the effects of aneuploidy on cis-gene expression is largely dosage compensated on proteome level, while the transcriptome generally scales with the chromosome copy numbers. By elegant exploitation of proteomic analysis of ubiquitinated lysine residues they show that there is more ubiquitinated peptides in four natural aneuploid isolates and these are enriched for proteins encoded on the aneuploid chromosomes. This is further complemented by findings that aneuploid yeast degrade model proteasome substrate faster than euploid strains. This is an interesting and comprehensive study.

We thank the reviewer for this positive summary of our work.

0.2 However, there are several points, which need to be clarified and which bring some interpretations into questions. Also, the way how the data is interpreted and the entire tone of the manuscript is somewhat troublesome. The authors repeatedly state that this is the largest data set, and the first time that dosage compensation has been demonstrated, that they solve an ongoing conundrum, and that datasets from other species were too small.

We thank the reviewer for this critical comment and apologize if our claims were stated in a way that led to misunderstandings. We address the listed concerns below.

“this is the largest dataset...”

To the best of our knowledge, this study is the first systematic multi-omic study of a large number of *natural* isolates of any species that spans the genome, transcriptome, proteome, ubiquitinome, and in this revision, also protein turnover. In general, wild species have barely been characterized at the proteome level. The studies closest to this investigation are: 1) Mann lab (Müller et al. 2020): analysis of proteomes across 100 species (does not compare isolates within one species, does not address aneuploidy, and does not integrate proteomic with genomic and transcriptomic data) (Müller et al. 2020); 2) Hafen group: analysis of genomes, transcriptomes and proteomes in 30 lines of the DGRP collection for factors influencing *Drosophila* wing size (Okada et al. 2019, 2016); and 3) Laukens group: study of aneuploidy in six cloned *Leishmania* strains across multiple -omics levels (Cuypers et al. 2022). In yeast, there is a study from the Gasch lab that analyzes a small number of aneuploid natural isolates, but only at the transcriptome and not at the proteome level (Hose et al. 2015). To record proteomes at scale, and to make them quantitatively comparable over a broad range of genetically and physiologically diverse isolates, we solved considerable technical obstacles, e.g., on the data processing side (Materials and Methods). Overall, this is a high quality, quantitative proteome dataset for large numbers of natural isolates.

*“... the first time dosage compensation has been demonstrated...”*

We never intended to claim to be the first to report dosage compensation at the proteome level, and apologize if this was not clear. Rather, our intention was to solve discrepancies in the literature, and once we found that natural isolates show chromosome-wide dosage compensation that is lacking in the lab strains, we propose new mechanisms. We cited previous studies and not only re-analyzed existing data but also generated new proteomic data for the laboratory-generated synthetic aneuploids (synthetic disomic strains from Torres et al., 2007 (Torres et al. 2007)). We take the reviewer's concern seriously, and thus reworked the text extensively for clarity.

It appears that we may have failed to clearly distinguish between gene-level (i.e. the attenuation of specific proteins or groups of proteins) and chromosome-wide dosage compensation, an issue that has been problematic in prior studies as well (Hose et al. 2015; Torres, Springer, and Amon 2016; Gasch et al. 2016). The new measurements generally confirm the original results and the conclusions (Dephoure et al. 2014) that synthetic disomic aneuploids do attenuate specific individual proteins, mostly protein complex subunits. An important distinction is that both the original and the new data reveal that this protein-level dosage compensation does not extend to significant effect sizes once considering the chromosome-wide level; the average protein encoded by a gene on a duplicated chromosome is also expressed close to 2-fold higher on both the transcript and the protein level, in the disomes (Fig. 2, 3).

This contrasts with the herein studied natural isolates, where proteomic dosage compensation extends across the entire chromosome. The effect is strong and significant (Fig. 3b, SI Table 13): across the natural aneuploids, 70% of the genes encoded on aneuploid chromosomes show dosage compensation (Fig. 2c, Fig. S5), and the effect sizes manifest on the chromosome-wide average: about 25% of the relative gene dosage due to aneuploidy is attenuated across the ~100 natural aneuploids (Fig. 3c), with individual strains deviating in part significantly from the average, showing 40% average attenuation and more (Fig. 3e). We now highlight this distinction more explicitly (e.g., lines 290ff., lines 379ff. and lines 421ff). Moreover, we have changed the structure of the manuscript, which we hope will make it easier for the reader to follow our argument. Before we dive into the chromosome-wide signals that are absent in the disomes and prevalent in the natural strains (Fig. 3), we now include a "gene-level" analysis (Fig. 2) that expands on previous studies whilst harnessing the size of our dataset. This analysis corroborates that *individual* proteins are attenuated in both synthetic and natural strains, but also demonstrates that gene-level attenuation is much broader and has a stronger effect size in the natural isolates than in the lab strains. We hope that the addition of the new results, together with the extensive re-writing and restructuring of our paper, help to clarify our claims.

“ ... they solve an ongoing conundrum... ”

To highlight the important distinction between attenuation of specific genes and chromosome-wide attenuation, we would like to refer to a debate that arose from the study "Dosage compensation can buffer copy-number variation in wild yeast" (Hose et al. 2015). In this paper, the Gasch lab recorded transcriptomes for six wild isolates and performed gene-level analyses, reporting that certain transcripts expressed from aneuploid chromosomes can be attenuated in wild isolates ("dosage compensation [at the mRNA level] is likely an inherent trait in *S. cerevisiae* that functions at a subset of yeast genes" (Hose et al. 2015)). This kind of mRNA-level dosage compensation especially applies to genes that are toxic when present in high copy numbers (Hose et al. 2015). However, as the subsequent reply paper by the Amon lab ("No current evidence for widespread dosage compensation in *S. cerevisiae*" (Torres, Springer, and Amon 2016)) and the reply-to-the-reply paper by the Gasch lab ("Further Support for Aneuploidy Tolerance in Wild Yeast and Effects of Dosage Compensation on Gene Copy-Number Evolution" (Gasch et al. 2016))

showed, there is no *chromosome-wide* dosage compensation at the transcriptome level occurring in natural yeast isolates.

The discrimination between chromosome-wide dosage compensation and the attenuation of specific mRNAs (Hose et al. 2015; Gasch et al. 2016) or proteins (Dephoure et al. 2014) is important because the mechanisms behind these modes of buffering require different explanations. In this revision, we include more mechanistic investigations, and associate chromosome-wide dosage compensation to a general increase in protein turnover.

Another important, yet essentially unresolved debate concerned diverging results between different collections of lab-generated aneuploids when it comes to the attenuation of protein complexes (Pavelka et al. 2010; Berman 2010; Torres et al. 2007), and another debate concerned the nature and importance of *trans* signatures (H. J. Tsai et al. 2019; Terhorst et al. 2020; H.-J. Tsai et al. 2020). The study of the natural isolates provides data that sheds a fresh light and largely resolves all three debates. Dosage compensation indeed can be extensive (in natural strains), protein complex members are attenuated in aneuploids – but are only one class of attenuated proteins, and many *trans signatures* are not unidirectional at the transcriptome plus are removed by the proteome in natural strains.

... that datasets from other species were too small.

We would like to emphasize that we are referring specifically to *natural isolate* libraries, and apologize if this was not sufficiently clear. It is correct that in other settings (e.g. cancer cells, human plasma, yeast libraries), we and others have created large proteomic datasets in the past (e.g. the largest mass spectrometry proteome of a model species is perhaps our recent survey of the yeast deletion collection which also revealed many aneuploidies (Messner, Demichev, Muenzner, et al. 2023)), but natural isolate libraries unleash a significant new potential that had thus far not been realized at the proteome level. We have reworded our manuscript extensively to make our objectives using natural isolates clear. In parallel, we have now worked on a much better incorporation of proteomic studies from other species addressing aneuploidy, including cancer cells (please see comments below).

*0.3 They propose in the discussion that possible further studies may reveal that “chromosome-wide dosage compensation could be a mechanism conserved across species”. The problem here is that the authors ignore a significant number of publications on this topic that clearly show that human aneuploidy is dosage compensated on proteome level in engineered aneuploids as well as in cancer samples (see below).*

We apologize if our text was confusing, we had aimed to discuss here specifically chromosome-wide dosage compensation, where the mechanisms are largely unclear, perhaps without making a succinct distinction to gene-specific dosage compensation (i.e. of protein subunit complexes). We have now expanded the literature corpus referenced (see also reply to Reviewer #1, point #2), and hope that this confusion is well mitigated in our revision.

*0.4 Additionally, the authors could have done more analysis with their rich data, which would bring further insight into the observed phenomenon.*

This is a very valuable point. We agree that the dataset is very rich and will likely be useful for addressing a broad diversity of questions, including issues concerning aneuploidy. We significantly expanded the revision by including, e.g., i) a detailed description of protein-level attenuation across protein groups, including an investigation of protein properties predictive of increased attenuation propensity, and ii) a quantitative strain-by-strain breakdown of attenuation and its association with isolate characteristics. The multi-omic study now also includes the first dataset of protein turnover in natural isolates.

Below are the major aspects which should be addressed.

1.1 Throughout the manuscript, the authors present the previous literature as if the dosage compensation on proteome level has never been demonstrated.

We apologize, as aforementioned, this certainly was not our intention! Rather, we engaged extensively with previous work, and aimed to bring more clarity into a controversial field where some studies in (lab-generated) aneuploids showed gene-by-gene level dosage compensation (Hose et al. 2015; Dephoure et al. 2014) while others did not (Pavelka et al. 2010; Torres, Springer, and Amon 2016). We were surprised to get a third answer; that the dosage compensation is not only manifesting at gene-by-gene level, but extends to the chromosome-wide level in natural isolates. We hope that the aforementioned reorganization and extensive revision of the manuscript, including the distinction between “gene-level” analysis (Fig. 2) and chromosome-wide attenuation (Fig. 3), and the comparison of gene-by-gene dosage compensation in natural isolates vs. lab strains helps to clearly illustrate the important differences between the natural aneuploids and lab-engineered disomic strains.

1.2 This is likely originating from the confusing and to a certain extent inconclusive previous works on aneuploidy in budding yeast. It is indeed an honorable task to bring more light into this difficult question that has been obscured by disagreements of several publications on this topic.

We thank the reviewer for raising this important topic. Indeed, the key motivation for expanding the repertoire of molecular biology (and proteomics specifically) with natural isolate libraries is to test the universality of potentially conflicting findings, and to address the reproducibility crisis. This dataset helps to shed light on divergent and controversial results obtained with yeast laboratory models which are not necessarily restricted to the study of aneuploidy; it might solve disagreeing results by looking beyond laboratory models and cell lines, thus potentially helping to make results more generalizable. The resource character of the study was also our motivation to be very strict in data quality and filtering (see also our response to the reviewer’s technical comment #1 on pages 17-18 in this reply). To make this point more visible, we have expanded the introduction with a new introductory paragraph to highlight this aim better.

1.3 However, it should be noted that the laboratory of Angelika Amon/Eduardo Torres repeatedly showed for the very same set of disomic lab strains that there is dosage compensation on proteome level, but not on transcriptome level. This is suggested first by Torres et al (2007, Science, 2010, Cell) and clearly stated in Dephoure et al (2014, Elife, see figures 2B, 3B).

Indeed, we engaged extensively with these studies (Torres et al. 2010, 2007; Dephoure et al. 2014) not only by referring to them in the manuscript, but also by i) including a re-analysis of the transcriptome data presented by Torres et al. 2007 (Torres et al. 2007) as part of Fig. 2 and Fig. 3; and by ii) generating a

310 new proteomic dataset for the disomic strain collection using state-of-the-art proteomic technology in order
to exclude that any differences in conclusions are of technical nature (as detailed below). Importantly, we
agree with their conclusions about synthetic strains, but we reach different conclusions for the natural
isolates as described above (general point II on pages 1-2 of this rebuttal letter) (and repeated here).

*1.4 Other laboratories did not reach the same conclusion, which is indeed interesting. The discrepancy
among other publications might be for example due to different pre-processing/normalization procedures,
and it would be important for the field if the authors would clearly identify the source of confusion on the
topic of dosage compensation in budding yeast.*

We heartedly agree about the importance of careful data normalization and processing when comparing
proteomic data. In particular, disagreements between manuscripts from the Amon, Gasch and Li labs were
based upon different data analysis strategies (Hose et al. 2015; Gasch et al. 2016; Torres, Springer, and
Amon 2016; Berman 2010; Pavelka et al. 2010; Torres et al. 2007). In order to fully rule out different
technical differences explaining diverging results, we obtained the disomic strain collection originally
described by the Amon lab (Torres et al. 2007) and generated a new dataset using the Ralser group's
proteomic platform and data normalization pipelines (SI Table 7). Both the old and new data reach largely
the same conclusions: the disomes attenuate specific proteins; among them, protein complex subunits are
enriched, and on average, no extensive chromosome-wide attenuation is evident. Thus, the differences
between disomes and natural isolates cannot be explained by the different technologies for the preparation
or normalization of the proteomic data. We now explain the normalization strategies more clearly in the
text (lines 266ff., lines 1143f.) and figures (Fig. S16, S4).

*2. The authors focus on yeast strains. However, it is incorrect to claim that the dosage compensation could
not be clearly shown in other species. Paper by Stingele et al (MSB, 2012) shows dosage compensation
on proteome level (enriched, among others, for subunits of macromolecular complexes) in six different
engineered human cell lines with extra chromosomes. Donnelly et al (EMBO J, 2014) shows that the
proteasome activity is increased in human aneuploid cells, and McShane et al (Cell, 2017) shows that
proteasomal degradation is responsible for a significant fraction of dosage compensation in aneuploid
human cells. Dosage compensation on protein levels was also recently demonstrated in human
monosomic cell lines by Chunduri et al (Nat Comms 2021) and analysis of transcriptome and proteome
from 375 cancer cell lines confirmed genome-wide dosage compensation on proteome level (). There are
several other papers showing various aspects of dosage compensation on protein level in cancer cell lines.
None of these papers is cited. From all the work on human cell lines, only the paper by Liu et al focusing
on trisomy 21 is mentioned (Nat Comms, 2017). The authors of course can choose to focus on yeast
strains, but then they should not make these generalized claims.*

We appreciate the reviewer's insightful comment, and as aforementioned, we apologize if our writing
implied we would claim to see dosage compensation for the first time, which was not our intention. We
acknowledge that our initial manuscript may have overly concentrated on chromosome-wide dosage
compensation, potentially under-emphasizing studies addressing aneuploidies and dosage compensation,
including at the gene-specific level, in species other than yeast. However, we would like to emphasize that
while there are mechanistic explanations for attenuation of specific proteins at the gene-by-gene level,
such as for the non-exponential degradation of surplus protein complex members (McShane et al. 2016),
comprehensive mechanistic explanations for chromosome-wide dosage compensation remain elusive.
The distinct difference between lab-engineered disomes and natural isolates provides a unique platform

to explore these mechanisms. Addressing the reviewer's points, we have expanded our manuscript to include references to relevant studies in mammalian and cancer cell lines. We also discuss potential applications of these yeast findings towards uncovering mechanisms mediating chromosome-wide dosage compensation in these models. We hope this revision provides a more balanced and comprehensive view of the subject matter.

3. A key issue with analysis of data from aneuploid strains is the normalization. The authors normalize the data to an "ideal euploid", which is a median of all euploid strains regardless their ploidy. This requires the assumption that proteome scales linearly with ploidy, which has never been demonstrated. In contrary, there is evidence that with increasing ploidy the proteome is sub-scaling (Gemble et al, Nature 2022; <https://www.biorxiv.org/content/10.1101/2021.05.06.442919v2>). Thus, the authors should normalize haploid aneuploids to haploid euploids, diploid to diploid euploids etc. Did they employ the normalization to the median euploid wild type for all their analysis? Did they perform forced normalization to 0? Was the forced normalization performed including the data from aneuploid chromosomes, or excluding them, or euploid and aneuploid chromosomes separately?

We appreciate the comment of the reviewer. Because there were no comprehensive proteomic studies addressing natural isolates in the literature, there were no "off the shelf" solutions for comparing peptide abundance data between very different isolates given their genetic diversity. Hence, developing applicable normalization strategies both for the proteomic raw data processing, and for analyzing the aneuploid versus the euploid strains, was a key and critical aspect of the work and is detailed in the results sections and the Materials and Methods part of the revised manuscript (please see "Proteomics data processing" and "Post-processing statistical analyses").

In order to address the reviewer's specific point, we have tested additional normalization strategies to compare between euploid and aneuploid proteomes in natural strains. The normalization strategy used in most parts of this study (comparison of the abundance of a certain mRNA or protein to its median abundance across all euploid strains) is now compared to the strategy suggested by the reviewer (comparison of mRNA or protein abundance of a haploid strain to the median abundance of that same mRNA or protein across all euploid haploid strains, diploid to all euploid diploids, triploid to all euploid triploids, etc.). The results have been included in Fig. S4. In both normalization strategies, the determined \log_2 expression ratios were further normalized strain-wise to ensure expression ratios of genes located on euploid chromosomes in both the euploid and the aneuploid strains centered around 0. The normalization factors were calculated using data of euploid chromosomes only, but subsequently applied to all chromosomes, aneuploid and euploid, of each strain (see also Materials & Methods). Of note, a similar normalization approach also was employed in Pavelka et al. 2010 (Pavelka et al. 2010). Further, when assessing the aneuploidy signal over the span of a growth curve, we used strain-by-strain comparisons at each time point to determine relative mRNA and protein abundances (see also Reviewer #2, point #8).

Median chromosomal expression levels of both normalization procedures – normalization across all euploid strains vs. normalization per ploidy – show consistently high correlation across ploidies (Fig. S4a, b), with Pearson correlation coefficients ranging between 0.92/0.91 (median chromosomal mRNA/protein expression correlation between the two normalization procedures for tetraploid strains) up to 1/0.99 (median chromosomal mRNA/protein expression correlation between the two normalization procedures for diploid strains). Therefore, the reviewer is correct that proteomes do not necessarily scale linearly with ploidy (Yahya et al. 2022). However, as the correlation analysis shows, this effect does not have a major influence on the relative expression picture obtained. We included these results in the manuscript.

4. The dosage compensation might be different in cells with higher ploidy than in haploids – one can imagine that 4N+1 results in different response than 1N+1 (this is also suggested by data from Dephoure et al, eLife 2014). This is also suggested here in the plot 3a – the dosage compensation in 3N+1 seems smaller than in other datasets. The authors need to rigorously test this.

We thank the reviewer for asking about the impact of basal ploidy, which addresses the differences between absolute vs. relative dosage compensation. The proteomes of this set of natural yeast strains, spanning multiple basal ploidies from 1n to 5n, are indeed an ideal model to test the relative contributions of ploidy on dosage compensation upon a chromosomal gain. As the reviewer correctly notes, Fig. 3b (part of previous Fig. 3a) illustrates that cells of higher ploidy show less *absolute* dosage compensation. This is consistent with the expectation that a lower relative gain in gene dosage will require less dosage compensation. For example, adding one chromosome to a pentaploid strain represents a much lower relative impact (where the relative gene dosage increases by 20%) compared to adding one chromosome to a haploid strain (with the relative gene dosage increasing by 100%).

In order to address the reviewer's question, we have elaborated this point more explicitly. In the previous manuscript version, the absolute extent of attenuation was visualized by comparing the shifts of the protein or the mRNA distribution to the expected relative expression value if no attenuation occurred (now Fig. 3a, b, former Fig. 3a). We now included an additional table in the SI material, SI Table 13, which statistically quantifies the extent of attenuation (compared to the expected median of relative expression based on the relative copy number of each chromosome) at both the mRNA and the protein level, for all relative chromosome copy number changes present in this set of natural aneuploid isolates.

The reviewer is correct that, on average, the proteins encoded on singly gained aneuploid chromosome(s) in triploid strains are slightly less attenuated (Fig. 3b, SI Table 13) than what would be expected. A caveat here is that this karyotype is rare in the natural strain collection so that the average values are based on a low number of strains. The dataset contained only four aneuploid triploid strains (strains ABV, AEE, APN, and BMG). Of these four strains, strain AEE showed significant attenuation (~ 43% relative attenuation, $p = 4.2 \cdot 10^{-08}$ after multiple comparisons adjustment) of the protein abundance of genes expressed on its 3n+1 chromosomes (chr. 5, 8, 9), whereas strains ABV, APN, and BMG were not significantly attenuated (SI Table 14, see also the remarks to point #5 of Reviewer #1 - strain-wise analysis of attenuation). Statistical assessment whether triploid strains are consistently less attenuated than expected by the regression across ploidies and relative copy number gains (Fig. 3c, g) would require extended sampling of natural yeast strains to discover and subsequently include more triploid aneuploid isolates. Because of the limited statistical power of this type of analysis with higher ploidy strains, we chose not to highlight it in this manuscript (besides showing the data in Fig. 3c, g, and SI Tables 13, 14).

5. The dataset allows the authors to address some further important questions. For example, could it be that the increased proteasomal degradation and increased dosage compensation correlates with the degree of aneuploidy (=number of imbalanced chromosomes), or type of aneuploidy (gains vs. loss)?

We thank the reviewer for encouraging us to expand our analysis. We now include i) data concerning the strain-wise extent of attenuation and its relationship to isolate characteristics (Fig. 3, SI Table 14, Fig. S11), including the degree of aneuploidy, and ii) data addressing the impact of chromosome losses (Fig. S9b, SI Table 13).

Specifically, we investigated whether the extent of attenuation observed per strain was correlated with the degree of aneuploidy. The degree of aneuploidy has previously been measured as the number of genes located on aneuploid chromosomes (e.g., Schukken and Sheltzer, 2022 (Schukken and Sheltzer 2022)). Since we observed extensive attenuation of the protein load, we defined an additional measure of degree of aneuploidy by calculating the ploidy-adjusted absolute number of protein copies per cell of all proteins encoded on aneuploid chromosomes in a given strain (referred to as “aneuploid protein load” in the manuscript). Of note, the two measures are highly correlated (PCC = 0.96, $p < 0.05$) (Fig. S11b). We found that the degree of aneuploidy correlated well (PCC = 0.441) with the extent of attenuation per strain (Fig. S11c). Furthermore, when regressing the aneuploid protein load against the median extent of attenuation for all strains with the same degree of aneuploidy, we found a good correlation (Fig. 3f). This indicates that the heavier the aneuploid protein load, the more a strain is attenuating this load, despite variability of the extent of attenuation between strains of the same karyotype (Fig. 3h). These observations go hand in hand with the observation that large chromosomes – i.e., chromosomes that will lead to a higher aneuploid protein load – are less often aneuploid in natural isolates (Scopel et al. 2021), presumably due to the higher demand that attenuation of the increased protein load puts on the aneuploid cell.

We observed chromosome losses in 11 out of 95 natural aneuploid isolates (see SI Table 1 and Fig. S2). Due to the low number of such cases, it is hard to provide a firm conclusion of whether losses are better or worse compensated than chromosome gains, but we do see that aneuploids that lost chromosomes are also buffered at the proteome level. In the revised manuscript, we now provide quantitative information about dosage compensation in isolates with chromosome losses (SI Table 13 and SI Table 14). For details on this, please also see our specific response to Reviewer #1, point #7.

6. Comparison of engineered and naturally occurring disomic strains revealed increased dosage compensation on protein level in naturally occurring strains. What are the additional proteins that become dosage compensated? In engineered strains typically subunits of macromolecular complexes are dosage compensated. Is there any subgroup of proteins enriched among the dosage compensated proteins in naturally occurring aneuploids?

We thank the reviewer for bringing up this important question, and we apologize for not having addressed it extensively in our first version. We have included a new Fig. 2 that addresses specifically attenuation at the level of individual genes. Natural strains attenuate ~70% of proteins encoded on aneuploid chromosomes (Fig. 2c, Fig. S5), thus spanning over almost all functional categories. Exceptions are proteins that are part of a set of growth-associated metabolic pathways that are not attenuated (Fig. S7).

Further, we investigated how intrinsic protein features and factors that are related to the synthesis or function of a protein (other than complex membership) can contribute to protein attenuation (Fig. 2f, Fig. S8a, b). We also analyzed whether proteins associated with specific cellular pathways or processes, that have certain molecular functions, or that are localized at certain cellular compartments are especially prone to protein- or mRNA-level attenuation (Fig. 2g, Fig. S7, SI Table 12). We find that being a subunit of macromolecular complexes remains the strongest factor in explaining attenuation, but equally, that MMC membership falls short of explaining the extent of dosage compensation as observed in the natural aneuploids.

*7.1 In human monosomic cells as well as in Drosophila lacking substantial parts of a chromosome, some dosage compensation has been observed also on a transcriptome level. Is this the case also in the naturally occurring aneuploid yeast lacking chromosomes?*

The reviewer raises an important point in asking specifically about isolates that incur chromosome losses. These are much more rare than chromosome gains in the natural strains, with only 11 of the 95 aneuploid isolates having a reduction in chromosome numbers (SI Table 1). Of these 11 strains, five were not significantly attenuated at either mRNA or protein level (isolates AVC, AVP, BDE, BRP, and CRA), while four were significantly attenuated at the protein level, but not at the mRNA level (isolates BFM, BIG, BML, and CML) (SI Table 14). Hence, we conclude that in the natural strains, chromosome losses are also predominantly buffered at the proteome level.

*7.2 Related to this - in Fig. 2a, there are very clearly some strains which show an increased genome signal, but reduced mRNA signal (e.g. the first aneuploid on the top, chromosome 1, generally chromosome 1 in most strains). Yet, the global quantification shows no dosage compensation on mRNA level. How comes?*

We agree that some isolates show a reduced mRNA signal. To better account for these isolates, we have expanded the analyses to provide a “*per-strain*” quantification of attenuation on both the mRNA and the protein levels (Fig. 3e-h and SI Table 14). This analysis revealed that three out of the 95 aneuploid isolates (BMQ, AQR, and BTS) had significant chromosome-wide attenuation at the mRNA level (SI Table 14). The proteomes of three other isolates (ARS, CKS, CMC) resemble those of euploid strains, strongly suggesting that the aneuploidy had been lost from the strains on which we measured the proteomes (SI Table 14, Materials and Methods). Because of the small number (3) of isolates exhibiting mRNA-level attenuation, we could not further address this issue with the current dataset. We certainly agree that the mechanisms underlying this mode of attenuation could be highly interesting, and we mention this now in the discussion (lines 623f.).

*8. Figure 1 is not informative. Panels a and b are well described in the text and could be in supplementary (or left out). It is not clear what is the purpose of panel c – it does not illustrate anything relevant for the manuscript. Panels d and e would be useful, but are too small. In d, it would be better to express the fraction of aneuploids in % (and state the total number of strains in each category). Panel e – is the distribution of aneuploid chromosomes the same for all different ploidy levels?*

Thank you for these suggestions. We made the necessary modifications to panels a and b to effectively highlight our proteomics pipeline and the integration of the -omics datasets. We omitted former panel c from the manuscript. Instead of the former panel e, we incorporated a new panel (Fig. 1d) that illustrates the distribution of aneuploidies, differentiating between single-chromosome and complex cases in the integrated data. Additionally, we have included a supplementary figure (Fig. S2b) depicting the distribution of aneuploidies across the chromosomes categorized by ploidy. Due to a lower number of isolates with higher ploidies, it is challenging to fully address the reviewer's inquiry regarding the distribution of aneuploid chromosomes. Nevertheless, it is worth noting that smaller chromosomes (Chr I, III, and IX) do exhibit a higher occurrence in haploid, diploid, and triploid isolates, as has been noted previously in Peter et al. and Scopel et al. (Peter et al. 2018; Scopel et al. 2021).

9. *The K—GG-modified peptides should be compared with the dataset from McShane et al (Cell, 2017) to test whether similar categories of proteins are degraded by proteasome in aneuploid yeast strains as in aneuploid human cell lines.*

We thank the reviewer for suggesting a comparison of our results regarding protein attenuation with the data obtained by McShane et al. (McShane et al. 2016). Such comparison has of course to be done with great caution, because we deal with different species and different contexts – the McShane dataset was recorded in a human cell line, while our dataset is recorded in natural yeast isolates. Nevertheless, we made every effort to address the reviewer's question, which led to interesting results. To do such a
comparison between human and yeast data, one cannot work at the peptide level as there are only very few peptides with identical sequences that belong to the same protein shared between the distant species. However, we could work at the protein level by comparing the attenuation between protein orthogroups. We thus identified the yeast homologs of the human proteins quantified and assigned as either exponentially degraded (ED), non-exponentially degraded (NED) or undefined by McShane et al.
(McShane et al. 2016): We identified 759 yeast homologs for 3187 human proteins covered in the McShane study (~ 23%), agreeing very well with the expected fraction of the human proteome that has yeast homologs. Of those 759 proteins, 146 were classified NED, 349 classified as ED, and 262 undefined; two proteins could not be unambiguously mapped to one of these categories.

For a total 246/757 proteins, we had data regarding their attenuation in natural aneuploid isolates and compared the results from McShane et al. with the data obtained in the gene-by-gene attenuation regression analysis of the revised manuscript (see also reply to Reviewer #1, point #6, as well as Fig. 2). We found that yeast proteins predicted to be NED showed the strongest attenuation (attenuation regression slope closest to 0), whilst ED and “undefined” proteins were still significantly attenuated, but to a lesser degree (Fig. S8c). Furthermore, neither of the three categories showed notable attenuation at the
mRNA level. Thus, this analysis indicated that the probability of a protein to be non-exponentially degraded, is, at least partially, conserved between human and yeast. The most likely explanation is the high conservation of large protein complexes. We have added this information to the main manuscript.

*10. Fig. 3c is a nice summary, but a fair comparison would be natural disomic strains vs laboratory disomic strains (unless the authors demonstrate that the dosage compensation is independent of basal ploidy).*

As suggested by the reviewer, we have added an SI figure that enables direct comparison of relative expression distributions between natural isolates carrying duplicated chromosomes and laboratory
disomes (Fig. S9a vs. Fig. 3a). Indeed, in Fig. 3c (also previously Fig. 3c), we quantify the *relative* mRNA and protein level buffering across chromosome copy number changes, which is around 25% across the ploidies included in this dataset (1n to 5n), and therefore, independent of basal ploidy. Please also refer to the answer to Reviewer #1, point #4, and SI Table 13. In addition, we performed a direct comparison
of the extent of dosage compensation for proteins shared between natural and lab-engineered strains with duplicated chromosomes (Fig. S6, lines 310ff. of the revised manuscript).

*11. Fig. 4a, b – the enrichments scores and log2 fold changes depicted are very small (<0.1 in 4b!). The scale on X and Y axes should be identical on the left and right plot in 4b to allow proper comparison. For fair comparison, only disomic natural strains should be compared with disomic engineered strains. In human aneuploid cells lines, the increased expression of proteasomeal subunits correlates with the amount of extra DNA (Donnelly et al, EMBO J 2014). Is it the case here as well?*

We thank the reviewer for asking about the effect sizes of proteasome level changes. “Proteasome” is the
top enriched term in the gene set enrichment analysis we describe. Indeed, the average effect sizes we
report are small in absolute terms, but highly significant, with some corrected p-values of differentially
expressed proteins being below 10^{-6} . In interpreting this result, one needs to keep in mind that we are
averaging the *trans* proteome response of close to one hundred genetically diverse natural aneuploid
isolates. As a consequence, the statistical power is excellent, but the effect sizes average out since all
strains have a varying fold-change in each protein. Lastly, it is actually remarkable that a signal indicating
upregulation of the proteasome could be detected across aneuploid isolates given the different genetic
backgrounds and their probable – not necessarily aneuploidy-related – adaptation to diverse ecological
niches. Keeping this in mind, we argue that the effect sizes for the top terms are around the scale one
could plausibly expect them.

In order to illustrate this point further, we have included a supplementary figure demonstrating the diversity
of the effect sizes in the revised manuscript (Fig S12a). In addition, given that the aneuploid isolates of the
integrated dataset are mostly diploid, we decided to test whether the proteasome upregulation we are
610 observing is driven exclusively by diploid isolates. We observed that this is not the case: both when
restricting the *trans* regulation analysis to diploid strains only, and when restricting the analysis to all strains
but diploid ones, upregulation of the proteasome can be observed (Fig S12d). We have added this to the
manuscript.

For the same reason of averaging across diverse strains, it would be misleading to directly compare the
fold-change values between a dataset that represents many highly genetically diverse natural
backgrounds, with any dataset that is generated in just one background (i.e., the disomes). We have
adjusted the previously differently scaled x- and y-axes as the reviewer suggested and included these re-
scaled figures in the supplement (Fig. S12e). However, this same scaling might be misleading as it can
suggest to the reader one could directly compare the fold-change values between the two very different
datasets, which one cannot. Therefore, we decided to keep the two differently scaled plots in the main
figure.

Lastly, we addressed the suggestion by the reviewer to check for any correlation between the expression
of proteasomal subunits (as identified by the KEGG term “Proteasome”) and the degree of aneuploidy
measured as the aneuploid protein load (compare also reply to Reviewer #1, point #5, and Fig. S11b, c).
We did not observe a statistically significant correlation (Fig. S12f).

*12. Discussion, first paragraph - the authors claim that their datasets is unique, advances proteome
technology, and such studies have not been previously possible due to the technical limitations of
proteomic platforms. This is a false statements. Comparable datasets have been obtained, e.g. the
transcriptomes and proteomes of hundreds of cancer cell lines from the CCLE. Their claim of highly
complete proteome for 796 strains cannot be validated, because the full data set is not provided.*

While we highly appreciate the reviewer’s comments, we have to disagree with the assertions they make
in this point and would like to reply in detail.

“the authors claim that their datasets is unique”

To the best of our knowledge, this dataset is the first multi-omic, proteomic-centric study addressing a large collection of wild isolates.

“advances proteome technology”

So far, almost all quantitative proteome studies in the literature, including our own studies, contain data from a single species. A notable exception is a paper from the Mann lab that records proteomes across very diverse species (Müller et al. 2020), but that study does not go into isolates of any same species; it does therefore neither aim at, nor captures the main focus of this study: the proteomic diversity within a species. The quantitative comparison of a large number of proteomes across genetically diverse natural isolates has, to our knowledge, not been done before, and adds extra challenges. Moreover, the proteomic platform that we present as part of this study combines several technical advances that we have recently introduced in order to improve scalability and quantitative robustness of proteomics. These developments include new and fast chromatography, new mass spectrometric acquisition schemes, and raw data processing software (Messner et al. 2020, 2021; Wang et al. 2022). This specific paper advanced further over our recent studies and introduces a semi-automated high-throughput platform for generating yeast proteomes at scale. We have used the experimental part of the platform also in a parallel study where we generated proteomes for a genome-wide collection of deletion strains (Messner, Demichev, Muenzner, et al. 2023), although the data analysis side there was much more straightforward and did not require the development of the pre-processing strategy for natural isolates. The success of that work provides orthogonal evidence for the unique potential to generate previously unachievable large-scale proteomic studies. To be able to provide large numbers of proteomes not only for lab strains, but also for natural isolates, we had to advance data processing pipelines to handle the inherent heterogeneity of the species, which represents another novel aspect of this study. This work required important methodological efforts to balance proteome coverage, sensitive peptide detection, and limiting false detection rates associated with the diverse genetic backgrounds and SNPs in the natural isolates, which required the evaluation of several strategies and settings including the comparison of experimental and in-silico libraries.

*“Comparable datasets have been obtained, e.g. the transcriptomes and proteomes of hundreds of cancer cell lines from the CCLE.”*

We highly appreciate that many excellent proteomic datasets exist in the literature, and indeed, the Pride database has grown to ~23,000 datasets as of to date, demonstrating the rapid growth in the proteomic field. Some of these datasets, like the cancer cell line proteomes of the CCLE collection mentioned by the reviewer, but also our own proteomic analysis on the yeast deletion collection (Messner, Demichev, Muenzner, et al. 2023), or the aforementioned study by Dephoure et al. in the disomes (Dephoure et al. 2014) can be linked to transcriptomic datasets and have been used to study dosage compensation in aneuploids. However, to the best of our knowledge, none of these datasets provide data for natural isolates and therefore empower the study of natural genetic diversity at the proteome level, specifically for finding new mechanisms that explain chromosome-wide dosage compensation, in contrast to non-exponential decay.

At this point, we would also like to emphasize that this study is a showcase of the recent advancements in high-throughput proteomics technology that were in part facilitated by our lab. Staying with the CCLE example pointed out by the reviewer, and whilst being fully appreciative of the massively beneficial resource this dataset represents for the research community, until recently, recording even moderately sized datasets was a truly gigantic effort. Achieving quantitative comparisons across sample series came with substantial drawbacks, including long acquisition times, missing values, batch effects, or

misidentifications due to selection if isobaric labels were used. For instance, our own first large proteomic
dataset encompassing all yeast kinase knock-out strains and published in 2018 still required six months
of measurement time (Zelezniak et al. 2018). The present dataset instead is generated in a semi-
automated pipeline that can be reproduced by others, offers a substantial gain in throughput, precision, is
695 much easier to scale, and cheaper to use. As a detailed discussion of the specifics of new proteomic
platform techniques would go beyond the scope of this study, we have summarized the recent advances
in high-throughput proteomics in a parallel review article and published it in an expert proteomic journal
(Messner, Demichev, Wang, et al. 2023), as well as expanded the discussion of this in the present
manuscript.

We would like to mention that ubiquitinomic and, with this revision, protein turnover data generated for
natural isolates are further unique aspects of our study.

*“Their claim of highly complete proteome for 796 strains cannot be validated, because the full data
set is not provided.”*

We have uploaded the raw data of the natural isolate proteome collection, as well as the raw data of the
disomic strain proteomes to MASSIVE, and will make them publicly available upon publication of our study.
In addition, we would politely like to point out that we included the full proteomic dataset for the 796 strains
in the initial submission as SI Table 6. Similarly, the proteomic data for the disomic strains had been
provided as SI Table 7. We now include a List of Tables in the manuscript, summarizing the content of the
provided SI tables.

*13. The discussion is largely overinterpreting the data and making sweeping generalized statements which
are not justified. Besides the already above mentioned examples the authors claim that “the increased
survival of aneuploid cells under stress conditions is often dependent on one or more key stress response
genes . . . “. For this they cite exactly one original papers. This does not sound like a general phenomenon.*

We apologize if the writing was too generalized. Our intention was to highlight the instances in which extra
copies of genes are sufficient to enable growth under stress conditions that limit the growth of euploid
progenitors. This has been seen in many cases, while the genes involved are sometimes less well
characterized. One of these examples is the study by Yona et al. (Yona et al. 2012) that we cited. However,
the upregulation of specific genes located on chromosomes that become aneuploid in order to deal with
stresses has indeed also been described in other circumstances. For example, in *C. albicans* and in *C.*
neoformans, Erg11 and a gene affecting efflux pump levels are responsible for the majority of the selective
advantage of an aneuploidy in the presence of different drugs (Selmecki et al. 2008; Sionov et al. 2010).
In addition, others have shown that aneuploidy confers adaptive characteristics and have identified the
genes involved (Yang et al. 2021; Beaupere et al. 2018). We have added more references that are
describing this phenomenon, and also refer now to “key response genes” instead of “key stress response
genes” in order to distinguish the response we are referring to from the general environmental stress
response. We have re-written the sentence and included additional references as follows: *“The increased
survival of aneuploid cells under stress can depend on one or more key response pathways that are
affected by genes on the aneuploid chromosome (Selmecki et al. 2006; Selmecki et al. 2008; Yang et al.
2019; Yang et al. 2021; Yona et al. 2012; Beaupere et al. 2018; Chen et al. 2012); they appear to act
through increased gene dosage. Moreover, the slower growth rates of some aneuploids and other general
features associated with aneuploidy (Tsai et al. 2019; Pavelka et al. 2010) might confer benefits as well. ”*

Technical comments

*1. The authors measure 933 strains, from which only data from 613 were used where the transcriptome and proteome could be matched. What exactly is the problem with the other data? Why could not be transcriptomes and proteomes of 1/3 of the samples matched? The authors state “Importantly, the exclusion of these strains did not bias the dataset for the study of aneuploidy” (Line 178, Page 5). How do they know? No data is shown to support this claim.*

We acknowledge that our original manuscript may not have fully clarified the process through which we arrived at the integrated dataset encompassing genomes, transcriptomes, and proteomes for 613 isolates. In response to the reviewer’s question, we have expanded the technical parts of the manuscript in the revision. The primary reason for excluding several strains pertains to the necessity of filtering at each level – genome, transcriptome, and proteome. Consequently, we had to omit any isolate where just one of these
datasets was not fully consistent or where data for a particular strain was unavailable. The impetus to improve data quality is primarily driven by the needs of data science. Machine learning and other regression methodologies are known to be acutely sensitive to the quality of data input. As the reviewer astutely pointed out (Reviewer #1, point #11), the average effect size across isolates at the proteome level can be moderate, necessitating precise data to draw solid conclusions from a biological perspective.

Our process began with an initial pool of 1,023 isolates. Applying rigorous filtering, we excluded strains with growth rates that were too low – a common confounding factor in proteomic studies. We also excluded samples with insufficient MS2 signal quality and those with total ion intensity and number of detected precursors that were either too small or too large. This rigorous filtering yielded proteomes for 796 isolates,
the proteomes of which are made available with this study (SI Table 6).

We had both proteomes (from this study) and transcriptomes (from Caudal et al. 2023 (Caudal et al. 2023)) available for 761 isolates. Gene copy number information, derived from Peter et al. (2018), was available for 759 of these 761 isolates. We further excluded 80 strains (as detailed in SI Table 3) from this dataset
due to inconsistencies between their annotated aneuploidy (from Peter et al. 2018) and the median gene copy number per chromosome. These discrepancies are likely due to segmental aneuploidies, shorter gene copy number variations, or ambiguities in threshold calling during aneuploidy assignment (see also response to Reviewer #1, minor comment #3).

Furthermore, we omitted 66 strains (as outlined in SI Table 4) where the aneuploidy determined at the genome level did not align with the transcriptomic data (refer to Materials and Methods). While we can only speculate about the causes of these mismatches, our hypotheses include unstable karyotypes, thresholding effects, and potential cross-contamination in a small number of strains. Importantly, we wish to highlight that no strains were excluded based solely on their proteome beyond the technical filters
described earlier.

We understand that our previous statement regarding the "unbiasedness" of the matched dataset might have been ambiguous. We must clarify that we cannot rule out the possibility that our stringent filtering procedures may have inadvertently eliminated certain unstable aneuploids. These could be isolates that
exhibited aneuploidy during genome sequencing but not during subsequent transcriptome determination, or vice versa. However, even if this was the case, making conclusive comparisons using non-matched data would be inappropriate. As a result, such removal was intentionally carried out.

We do not delve into the stability of aneuploids in this manuscript since this topic has been previously
explored (Hose et al., 2015; Pavelka et al., 2010). Nevertheless, our findings on the frequency of
aneuploidy suggest that most natural aneuploids are not unstable. This conclusion is further supported by
the fact that our filtering procedure did not alter the proportion of different ploidy levels relative to the original
genome dataset (Peter et al. 2018) (Fig. S2a). Moreover, the rarity of unstable aneuploids within the 1,011
*Saccharomyces cerevisiae* isolate collection is evident from our study. Despite the rigorous filtering, we
still capture every chromosome as aneuploid at least once, and in many instances multiple times, across
the 95 aneuploid strains (see Fig. S2b).

We have re-written the respective paragraph and added a caveat: *“It is possible that the strict filtering
against mismatched data points could bias the dataset against unstable aneuploids; these might have lost
or gained an aneuploidy during the generation of either the genome or transcriptome datasets (Peter et
al. 2018; Caudal et al. 2023). However, we achieved a similar distribution of ploidy levels in the integrated
dataset as were observed in the original genomes of the 1k collection (Peter et al. 2018), indicating that
most natural aneuploidies were sufficiently stable and did not bias the distribution of ploidy levels (Fig. S2a).”*

2. What is “biological signal”? (line 164, page 5)

We apologize for the unclear wording and have changed the sentence to read *“This detected technical
variation was much lower than the protein abundance variation across natural isolates (median CV of
32.8%) (Fig. S1b).”*

3. “1576 protein quantities” seems rather low (~ 25 % of the expressed yeast genome). The CCLE proteome allowed identification of significantly higher fraction of the expressed genome (Nusinow, Cell 2020). What is the reason?

We thank the reviewer for raising this question. In assessing the depth of our proteomes, one needs to
keep in mind that microbial and human cell line proteomes are very different matrices for proteomic
experiments. Because human cells, especially those cultivated in tissue culture and in rich media
formulations, contain many more highly abundant proteins, all discovery proteomic techniques identify
more proteins on human cell line protein extracts as they report in yeast or bacterial protein extracts, or
other matrices with a high dynamic range, such as human plasma or cerebrospinal fluid. We have
extensively benchmarked different acquisition schemes on these matrices, and find that this result is
largely independent of the proteomic method used (Messner et al. 2021; Wang et al. 2022; Szyrwiel et al.
2022).

Our data compares excellently with other yeast proteomic studies, even against many of much smaller
size, particularly in terms of quantitative precision and the low number (<4%) of missing values. Whilst in
very deep yeast proteomes, up to approximately ~4000 yeast proteins can be quantified (Ho,
Baryshnikova, and Brown 2018; Li et al. 2019), it is crucial to note that such experiments are typically
performed on lab strains, using chromatography of much lower throughput, often with pre-fractionation,
and indeed focus on a maximum number of proteins rather than on consistently quantifying proteins in a
large dataset. By quantifying more than 1500 proteins (corresponding to about 90% of the total proteomic
mass in yeast) consistently at very high precision across a collection of 796 natural isolates (SI Table 6),
our dataset stands out in the proteomic space.

Of note, the relative fraction of the expressed genome identified by Nusinow et al. in cell lines and our study in yeast indeed is comparable, despite the method used by Nusinov et al. being associated with significantly higher efforts to obtain a proteome for a lower number of samples. Nusinow et al. recorded proteomes for 375 cell lines and reported 12755 proteins detected in at least 1 or more samples, and 5143 proteins that they detected in all samples (Fig. 1b from (Nusinow et al. 2020)), equivalent to around 65% and 25% of the estimated size of the human proteome of 20,000 proteins. We measured proteomes of over 1000 yeast isolates, and, even *after* stringent precursor level pre-processing, found around 2100 proteins that were detected in at least one or more sample, and 1576 proteins that were detected in at least 80% of all isolates, corresponding to 31% and 23% of the *S. cerevisiae* proteome, respectively (see also Fig. S1d). Therefore, considering we deal with very different species and in our case with a higher number of samples plus genetically diverse natural isolates, the fraction of consistently quantified proteins in our study is comparable.

Minor comments

1. The figures are largely visually appealing and clear. However, the font is often very small and difficult to read in a print. Also, white text on a light green backgrounds is difficult to read. It would be great if the authors would adapt the figures for better legibility. Fig. 1 should be changed.

We appreciate the reviewer's feedback on our figures and apologize that they were in parts difficult to read in the original manuscript. We have adapted Fig. 1 to include the feedback we received (see Reviewer 1#, point #8 for a detailed description). We have also changed all instances of white text on green backgrounds to improve legibility of the figures. As for the font sizes to be used on the figures, we are using a minimal font size of 5 pt as required by the journal. However, we realized that we had scaled the figures in our original submission to 160 mm width and would like to apologize for this since it means that the figures and font sizes appeared too small. In the revised manuscript, we have inserted all figures with 180 mm width as required by the journal.

2. Were the disomic strains from Amon lab newly re-sequenced? In Methods it is stated that the status was confirmed by Torres et al, but this has been several years ago and aneuploid strains are genomically unstable and euploids regularly outgrow the aneuploids.

The reviewer is correct, aneuploid lab-engineered strains can be unstable and potentially revert to the euploid state. Therefore, we validated the aneuploidy in the disomes using proteomic data (see Materials and Methods, as well as previous Fig. S5, now Fig. S3). Indeed, two of the disomes had lost their extra chromosome (Disomes 12 and 14), and were thus excluded from our study, although we provide the processed data for the community (SI Table 7). We apologize that this information was somewhat buried in our previous manuscript, and highlight it now better.

Of note, the laboratory of Prof. Rong Li, who kindly provided us the Amon Disome collection, performed ploidy analysis and qPCR-based karyotyping when they received the collection themselves and found that Disome 13 was diploid with four copies of chromosome 13. Whilst we recorded the proteome for this strain and made it available as for the other Disomes (SI Table 7), we decided to not include it in any of the analyses of this manuscript.

*3. In the case of 95 aneuploid strains, was it always whole chromosome aneuploidy? Or where also segmental aneuploids considered?*

We thank the reviewer for raising this point. For this study, we consciously decided to focus on whole-chromosome aneuploidies. However, there are isolates carrying segmental aneuploidies within the 1k collection. In total, we found that 106 isolates of the collection exhibited a mismatch between the median gene copy number of at least one chromosome and the aneuploidy annotation provided in Peter et al. (Peter et al. 2018). 80 of these isolates are part of our complete dataset (so their proteomes are provided in SI Table 6), but were excluded when assembling the matched dataset (SI Table 3). We have highlighted this in the Materials and Methods section of the manuscript. Segmental aneuploidy is not the only reason why such a mismatch between median chromosomal gene copy numbers and aneuploidy annotation could occur since e.g., threshold effects when calling an aneuploidy can also play a role.

*4. Fig. 2a, the euploid natural strains – are these all haploids, or all diploids? Could authors eventually also mark the ploidy? The same for the aneuploids – it would be useful to distinguish the basal ploidy.*

We thank the reviewer for this helpful suggestion and have taken steps to improve the representation of the natural isolates, which have ploidies from 1n to 5n. Specifically, we have revised Fig. 1e (previous Fig. 2a) and are now showing the basal ploidy of the isolates to the top of the heatmaps. To enhance the clarity of the figure further, we have included a visual representation within the heatmap specifying the 518 (1n-4n) euploid natural isolates. As expected for euploid strains, these isolates exhibit negligible log₂ fold-changes in median per-chromosome expression levels.

*5. Figure 3a – the graphical depiction of the karyotypes is confusing. The best would be just to write it next to the plots. It would also save space. Haploids and diploids should be separated (bottom panel).*

We thank the reviewer for pointing this out and suggesting to improve our visualization of the karyotypes. We have improved Fig. 3a, b (also previously Fig. 3a, b) and related SI figures and now indicate the karyotypes as Xn +/- Y, with X denoting the ploidy, and Y the number of gained or lost chromosomes. We have also included the distribution plots for haploid euploid and diploid euploid, as well as duplicated chromosomes of haploid and diploid strains as an SI figure (Fig. S9a). These plots show clear dosage compensation for genes encoded on duplicated chromosomes for both haploid and diploid natural isolates, albeit the effect is visibly stronger in the nine haploid isolates than it is for the four diploid isolates. Please also see SI Table 14 for strain-wise information about dosage compensation effect size and significance. We consciously decided not to separate haploid and diploid isolates in the main figure (Fig. 3b) since we are quantifying dosage compensation there across ploidies, focusing on equivalent relative gains of chromosome copy numbers as the basis of comparison.

*6. Line 333, page 11 : “ . . . or related to transcription, which is altered in aneuploid to the aberrant chromosome number”. What do the authors mean? Do they now talk about the general transcriptional response to aneuploidy or the transcriptional changes of the genes encoded on the numerically aberrant chromosomes? They should change the sentence to avoid misinterpretations.*

We apologize for the ambiguous wording. We were referring to the expression changes (aneuploid vs. euploid strains) of genes that are expressed on a euploid chromosome in an aneuploid strains, i.e. in *trans*

to an aneuploid chromosome. We have changed the sentence in the revised manuscript to now read “To ask if specific biochemical or regulatory pathways are enhanced or reduced across natural aneuploids, we performed a KEGG gene set enrichment analysis (GSEA) focusing on the gene expression changes induced on euploid chromosomes. The gene set related to the ubiquitin proteasome system (UPS) was highly enriched among the differentially abundant proteins (Fig. 4b). Moreover, it was the only significantly enriched term that was neither related to potential differences in growth rate-related metabolism (i.e. oxidative phosphorylation, starch and sucrose metabolism), which is also slightly altered in the natural aneuploids (Peter et al. 2018; Scopel et al. 2021), nor related to transcription-associated processes (i.e. nucleotide metabolism, RNA transport, and RNA polymerases, Fig. 4b), that are plausibly associated with aneuploidy due to their increased need to synthesize DNA and RNA.”

7. Fig. S4 – there are no dark grey points visible.

This figure has been removed from the revised manuscript since we no longer include the re-analysis of previously published data on the induced-meiosis strains, instead citing the relevant literature (Pavelka et al. 2010; Larrimore et al. 2020).

Referee #2 (Remarks to the Author):

*The question of how aneuploidy affects cell physiology and promotes disease has a long history going
back to the incredible work of Boveri over 100 years ago. Over the last 15 years, yeast has been the setting
for many of the major advances and vigorous debates on this issue. It is a very interesting and also
fundamental problem. But is there anything new to say about it? This paper makes the pretty interesting
claims that wild strains of yeast follow different rules of aneuploidy tolerance than laboratory strains (i.e.,
superior dosage compensation), and that their more robust capacity in this relates to a physiological
adaptation involving the proteasome (i.e, higher proteasome levels). Interesting that the effect maps to the
proteasome and not to the chaperone pathway. The interesting panels 4B and 4C are key to the argument.
Fairly simple argument, which is good, and I think that if one can make it stick it can be a major paper, with
good novelty and generality. The issue of how domesticated laboratory strains differ from wild strains is
highly interesting and clearly germane not only to *S. cerevisiae*.*

We are grateful for the appreciation of the novelty of this work. The reviewer highlights the importance of
one of our main messages, that dosage compensation in natural isolates exceeds qualitatively and
960 quantitatively the gene-level dosage compensation observed in lab-generated synthetic aneuploids. We
found the reviewer's comments concerning the mechanism behind the observed dosage compensation
very helpful. Please find our point-by-point response to the reviewer's concerns below.

*0. I will focus on a key problem with the paper, maybe the key problem: validation of the proteasome effect.
In general the level of sophistication of the analysis tends to fall off when the argument reaches this point.*

This is an important general point: in addition to using the proteomes of natural isolates to address
generality and to solve discrepancies in the literature, as well as to compare synthetic with natural strains,
they can be exploited to identify new mechanisms, some of which may be difficult to detect in a laboratory
model. We agree that this opportunity was underdeveloped in the original manuscript. We therefore
substantially expanded our study in this direction.

First, following the direct suggestion of the reviewer, we addressed the role of the *RPN4* transcription factor
in mediating the increase of proteasome abundance in the natural strains (please see below in the specific
comments).

Second, in order to gain new mechanistic understanding, we focused on the observation that the structural
components of the ubiquitin proteasome system are induced in the aneuploid natural, but not in disomic
lab strains. Further, we analyzed the ubiquitinomic data reported in the original version of this manuscript
in a different way. While the data demonstrates that proteins encoded on aneuploid chromosomes are
more ubiquitinated, the data also shows that this increased ubiquitination does not occur in a
superstoichiometric manner in most isolates. On the one hand, this is an additional indication that the UPS
is not overloaded in the natural stains – in contrast to synthetic lab aneuploids. On the other hand, this
result argues against a hypothesis in which chromosome-wide dosage compensation is achieved by a
specific degradation of the proteins encoded on the aneuploid chromosomes. We hypothesized that
dosage compensation also could be achieved with a general increase in protein degradation. If that is the
case, we expect chromosome-wide dosage compensation to be associated with a global increase in
protein turnover. To gain insights into this possibility, we re-examined proteomic data from our recent yeast
deletion collection proteome project (Messner, Demichev, Muenzner, et al. 2023). We found that proteins

with a higher turnover rate, when encoded on an aneuploid chromosome, are more likely attenuated (Fig. 6f).

Third, to further substantiate the hypothesis, we teamed up with the lab of Matthias Selbach and conducted the first large-scale protein turnover study in natural isolates on any species. We found that chromosome-wide dosage compensation correlates with the global rate of protein turnover in natural isolates.

These results are included in new Fig. 5 and Fig. 6, as well as Fig. S14.

*1. Why do the wild strains show a perturbation of proteasome levels? This is essential to resolve and should not be terribly difficult I think. The proteasome is under complex control but as far as general controls over the level of the proteasome, it is at least thought to be pretty simple. The main control is via the transcription factor Rpn4, and a secondary control is via proteaphagy. Rpn4 has targets other than the proteasome however, so it should already be clear from the mRNA and protein data whether an Rpn4-driven compensatory response is operating. The other possible Rpn4-related hits may not assort into cookie-cutter GO terms, it has to be examined gene-by-gene. Also, if via Rpn4, proteasome upregulation would be readily evident at the RNA level. I may have missed this, but I didn't find any comment on this one way or another in the main text. So to be clear, it is critical to resolve whether Rpn4 has a role in this phenomenon. If not the authors might examine proteaphagy or whether a novel mechanism could be in play. One might look at Rpn4 itself, but that can be tricky as it is expressed at low levels and may have therefore fallen out of many of the datasets. Also one may see an effect on Rpn4 mRNA levels (a bit easier) but much of the regulation of Rpn4 is post-translational. If there is any suggestion of Rpn4 involvement, it raises the opportunity of genetic manipulation of Rpn4 in these strains to test specific hypotheses. For example, knocking out the RPN4 gene may normalize proteasome levels—easy experiment (also reducing state-state levels a bit in euploids) that could go right to the heart of things.*

Thank you for these valuable comments. Following the reviewer's suggestions, we analyzed the role of Rpn4 in controlling proteasome abundance in natural isolates. First, we would like to highlight that our recent parallel study investigated the proteomes of the yeast deletion collection in the lab background (Messner, Demichev, Muenzner, et al. 2023) and that study fully reflects the role of Rpn4 in controlling proteasome abundance in the lab strain (Fig. 5f, g in Messner et al., 2023 (Messner, Demichev, Muenzner, et al. 2023)).

To address the role of Rpn4 in natural aneuploid isolates, we conducted a series of analyses and obtained the following results:

- Rpn4 is a low abundant protein and was not quantified in the proteomics experiment. However, it was well captured in the transcriptomics data. We compared the mRNA levels of *RPN4* between aneuploid and euploid natural isolates. We found no significant difference in *RPN4* transcript abundances. Indeed, if there was any trend, *RPN4* mRNA levels were lower in the natural aneuploids (Fig. S14a).
- Within aneuploid strains, we could not detect a correlation between *RPN4* mRNA levels and the median abundance of structural components of the proteasome (KEGG) per strain (Fig. S14b).
- Rpn4 mRNA levels might obviously not reflect Rpn4 activity. Therefore, we asked if there is an induction of the Rpn4 regulome, including Rpn4 targets that are not necessarily proteasomal components. We extracted 116 genes annotated as Rpn4 targets from SGD. 51 of these genes had matched mRNA and protein abundance data in the natural isolate dataset. Indeed, *RPN4*

mRNA levels in natural isolates correlated with the mRNA levels of Rpn4 targets (Fig. S14c),
indicating that the Rpn4 regulome is conserved across the natural isolates. However, this Rpn4
regulome signature was only partially present at the protein level (Fig. S14d). In addition, whilst 33
of the 51 targets of Rpn4 showed – on average – induction across the natural aneuploids (Fig.
S14e, \log_2 protein expression ratios > 0), around a third of targets (18/51) were downregulated
(\log_2 protein expression ratios < 0). Most structural components of the proteasome show
upregulation at the protein level, and this upregulation is already present at the transcript level for
most of the genes (highlighted blue dots in Fig. S14e).

Taken together, these observations suggest that the Rpn4 regulome is not activated across the aneuploid
natural isolates, thus suggesting that it is not primarily Rpn4 activity that is responsible for the increase in
the proteasome abundance. This hypothesis is further supported by an additional analysis suggested by
the reviewer considering the frequency of the Rpn4-encoding chromosome being aneuploid (see Reviewer
#3, point #3, addressed below and in Fig. S14f-h).

The data also indicates that chromosome-wide dosage compensation is not generally associated with an
1055 increase in autophagy. Terms related to autophagy are downregulated in aneuploid versus euploid natural
strains (Fig. S15b, c). This result is in agreement with a high general protein turnover, which would reduce
the likelihood that damaged proteins accumulate. We updated the results section accordingly (lines 494ff.).
Instead, as shown below, our data links aneuploidy dosage compensation and protein turnover.

*2. Fig 4B (left) is interesting and important and looks good to the eye, but I don't feel comfortable in my
understanding of it. How many euploid strains are involved here, more importantly how may aneuploid?
Was only some fraction of the aneuploid strains used for this? Only the haploids? All of the haploids? Or
only haploids that are aneuploid for certain chromosomes (would not be surprising...)? Basically I read the
1065 legends as vague. More generally can one say that the aneuploid strains that were selected for
presentation were not cherry-picked on the basis of a property under examination?---I have no reason to
believe that but at the same time the text did not seem to clearly exclude that scenario. How may technical
replicates are involved (4B left)?*

We thank the reviewer for pointing out the unclear legend of Fig. 4c (previous Fig. 4b). No selection was
made, all 95 aneuploid isolates were included (Fig. 1c, d; SI Table 5, SI Table 15). The figure legend has
been improved accordingly, and the number of strains is explicitly stated in the Materials and Methods
section (line 1183). Due to the size of the natural isolate library, we did not perform replicate measurements
for proteomics, and instead decided to perform the comparison across strains of all ploidies for statistically
determining differential expression in *trans*, that is on euploid chromosomes in aneuploid strains. For the
lab-engineered disomic strains (less than 20 strains), biological triplicates were cultivated and analyzed as
described in the Materials and Methods section. Notably, enrichment of the proteasomal components in
natural isolates is also seen when only diploid strains are included, or when all diploid strains are excluded
from the analysis (Fig. S12d, also see reply to Reviewer #1, point #11), and when isolates with an
1080 aneuploidy of chromosome IV are excluded (see below, response to Reviewer #2, point #3).

*3. Rpn4 is present on chromosome 4. Did chromosome IV aneuploids show any perturbation of
proteasome levels?—I suspect so. Note that there is chromosome IV aneuploidy in the experimental (wild)
aneuploid strain CFV. How does Fig 4B (left) look when chromosome IV aneuploids are removed from the
pool? Are wild chromosome IV aneuploids better adapted to the aneuploid state?*

We thank the reviewer for raising this interesting question. In total, three isolates contain an aneuploidy of chromosome IV - isolates CFV, CAH, and CRI. Therefore, the frequency of chromosome IV aneuploidies is well below average, suggesting that chromosome IV aneuploidy is likely less well tolerated compared to other aneuploidies. One explanation for this could be the large size of chromosome IV - there is a general trend that aneuploidies of large chromosomes are less frequent than aneuploidies of shorter chromosomes (Scopel et al. 2021). In all three isolates, expression from the aneuploid chromosome is significantly attenuated at the protein level, but not at the mRNA level (SI Table 14, Fig. S14f). The isolates exhibit upregulation of the proteasome components in *trans* to the aneuploid chromosomes, but this upregulation is not significantly different from the one observed across other aneuploid isolates (Fig. S14g). Further, the upregulation of the proteasome remains apparent for the pool of aneuploid isolates even when isolates CFV, CAH, and CRI are removed (Fig. S14h, related to the previous Fig. 4b but without chromosome IV aneuploids). Taken together, this data is consistent with our other analysis shown above in response to Reviewer #2, point #1.

4. I would want to know the identity of each dot, each protein, in 4B (left). That may not fit into the main text of course but would be a good supplemental figure; same volcano plot, every blue dot labeled. For example, it could be that the positive hits are all from some specific part of the proteasome—there are 33 distinct gene products, counting only integral subunits. Are all 33 in there? I can't tell. And I would have a supplementary table showing those data as well as those for the many proteins that work hand in glove with the proteasome (I would be happy to prepare a list in that would help). I assume that no ubiquitin ligases or deubiquitinating enzymes seem to be affected—correct?

We thank the reviewer for their suggestion, adjusted the figure, and now label the dots in an SI figure as requested (Fig. S12b). To answer the question whether specific parts of the UPS are enriched, we obtained a UPS gene set from an extensive review article by Finley et al. (Finley et al. 2012). Of the around 200 proteins Finley et al. describe in their review, we quantified 55 in the natural yeast proteomes, including 10/10 proteins of the RP base, 13/14 proteins of the core particle, and 7/9 proteins of the RP lid. We find that especially structural components of the proteasome (proteasome core particle and RP base) are upregulated compared to their expression in euploid strains when expressed in *trans* to aneuploid chromosomes in natural yeast isolates (Fig. S12b, c, SI Table 15). We also quantified some ubiquitin ligases (2/11 E2 ligases, 8 E3 ligases) and 8 deubiquitinating enzymes in the dataset and they do not seem to be significantly regulated. We conclude that the proteasome itself is significantly upregulated in natural aneuploid yeast strains (with some inter-strain variability, see also Fig. S12a).

5. 4B and 4C are both interesting, very connected in spirit, but I'm not sure how they correlate. The questions are related to those directly above, how and why individual strains were chosen for analysis although the uncertainty relates mostly to 4B—without knowing that we can't make a confident comparison. Perhaps some of the information that I'm concerned with really is in the text, but if it is then at least I would think that these points could be explained more clearly to the reader.

Thank you. We have restructured this section of the manuscript. Since we observed that structural components of the UPS are upregulated across natural isolates, but not lab-engineered disomic strains (Fig. 4), we propose two hypotheses as to how an increase of the UPS could be linked to dosage compensation in aneuploids. First, proteins encoded on aneuploid chromosomes could be specifically targeted for degradation, as has been proposed for surplus protein complexes subunits that are non-

1135 exponentially degraded (McShane et al. 2016). Second, an increase in UPS components could reduce
protein levels via a higher general protein turnover rate.

To probe these hypotheses, we used three approaches. First, we expanded the analysis of the
ubiquitinomics profiling (now part of Fig. 5), showing that proteins encoded on aneuploid chromosomes of
1140 four natural isolates are stoichiometrically ubiquitinated, which argues against hypothesis #1. Second, from
new data measuring protein turnover rates for 55 natural isolates (Fig. 6), we found that the extent of
dosage compensation was positively correlated with an increase in turnover across isolates. Third, we re-
analyzed proteomic data from aneuploid strains all across the *Sc* gene deletion library (Messner,
Demichev, Muenzner, et al. 2023). This data, combined with previously published protein turnover data
from the Villén lab (Martin-Perez and Villén 2017) revealed that in aneuploid strains inadvertently present
in the deletion collection, proteins encoded on disomic chromosomes (1n+1) exhibited higher attenuation
if they had a comparatively short half life (lowest quartile of all turnover rates published by Martin-Perez et
al. (Martin-Perez and Villén 2017)) (Fig. 6f).

Taken together, these analyses support hypothesis 2, that general protein turnover and dosage
compensation are connected. We hope these new analyses, detailed in the results sections “Structural
components of the ubiquitin-proteasome system are induced differently in natural aneuploids and lab
disomes” and “Evidence that increased protein turnover rather than specific protein degradation explains
dosage compensation in natural aneuploids” help to better convey the connection of the UPS to the
1155 increase in protein turnover.

We resolved the previous ambiguities about which strains were included in the analysis concerning Fig.
4c (previous Fig. 4b) as noted in the reply to Reviewer #2, point #2.

*6. SI Fig 3c is another important component of the argument although put in the supplement. Not a hard
experiment but it needs work. On the one hand it is not good technically, but also, even apart from that it
would not be convincing. As above, it is not clear to me why we are looking at these four strains and
whether they are as representative as one would hope—perhaps they are, I just can't tell. The third set of
1165 plots (ARQ-CFV) is of no use, there simply have to be more replicates and it is not hard to see that the
one determination there for the euploid control is likely to be inaccurate. And for plots like this the y axis
should extend to zero. Actually better to have more than three replicates for each set of samples, as the
effect size is small and the sample to sample replicate variation not small. The second set is not showing
enough of a difference to be convincing. Why do this experiment so casually? There are no error bars and
1170 the legend speaks of a mean but no mean is shown. It is fine to measure proteasome activity using the
suc-LLVY-AMC assay. But it is essential to perform the assay +/- proteasome inhibitor and to subtract the
+inhibitor values from the -inhibitor values. Also it is recommended to perform the assay in the presence
and absence of 0.02% SDS; in the presence of SDS your readout is essentially of core particle levels as
the proteasome gate will be wide open.*

**AND**

*7. I felt uncomfortable with the pairing of wild strains according to their genetic closeness. Again this is
likely more a matter of presentation than of a methodological deficiency, it is just that I didn't find an
1180 objective measure of that shown, making it hard to evaluate the utility of these pairs. Also it would be
reassuring to know that these critical pairs of comparable strains are in each case well matched in their
growth properties, as well as can be expected.*

We appreciate the reviewer's constructive comments. We concur that while the results obtained with our proteasome activity experiments were in line with our conclusions, the interpretations did not achieve the level of robustness desired. As the reviewer notes, the technical challenges present are significant. However, a greater difficulty lies in choosing "representative" strain pairs for comparing UPS activities.

In the original manuscript, we acknowledged these limitations and presented the results of the proteasome activity experiments as complementary evidence. We have since included a considerable amount of new data concerning protein turnover, which is a more direct approach to underline our hypothesis. Given the lack of a straightforward approach to selecting "representative" isolates, and considering our newly added focus on protein turnover, we have chosen to omit the proteasome activity assays from our study.

*8. The difference between euploid and aneuploid strains highlighted in this work are subtle and I would recommend care as to whether they could have a metabolic basis. I would highly recommend a set of control experiments in which determinations of choice (eg that of SI Fig 3C or some proteomic analysis, which may give more precise values) are read across the span of a growth curve, say from OD 0.4 to OD 1200 3.0 (before and after the diauxic shift). As it is I had a hard time figuring out at what growth stage most of the experiments were done at and whether this was adequately controlled.*

Generally, experiments were performed using mid-log phase growth cultures and we now highlight this better in the Materials & Methods section (line 741).

To address the specific question of how protein expression from aneuploid chromosomes might differ between log phase vs. diauxic shift cultures, we performed an experiment to compare attenuation during the the growth of a diploid aneuploid strain, BBV, and a diploid euploid strain, AMC, in minimal medium from a low OD for 24 h and sampled them at five different time points from OD 0.3 (BBV)/0.4 (AMC) to OD 4.4 (BBV)/3.8 (AMC) (Fig. S10a). Isolates were grown in batch culture, pellets of equal cell density were lysed, prepared for proteomics, and proteomic experiments were conducted using DIA-MS (details in Materials & Methods section, lines 876ff.). The experiment was performed in biological triplicates. For each of the five time points, we calculated the median expression level per gene across the three replicates in the euploid strain, considering all proteins that had been measured in at least two of the three biological 1215 replicates of the euploid strain. Subsequently, for each strain and replicate, we calculated the log₂ fold-change between each protein and the median expression level determined in the euploid strain per time point, and normalized these log₂ protein expression values as described for the main proteomic dataset (see Materials & Methods (lines 1059ff.), as well as the reply to Reviewer #1, point #3). Relative expression levels between proteins encoded on euploid and aneuploid chromosomes were examined and the chromosomal average of protein expression from the aneuploid chromosome (chr III) in strain BBV was significantly increased compared to the expression of proteins from euploid chromosomes. This level was similar across all growth stages sampled (Fig. S10b) and comparable to the relative expression between the two isolates when measured as part of the whole 1000 isolate collection (Fig. S10c). Thus, at least in 1220 the analyzed strains, differences in protein expression resulting from aneuploidy were consistently observed across the growth phases. 1225

*9. Aggregates in yeast are often visualized using Hsp104-GFP fusion proteins. It may be good to give that a try in some of the strains. It is easy and has the potential to validate the idea that these strains have impaired proteostasis.*

We are grateful for the reviewer's insightful suggestion. However, our findings do not suggest that natural aneuploid strains experience an UPS overload, at least not to the extent previously proposed for synthetic aneuploids. Our newly incorporated data indicates not only an increase in protein turnover, but also a lack of detectable proteotoxic stress. For instance, key indicators such as HSP70, small heat shock proteins, autophagy, and vacuolar proteases, all exhibit downregulation in natural aneuploids. This evidence, counter to expectations of UPS overload, has been incorporated into our revised manuscript (Fig. S15). These findings serve to further differentiate the experiences of natural aneuploids from their synthetic counterparts, adding a nuanced understanding of their unique molecular landscapes.

Referee #3 (Remarks to the Author):

*In this manuscript, Muenzner et al. characterize the proteome of a large collection of wild and industrial yeast strains, and examine how the chromosome-wide dosage is compensated in aneuploids. For this study, the authors established a platform to quantify the proteome of nearly 800 strains. Several innovative steps were implemented to complete this tour de force that led the quantitation of ~1500 proteins across the examined strains. Notably, among the over 600 strains that could be matched to previously sequenced genomes (Peter J et al Nature 2018), nearly 100 strains were aneuploid, which is surprising given the fact lower cell fitness was observed in aneuploid yeast lab strains (Pavelka et al Nature 2010). The authors then examined in more details how the proteome is buffered in aneuploids, meaning how protein levels expressed from sur-numerous chromosomes are dampened. Similar to previous work (Dephoure N et al eLife, 2014), the authors found that this phenomena is controlled post-translationally and not at the mRNA level. They then showed that this buffering correlates with higher activity of the ubiquitin proteasome system (UPS). This study is interesting and unique. However, the main observations and conclusions are not entirely novel. While the authors show that the buffering may be greater in natural isolates in comparison to lab strains, the notion that many sur-numerous proteins are degraded or aggregate, especially if they are part of stable protein complexes is not new. Maybe the conclusions of this study would be more distinctive if the focus was on the overall proteome tuning, and how some proteins may be differentially expressed in natural isolates.*

We thank the reviewer for positively assessing our proteomic efforts, and for constructively commenting on the weaknesses of the study. We have substantially revised the manuscript as summarized in the “General remarks” section of this letter, and will reply point-by-point to the reviewer’s concerns. Overall, we think that the confusion about novelty emerged from our previously insufficient presentation of the distinct features of gene-specific versus chromosome-wide dosage compensation, and highlighting the differences between the current literature and the findings made in the natural strains (see also our general reply point II). We expanded both the section that highlights the substantial differences between dosage compensation in the synthetic and the natural strains (Fig. 2 and Fig. 3), and we also expanded the study to look at proteome tuning by including newly acquired protein turnover data for the natural isolates (Fig. 6).

*1. Dephoure et al eLife, 2014 previously showed that a significant proportion of proteins are expressed at lower levels than expected in disomic lab strains. This is notably shown in Figure 2B of the eLife publication, in which there is a clear binomial distribution. Similarly, the concept that the UPS is involved is not new, as Torres et al., Cell 2010 showed that aneuploid cells have an increased reliance on the UPS, and levels of proteasome subunits were shown to be more elevated in disomic lab strains (Figure 6 of Dephoure N. et al., eLife 2014).*

We thank the reviewer for this comment, which partially overlaps with points made by Reviewer #1 in questions #0.2 and #6, concerning the relationship of this work to previous studies in the lab-generated disomes (Torres et al. 2007, 2010; Dephoure et al. 2014). We would like to refer the reviewer to our answers to these points. In brief, we discover several important differences between natural and lab-generated aneuploids that exceed the differences between the two lab-generated collections of synthetic aneuploid strains that were intensely debated in the literature (Torres et al. 2007; Pavelka et al. 2010; Berman 2010). In order to better incorporate prior knowledge into our manuscript, we have included a new Fig. 2, in which we illustrate differences and similarities in gene-specific dosage compensation between the disomic strains and the natural isolates.

*2. Figure 2 and 3 could be combined and better quantitation should be provided. One issue is that, in*
Figure 2A, it appears that a large portion of strains have approximately equal mRNA and protein
enrichment. Including additional supplemental data with quantitative data could address this issue.
Interestingly the data in Fig 2B shows a large variability at the mRNA levels in AHS strains, suggesting a
1295 *potential larger buffering for other proteins; besides AHS, more examples should be provided. Fig 3C may*
be more appropriate for a supplemental material.

We agree and thus combined, extended and quantified the previous figures 2 and 3 as follows:

- previous panel 2b and previous Fig. 3 are now found as part of Fig. 3;
- - the qualitative illustrations for natural isolates (heatmaps) formerly in Fig. 2a are now a separate
panel in Fig. 1 (Fig. 1e), with improvements for visualization as suggested by Reviewer #1, minor
point #4;
- addition of a supplemental table that details the quantitation of attenuation at both the mRNA and
protein levels across strains (SI Table 13, quantifying Fig. 3b of the revised manuscript);
- - new analyses pertaining to strain-wise attenuation at both the mRNA and protein level (SI Table
14, quantifying the heatmap from Fig. 1e strain by strain and underlying the analyses shown in Fig.
3e-h);
- supplementary pdf file 1 provides plots as in previous Fig. 2b, now Fig. 3d for every individual
natural aneuploid isolate in the integrated dataset

We kept Fig. 3c in the main results section (also Fig. 3c in the revised manuscript), because it not only
clearly illustrates the difference in attenuation between the disomic strains and natural isolates, but also
allows quantification of the relative extent of attenuation at the mRNA and protein levels across basal
ploidies.

*3. Curiously, disomic lab strains analyzed in this manuscript seem to display a lower buffering capacity, in*
comparison to the previously published results (Dephoure N et al eLife 2014). For instance, many proteins
located on disomic chromosome 9 display levels similar to control cells in Figure 1D of the 2014
*publication, which is not the case in the data presented in Figure 2B of the current manuscript. One*
potential issue is that natural and disomic lab strains in this study were grown in different media (synthetic
minimum without amino acids vs. SD-HIS+G418), which is somewhat problematic.

We thank the reviewer for raising this question. To directly address this concern, we conducted a
comparison between the proteomic data of the disomic strain collection recorded by Dephoure et al.
(Dephoure et al. 2014) and our proteomic data for the same collection obtained through DIA-MS. In short,
the two sets of proteomic data agree very well (Fig. S16), supporting the conclusion that differences
described between the lab-engineered strains and the natural isolates are not attributable to technical
factors.

In more detail, Dephoure et al. 2014 performed two kinds of proteomics experiments: in one, they grew
the disomic strain collection on selective medium (-Lys-His+G418) and acquired the proteomes using
SILAC (Fig. 1b of the 2014 paper); in the other one, they grew the disomic strain collection on YPD and
measured the proteomes using a TMT labeling approach (Fig. 1d of the 2014 paper). The SILAC media is
1335 much closer to our growth condition, and therefore, we compared our proteomics data to the SILAC data.

The Disomes 01, 02, 05, 09, 10, 11, 15, 16, and the W303 wild-type strain were sufficiently stable so that the aneuploidy was maintained in both the previous and our experiments. For all strains, the distributions of relative protein expression (relative to the euploid wild-type W303 strain) for both euploid and aneuploid chromosomes correlate very well (Fig. S16a). Indeed, there is no statistically significant difference in attenuation levels of proteins located on aneuploid chromosomes between the Dephoure et al. 2014 data and the DIA-MS data ($p \gg 0.05$ for all disomic chromosomes, Fig. S16b). The comparison is included as a supplemental figure.

As the reviewer correctly points out, media composition could affect the attenuation. However, Dephoure et al. compared proteomic data of the disomic strains cultivated in both rich medium (YPD) and in SILAC medium (SD-Lys-His+G418 with added light or heavy lysine) and concluded that “on average, increases in gene copy number lead to proportional increases in mRNA and protein levels independent of growth conditions” (Dephoure et al. 2014). Also, as described in the reply to Reviewer #2, point #8, we included a growth curve analysis in this revision, which shows that the pervasiveness of the aneuploidy on the proteome is largely robust across different growth phases (newly added Fig. S10).

4. Enrichment of K-GG on proteins encoded on aneuploid chromosome may not be a true enrichment. The increase of ubiquitination may be simply due to higher protein levels. The authors need to normalize to the abundance of the proteins from which the K-GG peptides originate, to account for the increased abundance of proteins. They should also try to better correlate observed reduced levels at the proteome level in comparison to increased ubiquitination. Significance of this potential enrichment (Figure 4C) and of the elevated proteasome activity (Figure S3C) should be examined with statistical tests.

We followed up on this suggestion in the context of the new analyses performed, and it led to an interesting finding that aligns well with our new results obtained using turnover measurements. Ubiquitinomic data was compared with protein abundance measurements, highlighting another difference between gene-level dosage compensation as observed in the synthetic aneuploids and the chromosome-level dosage compensation in the natural strains: the proteins encoded on the aneuploid chromosomes are more ubiquitinated than protein expressed from euploid chromosomes, but in three out of the four strains analyzed, they were ubiquitinated stoichiometrically relative to the observed relative abundance change of the proteins due their increased gene dosage (Fig. 5b). This result suggests that these proteins are not specifically targeted for degradation (as seen with excess protein subunits (McShane et al. 2016)), but rather, that the increased abundance of the UPS may indicate a general increase in protein turnover in the natural isolates. For this reason, we re-analyzed a recent proteomic dataset acquired by our group for the yeast genome-scale deletion collection (Messner, Demichev, Muenzner, et al. 2023). In this dataset, we found that proteins that have a fast turnover are better dosage-compensated if encoded on aneuploid chromosomes (included as Fig. 6f in the revised manuscript). As noted above, a comprehensive dynamic SILAC experiment was performed to measure protein turnover in 55 euploid and aneuploid natural strains. These turnover measurements demonstrated that chromosome-level dosage compensation correlates with global protein turnover (Figure 6e).

The proteasome activity experiments (previous Fig. S3c) were removed from the revised manuscript as described in the response to Reviewer #2, point #6.

Perhaps, one potential unique future avenue for this study would be to delineate how elevated proteasomes are regulated in aneuploid cells, and how come chaperones and other elements of the protein homeostasis network are not solicited.

We thank the reviewer for this valuable suggestion. In the original manuscript, a GO term enrichment analysis placed the UPS system at the top of the list of processes induced in the natural aneuploids in *trans* (Fig. 4b). And, indeed, no gene sets related to chaperones and other elements of the protein homeostasis network were enriched in this analysis. However, a lack of enrichment at the gene-set level
does not preclude that individual proteins, or smaller functional units not captured by gene ontologies, are induced or downregulated. Thus, we conducted an orthogonal analysis of the data and categorized proteins of the proteostatic network as described by Rizzolo et al. (Rizzolo et al. 2017). Most of the terms and genes, such as HSP90 co-factors, HSP10, and HSP60, had no differences between euploids and aneuploids with three notable exceptions. Small heat shock proteins (HSP12, HSP26, HSP31), as well as
some components of the Hsp70 family (SSE2, SSA4, SSB1) were downregulated in the natural aneuploids. Similarly, proteins annotated with the KEGG term “Autophagy” and vacuolar proteases were downregulated across the natural aneuploids. We did not necessarily expect this result, but it matches the observation of an increased protein turnover that would also reduce protein damage. In contrast, chaperones of the chaperonin Cct ring complex and the prefoldin co-chaperone complex were
upregulated, consistent with increased protein turnover also requiring an increased translation rate. These observations are highly important, as they support the increased protein turnover in natural aneuploids, and show that there is no evidence that natural aneuploids suffer from proteostatic stress at least in the proteomic data.

*5. It should be noted that many of the examined strains are not necessarily “natural” but have been used by humans for many different processes (e.g. fermentation), in which aneuploidy may have been selected to cope with the additional stresses. Have the authors examined whether aneuploidy is more frequent in isolates associated to industrial activities vs. natural habitats?*

We agree with the reviewer, the collection of natural isolates contains both environmental and domesticated isolates, and we now add the specific definition in the manuscript. The relationship between aneuploidy and niche adaptation has been examined in the literature. On the basis of the same isolate library as used in our manuscript, Scopel et al. reported that isolates associated with bakeries, sake production, and “human activity” have higher frequencies of aneuploidy compared to natural strains
(Scopel et al. 2021). A similar conclusion was obtained by Gallone et al. in another collection of environmental versus domesticated isolates (Gallone et al. 2019). An intuitive explanation of this finding is the nutrient-rich niche fermentation processes provide. However, this observation does not systematically apply to all domesticated strains. For example, wine- and biofuel-associated strains are less likely aneuploid. Also, Scopel et al. describe that there is “no evidence for selection of specific chromosome amplifications in industrial strains”, and that, despite many potential confounding effects, the overall
“prevalence of chromosome gain and loss varies by clade and can be better explained by differences in genetic background than ecology”. We are now referencing the paper by Scopel et al. (Scopel et al. 2021) in the manuscript.

*6. Unless I missed it, growth rate of aneuploid cells in the 1011 collection is only discussed in the introduction (line 106) and the results presented in Figure S3A is not discussed anywhere else. It should be noted that in some cases, growth differences between lab disomic strains can be very subtle, and only*

*reliably quantified when both strains are mixed together in a competition growth assay. Therefore, one should be cautious when drawing conclusions from growth rate data.*

We agree with the reviewer. While published studies (i.e., Dephoure et al. 2014 (Dephoure et al. 2014)) suggest that different growth phases do not have a significant impact on dosage compensation despite differing growth rates, we scaled back the discussion about the role of growth rates and excluded previous Fig. S3a from the manuscript. Please also refer to Reviewer #2 point #8, where we investigated the impact of growth stages on dosage compensation, showing that the extent of attenuation stays constant over the span of a growth curve.

**Minor:**

Line 143: redundant statement “Indeed, to our knowledge, DIA-NN is, to the best of our knowledge...”

This is corrected in the revised manuscript.

Line 330: typo? “The proteasome was further the only enriched term...”

We apologize for the typo and have corrected the sentence in the revised manuscript.

Lines 808 – 809: Only uses ORFs with systematic names, so there is the potential that the effect is due to non-systematically named ORFs which we know are present in these strains.

We now explain the ORF name usage explicitly in the Materials & Methods section (lines 1022ff.).

In addition, we tested a pre-processing approach accounting for all 1640 non-canonical ORFs for which we had a representative sequence in the subset of strains in which each gene was detected and found that only ~30 proteins were detected in the final dataset. When detected, these few proteins had very low relative abundance, suggesting that most non-canonical ORFs are likely expressed below the detection threshold or are condition-specific. As a result, we think that these non-systematically named ORFs are unlikely to have a significant impact on the present study. We added these details to the manuscript (lines 184ff).

References

- Beaupere, Carine, Leticia Dinatto, Brian M. Wasko, Rosalyn B. Chen, Lauren VanValkenburg, Michael G. Kiflezghi, Mitchell B. Lee, et al. 2018. "Genetic Screen Identifies Adaptive Aneuploidy as a Key Mediator of ER Stress Resistance in Yeast." *Proceedings of the National Academy of Sciences of the United States of America* 115 (38): 9586–91.
- Berman, Judith. 2010. "When Abnormality Is Beneficial." *Nature* 468 (7321): 183–84.
- Brennan, Christopher M., Laura Pontano Vaites, Jonathan N. Wells, Stefano Santaguida, Joao A. Paulo, Zuzana Storchova, J. Wade Harper, Joseph A. Marsh, and Angelika Amon. 2019. "Protein Aggregation Mediates Stoichiometry of Protein Complexes in Aneuploid Cells." *Genes & Development* 33 (15-16): 1031–47.
- Caudal, E., Victor Loegler, F. Dutreux, N. Vakirlis, E. Teyssonnière, C. Caradec, A. Friedrich, J. Hou, and J. Schacherer. 2023. "Pan-Transcriptome Reveals a Large Accessory Genome Contribution to Gene Expression Variation in Yeast." *bioRxiv*. <https://doi.org/10.1101/2023.05.17.541122>.
- Cuypers, Bart, Pieter Meysman, Ionas Erb, Wout Bittremieux, Dirk Valkenburg, Geert Baggerman, Inge Mertens, et al. 2022. "Four Layer Multi-Omics Reveals Molecular Responses to Aneuploidy in Leishmania." *PLoS Pathogens* 18 (9): e1010848.
- Dephoure, Noah, Sunyoung Hwang, Ciara O'Sullivan, Stacie E. Dodgson, Steven P. Gygi, Angelika Amon, and Eduardo M. Torres. 2014. "Quantitative Proteomic Analysis Reveals Posttranslational Responses to Aneuploidy in Yeast." *eLife* 3 (July): e03023.
- Finley, Daniel, Helle D. Ulrich, Thomas Sommer, and Peter Kaiser. 2012. "The Ubiquitin-Proteasome System of *Saccharomyces Cerevisiae*." *Genetics* 192 (2): 319–60.
- Gallone, Brigida, Jan Steensels, Stijn Mertens, Maria C. Dzialo, Jonathan L. Gordon, Ruben Wauters, Florian A. Theßeling, et al. 2019. "Interspecific Hybridization Facilitates Niche Adaptation in Beer Yeast." *Nature Ecology & Evolution* 3 (11): 1562–75.
- Gasch, Audrey P., James Hose, Michael A. Newton, Maria Sardi, Mun Yong, and Zhishi Wang. 2016. "Further Support for Aneuploidy Tolerance in Wild Yeast and Effects of Dosage Compensation on Gene Copy-Number Evolution." *eLife* 5 (MARCH2016): 1–12.
- Ho, Brandon, Anastasia Baryshnikova, and Grant W. Brown. 2018. "Unification of Protein Abundance Datasets Yields a Quantitative *Saccharomyces Cerevisiae* Proteome." *Cell Systems* 6 (2): 192–205.e3.
- Hose, James, Chris Mun Yong, Maria Sardi, Zhishi Wang, Michael A. Newton, and Audrey P. Gasch. 2015. "Dosage Compensation Can Buffer Copy-Number Variation in Wild Yeast." *eLife* 4 (MAY): 1–28.
- Larrimore, Katherine E., Natalia S. Barattin-Voynova, David W. Reid, and Davis T. W. Ng. 2020. "Aneuploidy-Induced Proteotoxic Stress Can Be Effectively Tolerated without Dosage Compensation, Genetic Mutations, or Stress Responses." *BMC Biology* 18 (1): 117.
- Li, Jiaming, Joao A. Paulo, David P. Nusinow, Edward L. Huttlin, and Steven P. Gygi. 2019. "Investigation of Proteomic and Phosphoproteomic Responses to Signaling Network Perturbations Reveals Functional Pathway Organizations in Yeast." *Cell Reports* 29 (7): 2092–2104.e4.
- Martin-Perez, Miguel, and Judit Villén. 2017. "Determinants and Regulation of Protein Turnover in Yeast." *Cell Systems* 5 (3): 283–94.e5.
- McShane, Erik, Celine Sin, Henrik Zauber, Jonathan N. Wells, Neysan Donnelly, Xi Wang, Jingyi Hou, et al. 2016. "Kinetic Analysis of Protein Stability Reveals Age-Dependent Degradation." *Cell* 167 (3): 803–15.e21.
- Messner, Christoph B., Vadim Demichev, Nic Bloomfield, Jason S. L. Yu, Matthew White, Marco Kreidl, Anna-Sophia Egger, et al. 2021. "Ultra-Fast Proteomics with Scanning SWATH." *Nature Biotechnology* 39 (7): 846–54.
- Messner, Christoph B., Vadim Demichev, Julia Muenzner, Simran K. Aulakh, Natalie Barthel, Annika Röhl, Lucía Herrera-Domínguez, et al. 2023. "The Proteomic Landscape of Genome-Wide Genetic Perturbations." *Cell* 186 (9): 2018–34.e21.
- Messner, Christoph B., Vadim Demichev, Ziyue Wang, Johannes Hartl, Georg Kustatscher, Michael Müllleder, and Markus Ralser. 2023. "Mass Spectrometry-Based High-Throughput Proteomics and Its Role in Biomedical Studies and Systems Biology." *Proteomics* 23 (7-8): e2200013.

- Messner, Christoph B., Vadim Demichev, Daniel Wendisch, Laura Michalick, Matthew White, Anja
Freiwald, Kathrin Textoris-Taube, et al. 2020. "Ultra-High-Throughput Clinical Proteomics Reveals
Classifiers of COVID-19 Infection." *Cell Systems* 11 (1): 11–24.e4.
- Müller, Johannes B., Philipp E. Geyer, Ana R. Colaço, Peter V. Treit, Maximilian T. Strauss, Mario
Oroshi, Sophia Doll, et al. 2020. "The Proteome Landscape of the Kingdoms of Life." *Nature* 582
(7813): 592–96.
- Nusinow, David P., John Szpyt, Mahmoud Ghandi, Christopher M. Rose, E. Robert McDonald 3rd,
Marian Kalocsay, Judit Jané-Valbuena, et al. 2020. "Quantitative Proteomics of the Cancer Cell Line
Encyclopedia." *Cell* 180 (2): 387–402.e16.
- Okada, Hirokazu, H. Alexander Ebhardt, Sibylle Chantal Vonesch, Ruedi Aebersold, and Ernst Hafen.
2016. "Proteome-Wide Association Studies Identify Biochemical Modules Associated with a Wing-
1530 Size Phenotype in *Drosophila Melanogaster*." *Nature Communications* 7 (September): 12649.
- Okada, Hirokazu, Ryohei Yagi, Vincent Gardeux, Bart Deplancke, and Ernst Hafen. 2019. "Sex-
Dependent and Sex-Independent Regulatory Systems of Size Variation in Natural Populations."
Molecular Systems Biology 15 (11): e9012.
- Pavelka, Norman, Giulia Rancati, Jin Zhu, William D. Bradford, Anita Saraf, Laurence Florens, Brian W.
Sanderson, Gaye L. Hattem, and Rong Li. 2010. "Aneuploidy Confers Quantitative Proteome
Changes and Phenotypic Variation in Budding Yeast." *Nature* 468 (7321): 321–25.
- Peter, Jackson, Matteo De Chiara, Anne Friedrich, Jia-Xing Yue, David Pflieger, Anders Bergström,
Anastasię Sigwalt, et al. 2018. "Genome Evolution across 1,011 *Saccharomyces Cerevisiae*
Isolates." *Nature* 556 (7701): 339–44.
- Rizzolo, Kamran, Jennifer Huen, Ashwani Kumar, Sadhna Phanse, James Vlasblom, Yoshito Kakiyara,
Hussein A. Zeineddine, et al. 2017. "Features of the Chaperone Cellular Network Revealed through
Systematic Interaction Mapping." *Cell Reports* 20 (11): 2735–48.
- Schukken, Klaske M., and Jason M. Sheltzer. 2022. "Extensive Protein Dosage Compensation in
Aneuploid Human Cancers." *Genome Research* 32 (7): 1254–70.
- Scopel, Eduardo F. C., James Hose, Douda Bensasson, and Audrey P. Gasch. 2021. "Genetic Variation
in Aneuploidy Prevalence and Tolerance across *Saccharomyces Cerevisiae* Lineages." Edited by M.
Hahn. *Genetics* 217 (4). <https://doi.org/10.1093/genetics/iyab015>.
- Selmecki, Anna, Maryam Gerami-Nejad, Carsten Paulson, Anja Forche, and Judith Berman. 2008. "An
1550 Isochromosome Confers Drug Resistance in Vivo by Amplification of Two Genes, ERG11 and
TAC1." *Molecular Microbiology* 68 (3): 624–41.
- Sionov, Edward, Hyeseung Lee, Yun C. Chang, and Kyung J. Kwon-Chung. 2010. "Cryptococcus
Neoformans Overcomes Stress of Azole Drugs by Formation of Disomy in Specific Multiple
Chromosomes." *PLoS Pathogens* 6 (4): e1000848.
- Stingele, Silvia, Gabriele Stoehr, Karolina Peplowska, Jürgen Cox, Matthias Mann, and Zuzana
Storchova. 2012. "Global Analysis of Genome, Transcriptome and Proteome Reveals the Response
to Aneuploidy in Human Cells." *Molecular Systems Biology* 8 (608).
<https://doi.org/10.1038/msb.2012.40>.
- Szyrwiel, Lukasz, Ludwig Sinn, Markus Ralser, and Vadim Demichev. 2022. "Slice-PASEF: Fragmenting
All Ions for Maximum Sensitivity in Proteomics." *bioRxiv*. <https://doi.org/10.1101/2022.10.31.514544>.
- Terhorst, Allegra, Arzu Sandikci, Abigail Keller, Charles A. Whittaker, Maitreya J. Dunham, and Angelika
Amon. 2020. "The Environmental Stress Response Causes Ribosome Loss in Aneuploid Yeast
Cells." *Proceedings of the National Academy of Sciences of the United States of America* 117 (29):
17031–40.
- Torres, Eduardo M., Noah Dephoure, Amudha Panneerselvam, Cheryl M. Tucker, Charles A. Whittaker,
Steven P. Gygi, Maitreya J. Dunham, and Angelika Amon. 2010. "Identification of Aneuploidy-
Tolerating Mutations." *Cell* 143 (1): 71–83.
- Torres, Eduardo M., Tanya Sokolsky, Cheryl M. Tucker, Leon Y. Chan, Monica Boselli, Maitreya J.
Dunham, and Angelika Amon. 2007. "Effects of Aneuploidy on Cellular Physiology and Cell Division
in Haploid Yeast." *Science* 317 (5840): 916–24.
- Torres, Eduardo M., Michael Springer, and Angelika Amon. 2016. "No Current Evidence for Widespread
Dosage Compensation in *S. Cerevisiae*." *eLife* 5 (MARCH2016): 1–19.
- Tsai, Hung Ji, Anjali R. Nelliati, Mohammad Iqbal Choudhury, Andrei Kucharavy, William D. Bradford,
Malcolm E. Cook, Jisoo Kim, et al. 2019. "Hypo-Osmotic-like Stress Underlies General Cellular

Defects of Aneuploidy.” *Nature* 570 (7759): 117–21.

Tsai, Hung-Ji, Anjali R. Nelliath, Andrei Kucharavy, Mohammad Iqbal Choudhury, Sean X. Sun, Michael C. Schatz, and Rong Li. 2020. “On the Transcriptomic Signature and General Stress State Associated with Aneuploidy.” *arXiv [q-bio.GN]*. arXiv. <http://arxiv.org/abs/2007.14585>.

Wang, Ziyue, Michael Mülleler, Ihor Batruch, Anjali Chelur, Kathrin Textoris-Taube, Torsten Schwecke, Johannes Hartl, et al. 2022. “High-Throughput Proteomics of Nanogram-Scale Samples with Zeno SWATH MS.” *eLife* 11 (November). <https://doi.org/10.7554/eLife.83947>.

Yahya, G., P. Menges, P. S. Amponsah, D. A. Ngandiri, D. Schulz, A. Wallek, N. Kulak, et al. 2022. “Sublinear Scaling of the Cellular Proteome with Ploidy.” *Nature Communications* 13 (1): 6182.

Yang, Feng, Vladimir Gritsenko, Hui Lu, Cheng Zhen, Lu Gao, Judith Berman, and Yuan-Ying Jiang. 2021. “Adaptation to Fluconazole via Aneuploidy Enables Cross-Adaptation to Amphotericin B and Flucytosine in *Cryptococcus Neoformans*.” *Microbiology Spectrum* 9 (2). <https://doi.org/10.1128/spectrum.00723-21>.

Yona, Avihu H., Yair S. Manor, Rebecca H. Herbst, Gal H. Romano, Amir Mitchell, Martin Kupiec, Yitzhak Pilpel, and Orna Dahan. 2012. “Chromosomal Duplication Is a Transient Evolutionary Solution to Stress.” *Proceedings of the National Academy of Sciences of the United States of America* 109 (51): 21010–15.

Zelezniak, Aleksej, Jakob Vowinckel, Floriana Capuano, Christoph B. Messner, Vadim Demichev, Nicole Polowsky, Michael Mülleler, et al. 2018. “Machine Learning Predicts the Yeast Metabolome from the Quantitative Proteome of Kinase Knockouts.” *Cell Systems* 7 (3): 269–83.e6.

Reviewer Reports on the First Revision:

Referees' comments:

Referee #1 (Remarks to the Author):

The revised manuscript by Muenzner et al is significantly improved in comparison with the first version. In particular they now made very clear the key difference between natural isolates and laboratory-engineered aneuploid cells, and elucidated why this is important. The new figures are much easier to follow and contribute to better understanding of the key message. While the underlying mechanism is far from being clarified, they bring forward important results, which provide plausible and interesting explanation of the observed chromosome-wide attenuation of gene expression in aneuploid cells. In summary, the manuscript provides a tour-de-force experimental approach, impressive and very useful data set, as well as a novel view on dosage compensation in response to aneuploidy, and can be recommended for publication.

There are some small suggestions that would help further improve the manuscript.

Major points:

1. Why is the protein level attenuation and mRNA-level attenuation identical in triploid strains? Figures 3b and 3g show this clearly, but the authors do not discuss the point. This is very surprising, particularly when compared with the strong difference observed in 4n cells. The authors should briefly comment on this.
2. The discussion is overly lengthy, and it often reads as a summary, redundant with the Results. Some parts are confusing (some examples below).
3. Lines 590 - 600: “. . . several of the trans signatures in synthetic aneuploids might not necessarily be adaptive, but could actually represent secondary gene expression . . .” This is confusing. First, in this context, “adaptive” and “secondary” are not necessarily excluding each other. Even secondary changes can be adaptive, and vice versa. Second, maybe the authors wanted to say that the trans signatures in synthetic aneuploids represent not an adaptive response, but rather an early stress response. If this is the case, then this is nothing new and has been proposed previously – the trans signature in engineered aneuploids is an early response to aneuploidy associated stresses, not an adaptation. The manuscript brings additional support for this notion by showing that adapted aneuploid do not have this response.
4. It is obvious that the attenuation natural strains is more general than in laboratory strains, yet again some statements are way too general –e.g. 630 :” Dosage compensation in natural aneuploids is thus far broader and stronger than the previously well-described attenuation of protein complex members in the lab-generated aneuploids 17.” This sentence evokes that in the engineered aneuploids only protein complex members are attenuated, which is not the case, as the authors themselves show (Fig. 2e).

Minor suggestions:

1. The authors did an impressive job in adjusting the referencing and reporting on previous published data in an unbiased, less generalized way. Yet, some sentences should be still adjusted. For example, they

state (95): “Thus far, the literature about the role of dosage compensation aneuploidy tolerance has been controversial.” This is true only for budding yeasts and the authors should make this point clearly. They explain in next sentence, that “In *Saccharomyces cerevisiae*”, but never make the point that this really affected only budding yeast. As far as this reviewer is concerned, there has not been a single paper published suggesting that there is no dosage compensation on protein level in human aneuploid cells.

2. I would suggest to split the sentence in row 90 – it is unnecessary long and lumps together two different messages. Alternatively, replace “and in humans” with “while in humans”.: 90: In eukaryotic parasites such as *Leishmania donovani* or *Giardia intestinalis*, aneuploidies have been associated with virulence and immune escape 20, and in humans, aneuploidies cause disorders such as Down syndrome, but they can also provide growth advantages to cancer cells, and are associated with malignancy, invasiveness, and drug tolerance 33,36–41.

3. Row 270, sentence “There was a high correlation between relative expression values calculated across all euploid strains, and ploidy-wise calculated relative expression values (Fig. S4), indicating that non-linear scaling of the proteome with ploidy 72 had no significant effect on the data normalization strategy.” It should be probably rather stated that it had no effect on the outcome of the used normalization strategy, than effect on the strategy itself

Referee #2 (Remarks to the Author):

This study by Muenzner et al attempts to resolve how dosage compensation is achieved in natural strains of yeast. As previously noted it is an interesting issue. The present study is a very long story but the answer they arrive at is simple: there is an enhancement of overall protein degradation, probably stemming from proteasome levels being elevated. Although the study is a heroic effort by a truly impressive collaborative group, and although the manuscript is in many ways improved, it comes down in the end to Fig 6d, the endpoint of the argument. One would like to see statistical significance to the outcome there but by the author's own account the difference between aneuploid and euploid strains in Fig 6d is not significant. I don't mean to say it's not real or not the answer but it is not demonstrated. It could well be a matter of too few n's in the euploid dataset--you have to wonder where the true median of a larger dataset of that kind would be. It is debatable to what extent the problem presented by 6d is rectified by 6e. I am more positive about the other side of the equation, the proteasome being implicated in dosage compensation (I count s12c as a key positive, and the "core particle" box in S12b is also, to me, convincing).

Minor points

1. I didn't quite get the title, it seems a bit nebulous, and doesn't seem to reflect the main argument of the paper according to my reading.

2. The main text starts up with a long aside on the so-called reproducibility crisis in science, which we

generally follow in newspapers. I don't see how that has any place in this particular text, which is anyway vastly too long for a Nature paper. The paper should stick to the main point.

3. Nature instructions call for fewer figures I think. I'd say Fig 1 could be in the supplement. Fig 5 as well, negative data basically, however hard-won. On the other hand as mentioned above I like S12c and the core particle panel of S12b, one of both could potentially be in the main Fig set to my mind. I'm a bit worried about the CP chaperone annotation, only two for the five known ones are presented, which could lead to stonger apparent effect than is the case. Are the data lacking? It would be good to know why. The RP chaperones are also under-annotated. The CP volcano looks totally clean by eye but where is the 14th subunit?

4. The legend to Fig 4a refers to expression levels being represented in gray but I couldn't detect any gray in the figure. Maybe it needs to be blown up for that--?

5. I found myself wondering whether there is a solid UPS enrichment if you remove the proteasome and proteasome chaperones from the analysis (Fig 4b).

Referee #3 (Remarks to the Author):

This is the second submission of the Muenzner et al. manuscript on the natural diversity of the yeast proteome. Overall, this is a significantly improved article with a substantial number of new experiments. They have also addressed numerous critics from previous revision. However, there are three major points that still need to be addressed before this article can be published in Nature. There are also several minor issues that need to be resolved.

1) As the authors highlight in their rebuttal letter, this manuscript is the first multi-omic study of such a large number of natural isolates. However, only a small portion of the data is really analyzed. This seems like a missed opportunity. The transcriptomics and proteomic analysis extend to 613 isolates, but less than 100 are aneuploids. More bluntly, what is the point of analyzing over 500 euploid strains when only a few dozen were required if the sole focus of this study is aneuploidy? A complete analysis of the data would be beyond the scope of this article and would be more appropriate for a different article format for another journal. Nonetheless, I feel there are several avenues that could be explored to address this point and make this manuscript stronger. This is something I indicated in my opening comments after reviewing the first version of the manuscript.

Some of the most obvious questions that I have are whether proteins with the highest attenuation in aneuploid isolates also show tighter control in euploid strains and/or more variability in their mRNA levels. What are the main characteristics of the proteins with higher and lower variability among euploid isolates? Are these genes more or less essential? Is there a group of genes with a particularly high dichotomy between mRNA and protein levels? All these questions are potentially related to buffering

and tolerance observed in aneuploidy strains. For instance, the authors already show that the standard deviation of buffered proteins in euploids is higher than that of non-attenuated proteins (Figure S8b). This data should be better highlighted. More generally, the authors have unique set of data to show how the proteome varies in yeast. Given the high genetic variability between isolates, has the proteome an intrinsic ability to buffer itself (similarly to proteins to supernumerary chromosomes), or is a certain plasticity actually tolerable.

2) While the first section of the article reads very well and is interesting, the manuscript becomes confusing, and the message becomes less clear in the middle, especially when the results in Figure 3 are presented.

The type of analyses presented in Figure 3 does not seem to be fundamentally different from the results presented in Figure 2a-c. Unless there is some important nuance that I have missed, are the results in Figure 3c not just another way to confirm the data in Figure 2a? Importantly, because these results are presented after Figs. 2d-g, in which a "higher level" analysis is provided, the Figure 3 results seem in part redundant with the prior section and interrupt the overall flow.

Overall, the data in 3a-b seem to show a more minor nuance between natural isolates and engineered strains, which weakens the main message (this was one of the reason I was previously not enthusiastic about this manuscript). Is the distribution of protein abundance on supernumerary chromosomes in Figure 3a really binomial? It definitely has a shoulder, but that could be driven by a few disomic strains where the attenuation amplitude is greater (this is important because the distributions of natural isolates mostly rely on aneuploidy from three different chromosomes). More appropriate would be to compare the distribution of proteins on the same chromosome. Indeed, when comparing the data of chromosome 9 (between the disomic and selected aneuploid isolate shown; Figure 3d), there is no apparent difference in the distribution. Instead, the main difference is in the amplitude, which was already nicely shown in Figure 2b-c.

Therefore, the manuscript would be strengthened if results in Figure 3 were integrated with results in Figure 2a-c, and the second part of Figure 2 were presented afterward. Some of these analyses need to be better executed and presented, as these results are very interesting. First, I am confused about the number of proteins analyzed. In Figure S6, the authors present data for 680 proteins from disomic strains but only 107 proteins from natural isolates. In the previous section, it was indicated that 827 genes were analyzed from natural isolates (line 283). Why are only 107 proteins considered? The method sections also indicates that 630 (and not 680) genes were analyzed for disomic strains (line 99). Then in the next panel (Figure S6b), 283 proteins are then shown for natural isolates, without any good explanation to describe their origin. In this case, the number of proteins in each category (attenuated vs. not) should also be specified. Thirdly, statistical significance in Figure S6c should be tested. One issue is that only 52 proteins are considered in this analysis. This is likely due to the fact that proteins are only considered if they are represented in an aneuploid from at least three isolates, which basically only allows deriving data from proteins on chromosomes I, III, and IX. In contrast, proteins in engineered strains are not filtered with the same criteria because a disomy is only represented by one strain. In

other words, why not include proteins from all aneuploid isolates?

The results in Figure S8 are very interesting. However, the analysis in 8b has to be redone because a t-test is not adequate, as data are not normally distributed in many cases (the non-parametric Wilcoxon test would be more appropriate in these conditions). The p-values would also need to be corrected as the same data are tested multiple times. The number of proteins in each bin should also be specified. As indicated in point 1) some of these results are very interesting but not well emphasized.

3) Finally, the data on protein turnover cannot be properly evaluated due to a lack of details, and there are some important considerations that may have been overlooked. Therefore, I cannot agree with the authors' conclusions. Measuring protein turnovers in so many isolates is a major effort, and the authors should be commended. But, unless I missed it, there are no details in the methods section that explain how the calculations were made for protein turnover and turnover rates (the unit has not been specified for the latter). Surprisingly, some strains display very low average protein turnover in the profiles shown in Figure 6a. Could some of these differences be caused by much slower or faster uptake of SILAC labeled lysine residues? SILAC is normally done in *lys2Δ* (and *arg4Δ*) cells that are isogenic. In this case, none of the analyzed strains were deficient in lysine synthesis, and amino acid transport could be very variable across different genetic backgrounds. More information needs to be provided, and the authors should ensure that lysine uptake does not influence the results.

The capstone results presented in Figure 6e are very weak. The correlation seems to be driven only by a few data points. Importantly, the marginal difference between aneuploid and euploid isolates in Figure 6d is simply not significant. Additionally, the authors overlooked the fact that part of their argument is somewhat "circular." Because there is a large number of proteins on supernumerary chromosomes that are attenuated (i.e., degraded), these proteins could reduce the average protein half-life in these strains. Therefore, the analysis would need to be redone by excluding these proteins from the analysis and/or showing that the increased turnover rate applies to all proteins regardless of whether they are on a supernumerary chromosome. Another element that the authors may not have taken into consideration is that while proteasome subunits are, on average, expressed at higher levels in aneuploid isolates, this is only the case in 1/3 to 2/5 of the cases (Figure S12a). In about 1/2 of aneuploid isolates, average proteasome subunit levels are not altered, while they are even reduced in ~1/8 of the strains. I am not sure how the strains were selected in Figure 6, but it could be that the proposed increased turnover may only be a strategy adopted in some isolates where proteasomes are expressed at higher levels.

Minor comments

I don't understand the significance of the first paragraph in the introduction. The reproducibility crisis is an issue but how do the authors address this and why is that related to their study is not clear to me. This should be rewritten so that the manuscript is better introduced.

It is unclear which portion of aneuploidy is from a complete chromosomes gain/loss and what threshold was used (I may have missed that info in the method section). More importantly, how often smaller

portions of chromosomes are duplicated or loss, which could account for additional change in RNA/protein levels. This is also related to point 1.

While the proteomic results are shared in the supplementary tables, I was not able to see the RNA-seq results. It would be important to provide these results as well.

The results ~line 205 should also specify the number of precursors/peptides quantified across the strains.

Figure S1f. Specify number of replicates

Figure S1g. Redo Venn diagram to better represent the high overlap

Figure S8c. What data was compared to derived the p-values in this analysis?

Figure 4c. Many data points have the same p-values with jumps in between. This should not be the case. Perhaps some values were rounded or not properly converted in the pipeline (p-values seem relatively low).

Figure 6b and 6c should be moved to SI

Fig S14f-h. I do not understand why these results are shown. They need to be cited in the text or more information should be provide in legend. In h, the authors indicate that only strain with chromosome IV are analyzed but then indicate 92 strains were considered that correspond to total number of aneuploids strains

The order of figure panels often don't match the order of appearance in the text

Author Rebuttals to First Revision:

Referees' comments:

Referee #1 (Remarks to the Author):

*The revised manuscript by Muenzner et al is significantly improved in comparison with the first version. In particular they now made very clear the key difference between natural isolates and laboratory-engineered aneuploid cells, and elucidated why this is important. The new figures are much easier to follow and contribute to better understanding of the key message. While the underlying mechanism is far from being clarified, they bring forward important results, which provide plausible and interesting explanation of the observed*
*chromosome-wide attenuation of gene expression in aneuploid cells. In summary, the manuscript provides tour-de-force experimental approach, impressive and very useful data set, as well as a novel view on dosage compensation in response to aneuploidy, and can be recommended for publication.*

We thank the reviewer for positively evaluating our revision.

There are some small suggestions that would help further improve the manuscript.

*Major points:*

1. Why is the protein level attenuation and mRNA-level attenuation identical in triploid strains? Figures 3b and 3g show this clearly, but the authors do not discuss the point. This is very surprising, particularly when compared with the strong difference observed in 4n cells. The authors should briefly comment on this.

The attenuation at the mRNA level in triploid aneuploid isolates indeed appears stronger than in diploid or tetraploid isolates. However, there that are only four aneuploid triploid isolates with a single relative chromosome copy number change across all aneuploid chromosomes present in the dataset (in contrast to a higher number of haploid and diploid
isolates, SI Table 12). Given the high diversity of natural strains, this observation thus might be an outlier rather than a true biological difference. We have added a notice about this to the respective observation in the figure legend (Fig. S4d, previously Fig. 3g).

*2. The discussion is overly lengthy, and it often reads as a summary, redundant with the Results. Some parts are confusing (some examples below).*

We acknowledge our discussion was lengthy, we have rewritten the discussion and shortened and streamlined it considerably.

3. Lines 590 - 600: “. . . several of the trans signatures in synthetic aneuploids might not necessarily be adaptive, but could actually represent secondary gene expression . . .” This

*is confusing. First, in this context, “adaptive” and “secondary” are not necessary excluding*
each other. Even secondary changes can be adaptive, and vice versa. Second, maybe the
authors wanted to say that the trans signatures in synthetic aneuploids represent not an
adaptive response, but rather an early stress response. If this is the case, then this is nothing
new and has been proposed previously – the trans signature in engineered aneuploids is an
50 *early response to aneuploidy associated stresses, not an adaptation. The manuscript brings*
additional support for this notion by showing that adapted aneuploid do not have this
response.

We apologize for the ambiguous wording, indeed the term ‘adaptive’ is typically used in the
evolutionary sense - but it is also often used, as we used in this particular paragraph, to
55 indicate a regulated, functional response. We have extensively rewritten the text to avoid
the term in the context of the trans signatures.

The novelty of this paragraph stems from the observation of transcriptional changes
resembling the ESR, APS, and CAGE signatures in the natural isolates and showing the
60 lack of directionality of the signatures (sometimes they are upregulated, sometimes
downregulated), as well as the mitigation of these responses at the proteome level. Our data
thus suggests that ESR, APS, and CAGE are strain-specific transcriptomic signatures that
are mitigated at the proteome. We conclude that they are not representative of a
generalizable response to aneuploidy, as it was suggested previously.

4. It is obvious that the attenuation natural strains is more general than in laboratory strains,
yet again some statements are way to general –e.g. 630 :” Dosage compensation in natural
aneuploids is thus far broader and stronger than the previously well-described attenuation
*of protein complex members in the lab-generated aneuploids 17.” This sentence evokes*
than in the engineered aneuploids only protein complex members are attenuated, which is
not the case, as the authors themselves show (Fig. 2e).

We have revised this paragraph to make it more concise. We aimed to put our results in the
75 context of previous literature, in which dosage compensation in the disomes, but also in
aneuploid mammalian cells, was attributed to attenuation of surplus protein complex
subunits^{1,2}, rather than our own proteomic analysis of the disomes. It is correct that our own
results put a slightly more nuanced picture to the previous findings in disomes. Nonetheless,
also in our dataset, we find that protein complex subunits dominate dosage compensation
in the disomes. We revised the respective results section accordingly now using more
neutral language.

Minor suggestions:

1. The authors did impressive job in adjusting the referencing and reporting on previous
published data in an unbiased, less generalized way. Yet, some sentences should be still
adjusted. For example, they state (95): “Thus far, the literature about the role of dosage
compensation aneuploidy tolerance has been controversial.” This is true only for budding

*yeasts and the authors should make this point clearly. They explain in next sentence, that*
“In Saccharomyces cerevisiae”, but never make the point that this really affected only
budding yeast. As far as this reviewer is concerned, there has not been a single paper
published suggesting that there is no dosage compensation on protein level in human
aneuploid cells.

We agree, the controversy was triggered by diverging results between the synthetic yeast
aneuploid strain collections generated by the Amon and Li labs. However, we would like to
emphasize that debate – around both the findings and the divergence between the findings
– received broad attention and influenced the aneuploidy literature far beyond the yeast
field. Next to the systematic nature of these investigations, an important aspect is that
synthetic yeast aneuploids represent a state in which cells are not yet adapted to aneuploidy,
and that the yeast data achieved excellent signal to noise – especially because the
aneuploids could be directly matched to a euploid parent. In many models, it remains a
challenge to make similar contrasts, either because the direct parental cells might not be
known, or, because one can only sample cells that are already adapted to aneuploidy. One
way of interpreting our data that contrasts disomes with natural aneuploids is therefore that
longer-term adaptation to aneuploidy changes the picture originally presented for yeast;
suggesting that the adapted, natural yeasts better resemble the situation reported for cancer
cells or aneuploid parasites. We believe that specifically for this reason, our results are also
important information for work in other models, specifically those where one might be more
likely to see only the “adapted” stage, but not the situation before cells adapted to
aneuploidy, where it is not possible to make such comparisons.

In order to avoid any misleading impression, we restructured the respective introduction
paragraph and now start with the results obtained in mammalian cells and *L. donovani*,
before coming to results in yeast and the controversies triggered not only by the differences
between the synthetic yeasts and mammalian cells, but also by the disparate results
between the two synthetic strain collections themselves. We also improved and shortened
the discussion, highlighting the contrast between naive and adapted aneuploids. We indeed
believe that this is an aspect that gives additional value to our manuscript.

2. I would suggest to split the sentence in row 90 – it is unnecessary long and lumps together
two different messages. Alternatively, replace “and in humans” with “while in humans”:
*In eukaryotic parasites such as Leishmania donovani or Giardia intestinalis, aneuploidies*
have been associated with virulence and immune escape 20, and in humans, aneuploidies
cause disorders such as Down syndrome, but they can also provide growth advantages to
cancer cells, and are associated with malignancy, invasiveness, and drug tolerance 33,36–
41.

We thank the reviewer for this suggestion. Please see the response to minor question 1; we
have restructured and improved this paragraph.

3. Row 270, sentence “*There was a high correlation between relative expression values calculated across all euploid strains, and ploidy-wise calculated relative expression values (Fig. S4), indicating that non-linear scaling of the proteome with ploidy 72 had no significant effect on the data normalization strategy.*” It should be probably rather stated that it had no effect on the outcome of the used normalization strategy, than effect on the strategy itself

Thank you for the comment; we changed the sentence as suggested to: “*There was a high correlation between relative expression values calculated across all euploid strains, and ploidy-wise calculated relative expression values (Fig. S9), indicating that non-linear scaling of the proteome with ploidy had no significant effect on the outcome of the used data normalization strategy.*”

Referee #2 (Remarks to the Author):

*This study by Muenzner et al attempts to resolve how dosage compensation is achieved in natural strains of yeast. As previously noted it is an interesting issue. The present study is a very long story but the answer they arrive at is simple: there is an enhancement of overall protein degradation, probably stemming from proteasome levels being elevated.*

We thank the reviewer for evaluating our manuscript, and for the many positive statements, but also for the constructive critique in the previous and this revision round. We reply point-to-point as below.

*Although the study is a heroic effort by a truly impressive collaborative group, and although the manuscript is in many ways improved, it comes down in the end to Fig 6d, the endpoint of the argument. One would like to see statistical significance to the outcome there but by the author's own account the difference between aneuploid and euploid strains in Fig 6d is not significant. I don't mean to say it's not real or not the answer but it is not demonstrated.*
*It could well be a matter of too few n's in the euploid dataset--you have to wonder where the true median of a larger dataset of that kind would be. It is debatable to what extent the problem presented by 6d is rectified by 6e. I am more positive about the other side of the equation, the proteasome being implicated in dosage compensation (I count s12c as a key positive, and the "core particle" box in S12b is also, to me, convincing).*

We thank the reviewer for providing this extremely useful feedback and apologize for our illustration in Fig. 6d, which was confusing because it implied the relationship between turnover and dosage compensation might be not statistically proven in natural isolates. When comparing the spread of turnover over all isolates (the boxplot the reviewer is referring to, previous Fig. 6d, now Fig. 4d), it is important to keep in mind that there is not only a broad spread in turnover rates, but also in dosage compensation across natural isolates.
One would expect a signal of increased turnover in isolates with a corresponding signal in dosage compensation. We have now expanded Fig. 4d (previous Fig. 6d) by adding a column specifically highlighting the aneuploid strains with strong dosage compensation to the
boxplot. In comparison to the euploid strains, these isolates show significantly elevated protein turnover.

**Fig. 4d:** Comparison of median turnover rates in euploid isolates vs. all aneuploid isolates or isolates exhibiting high attenuation. P-values were determined using two-sample Wilcoxon tests.

Having said that, we agree with the reviewer that providing additional evidence about the role of protein turnover in chromosome-wide dosage compensation in natural aneuploid isolates would strengthen this study. In addition to the results shown above, the previous version contained a re-analysis of proteomic data acquired as part of the yeast gene deletion collection³, which revealed that, when encoded on an aneuploid chromosome, high turnover proteins exhibit higher dosage compensation than proteins with a low turnover. We now start Fig. 4 with this result. Furthermore, we now added important new analysis results. These show that:

i) In natural isolates, more proteins exhibit an increase in protein turnover rates when the protein is expressed from an aneuploid chromosome (and thus likely dosage compensated) vs. when the same protein is expressed from a euploid chromosome in a euploid or aneuploid strain (included as Fig. 4f in the revision).

ii) The increase of attenuation with higher overall protein turnover (previous Fig. 6e, now Fig. 4e) can be attributed to increasing attenuation of proteins with rising overall protein turnover rates, meaning that for the majority of proteins encoded on aneuploid chromosomes, attenuation increases when they are expressed in a high-turnover isolate compared to a low-turnover isolate (Fig. 4e of the revised manuscript). As an example, we illustrate this in Fig. 4g using protein Age2: in euploid isolates, Age2 levels do not correlate with overall isolate turnover rates. However, when expressed from aneuploid chromosomes, relative protein levels of Age2 decrease with increasing isolate turnover rate (negative correlation). A similar relationship is especially pronounced for the group of protein complex members (Fig. 4i), which are also among the top class of dosage-compensated proteins.

iii) Within euploid isolates, structural components of the proteasome behave differently. Most of them do indeed show increased abundance with increasing overall

isolate turnover (Fig. S8c), supporting the reported association between protein
turnover and proteasome abundance.

Minor points

1. I didn't quite get the title, it seems a bit nebulous, and doesn't seem to reflect the main argument of the paper according to my reading.

We have now changed the title to read: “*The natural diversity of the yeast proteome links aneuploidy tolerance to protein turnover*”. We welcome any input from editors and reviewers.

2. The main text starts up with a long aside on the so-called reproducibility crisis in science, which we generally follow in newspapers. I don't see how that has any place in this particular text, which is anyway vastly too long for a Nature paper. The paper should stick to the main point.

We apologize if this aspect came across as overemphasized; it is now shortened to half a sentence.

3.1 Nature instructions call for fewer figures I think. I'd say Fig 1 could be in the supplement. Fig 5 as well, negative data basically, however hard-won. On the other hand as mentioned above I like S12c and the core particle panel of S12b, one of both could potentially be in the main Fig set to my mind.

We substantially shortened the paper to make it more concise and to fit within the page and character limits of the journal, and thank the reviewer for their useful suggestions regarding figure organization. We reduced the number of main figures to four by i) moving the previous Fig. 5 into the SI material, as suggested here, and by ii) combining the results detailing the quantification of dosage compensation (former Figs. 2 and 3) into a single figure (now Fig. 2) in response to a suggestion from reviewer 3. We also now include panel (c) from previous Fig. S12 in the main Fig. 3d. To focus better on the core message, we moved two panels (1c, d) from Fig. 1 to the supplementary material and removed panel 1b completely from the manuscript as it is sufficiently described in the Results section and Materials and Methods.

3.2 I'm a bit worried about the CP chaperone annotation, only two for the five known ones are presented, which could lead to stonger apparent effect than is the case. Are the data lacking? It would be good to know why. The RP chaperones are also under-annotated. The CP volcano looks totally clean by eye but where is the 14th subunit?

We thank the reviewer for asking about coverage of UPS components. For our systematic analysis, we used the systematically curated KEGG terms, and for specific analysis, we obtained gene annotations from the literature as indicated. In particular, the following UPS

components referring to the annotation provided by Finley et al., 2012 ⁴ are quantified in the proteomes:

UPS component	number of proteins according to Finley et al., 2012	number of proteins quantified in dataset
core particle	14	14
RP base	10	10
RP lid	9	7
proteasome-associated proteins	9	5
CP chaperones	5	2
RP chaperones	4	1
E1 enzyme	1	1
E2 ligases	11	2
E3 ligases - HECT E3s	5	1
E3 ligases - Rsp5 adaptors	17	1
E3 ligases - RING E3s	42	3
E3 ligases - Ubox proteins	2	1
E3 ligases - RBR E3s	2	0
E3 ligases - CRL core components	7	2
E3 ligases - F-box proteins	22	0
E3 ligases - Substrate receptors of Cul3 and Rtt101 ligases	7	0
E3 ligases - APC cyclosome core components	13	0
E3 ligases - APC cyclosome substrate receptors	3	0

DUB	20	8
-----	----	---

265 The proteomes thus provide complete coverage of the 14 core particle components and the
RP base as listed by Finley et al., 2012, as well as almost complete coverage of the
regulatory particle lid. The proteomic dataset quantifies 2 of 5 CP chaperones and 1 of 4 RP
chaperones; other subunits are below the detection threshold. We would note that the
270 proteomic method is untargeted; thus, even if a protein is not captured by either the KEGG
terms or the literature curation used, it is still in the dataset if expressed within the dynamic
range covered.

We thank the reviewer for spotting that the 14th subunit of the core particle in the proteomic
data was missing from the Volcano plots. It turns out this was due to an annotation mistake
caused by the ambiguous gene symbol *PRS3* used by Finley et al., 2012⁴). In the systematic
yeast gene annotation, *PRS3* refers to the enzyme YHL011C, 5-phospho-ribosyl-1(alpha)-
pyrophosphate synthetase – a low abundance enzyme that we did not quantify. We now
corrected the gene symbol, which recovered the 14th subunit of the core particle that is now
displayed in the volcano plots. We thank the reviewer for noticing this annotation problem
and have corrected it in the manuscript (affecting Fig. S5b and Fig. 3d, which are the
previous Fig. S12b, c), as well as in the SI Table 14 (where the 14th subunit was already
present in the last version of the manuscript, but not annotated as “core particle”).

*4. The legend to Fig 4a refers to expression levels being represented in gray but I couldn't
detect any gray in the figure. Maybe it needs to be blown up for that--?*

We thank the reviewer for this suggestion. We now included a high-resolution pdf of the
heatmap, including isolate names, in the supplementary material to enable easier
comparison between isolates for the interested reader. This also makes it easier to detect
the grayed out parts of the heatmap (supplementary pdf 2). In addition, we modified the
figure legend to read “*Genes that are located on aneuploid chromosomes in a respective
isolate are omitted from trans expression analyses and therefore shown in gray.*”.

*5. I found myself wondering whether there is a solid UPS enrichment if you remove the
proteasome and proteasome chaperones from the analysis (Fig 4b).*

In order to assess *trans* expression of wider UPS components in aneuploid isolates, we split
the UPS annotation as provided by Finley et al., 2012⁴ as shown in previous Fig. S12b, c
(now Fig. S5b and Fig. 3d). This detailed analysis revealed that it is *specifically* the structural
components – core particle and RP base – that are increased in *trans* across aneuploids at
the proteome level, and this specificity accounts for the enrichment of the KEGG term
“Proteasome” (Fig. 3b-d). We have edited the text for clarity.

Referee #3 (Remarks to the Author):

*This is the second submission of the Muenzner et al. manuscript on the natural diversity of the yeast proteome. Overall, this is a significantly improved article with a substantial number of new experiments. They have also addressed numerous critics from previous revision. However, there are three major points that still need to be addressed before this article can be published in Nature. There are also several minor issues that need to be resolved.*

We thank the reviewer for positively assessing our revised manuscript. We appreciate the helpful suggestions to improve the study even further, and have addressed the concerns point-by-point below.

*1) As the authors highlight in their rebuttal letter, this manuscript is the first multi-omic study of such a large number of natural isolates. However, only a small portion of the data is really analyzed. This seems like a missed opportunity. The transcriptomics and proteomic analysis extend to 613 isolates, but less than 100 are aneuploids. More bluntly, what is the point of analyzing over 500 euploid strains when only a few dozen were required if the sole focus of this study is aneuploidy? A complete analysis of the data would be beyond the scope of this article and would be more appropriate for a different article format for another journal. Nonetheless, I feel there are several avenues that could be explored to address this point and make this manuscript stronger. This is something I indicated in my opening comments after reviewing the first version of the manuscript.*

We acknowledge and appreciate the reviewer's recognition of the value inherent in the study's multi-omic dataset of natural isolates, which we present as a significant resource to the community. We also agree that many research questions can be addressed with a dataset of this size, which spans genome, transcriptome, and proteome at high precision. 335 And we agree that we cannot fit all possible research directions in a single manuscript. Thus, we addressed all specific questions that relate to the biology of aneuploids, and thus fall within the scope of this manuscript (see the detailed comments below). In response to this point, we improved the discussion section, to better highlight several other types of questions that will profit from using this dataset as a resource.

As we outline in the next paragraph, we think there are good reasons to use aneuploidy as a 'use case' to demonstrate the power of a natural isolate library proteome. That said, of course, the euploid isolate data is a crucial component of the manuscript. First, it was the first step of our study to consider all isolates of the 1,011 genomes project, not at least to 345 identify the aneuploids, and to quality control the different -omics layers of the dataset so that potential mismatches are avoided. Second, many of the key conclusions are derived from comparing aneuploids with euploids. Here, it is important to keep in mind that one cannot make direct matches between euploid parents and aneuploid derivatives in a natural isolate library because of their high level of diversity. Importantly, the large size of the dataset 350 mitigates this constraint, allows us to improve the signal to noise ratio, and enhances the robustness and reproducibility of the conclusions. While using a smaller subset of euploid

isolates is possible for specific analyses, reducing the number of euploid isolates in this study would reduce statistical power and increase the risk of sampling biases, without providing a tangible benefit. Third, including the euploid isolate data is essential for the resource character of the dataset, and thus gives the study a much broader impact; i.e. because we started with all strains of the 1,011 genomes project and present high-quality proteomes for a total of 796 isolates, this dataset can be used by the community to address a broad range of research questions. In summary, we think that this study of aneuploidy gains both quality, impact, and resource character from the presence of a strong dataset that is based on the entire strain library of the 1,011 genomes project, which contains both euploids as well as aneuploids.

We chose to study aneuploidy tolerance because some other attractive options, like protein dynamics studies, are not uniquely dependent on a natural isolate proteome. For example, one of our recent publications includes general analyses of protein dynamics similar to the ones the reviewer notes, using a genome-wide *S.c.* knockout collection ³. By focusing on aneuploidy instead, we exploit the natural isolate proteomes, because i) it leverages a key finding that was made specifically in natural isolates, namely the unexpectedly high prevalence of aneuploids, and ii) questions of dosage compensation specifically manifest at the proteome level. Furthermore, we show that the analysis of natural isolates can be used to address the generalizability of findings made with lab strains, and can illuminate complex problems such as aneuploidy at the molecular level. Examples include: 1) finding the upregulation of structural components of the proteasome; 2) revealing that previously discussed *trans* transcriptional signatures (ESR, APS or CAGE) are all present and all mitigated in the proteomes; and 3) identifying clear evidence supporting the relationship between dosage compensation and protein turnover, two molecular phenotypes not readily accessible in the lab strains.

In summary, we chose to study aneuploidy tolerance, not only because it is an important problem, but also because it highlights the power afforded by the analysis of natural isolate libraries. In addition, however, we would like to emphasize that also the resource character of our study provides impact. Specifically because this study is based on the entire strain collection of the 1011 genes project, and because it includes all euploids and aneuploids, this work will facilitate comprehensive investigations of other important research questions with the data. Indeed, we intend for this resource to enthusiastically promote accessibility for such approaches, by making all of the data – most of which was not trivial to generate – openly available.

Some of the most obvious questions that I have are whether proteins with the highest attenuation in aneuploid isolates also show tighter control in euploid strains and/or more variability in their mRNA levels. What are the main characteristics of the proteins with higher and lower variability among euploid isolates? Are these genes more or less essential? Is there a group of genes with a particularly high dichotomy between mRNA and protein levels?

We thank the reviewer for this question. Essential genes have significantly lower variability in protein abundance across euploid isolates. In interpreting this result, one however needs

to keep in mind that essential genes are, on average, more abundant than non-essential genes, which confounds this relationship. This information has been included in Fig. S2g.

Fig. S2g: Variability (standard deviation) of protein abundance levels across euploid isolates for essential vs. non-essential genes. Distributions medians were compared using a two-sample Wilcoxon test.

The reviewer asks an interesting general question: do specific groups of proteins show differences in variability at the mRNA level versus the protein level? We detect an enrichment for metabolic enzymes to be more stable at the protein level compared to their mRNA level, and proteins related to posttranslational modification, RNA processing, or transport to be more variable in natural strains. We have not seen a similar enrichment among the proteins that are more or less dosage compensated, and thus we think this result is more indirectly related to the dose compensation in aneuploids. We have added this result in Fig. S2h.

■ FDR ≤ 0.05 ■ FDR > 0.05

Fig. S2h: Gene set enrichment analysis of the differences between mRNA and protein abundance variability (measured as standard deviation) across euploid isolates. Positive normalized enrichment scores indicate higher variability at the mRNA vs. the protein abundance level, negative normalized enrichment scores indicate higher variability at the protein vs. the mRNA abundance.

All these questions are potentially related to buffering and tolerance observed in aneuploidy strains. For instance, the authors already show that the standard deviation of buffered proteins in euploids is higher than that of non-attenuated proteins (Figure S8b). This data should be better highlighted.

We agree and now highlight this result in the manuscript (page 5, lines 159-161).

More generally, the authors have a unique set of data to show how the proteome varies in yeast. Given the high genetic variability between isolates, has the proteome an intrinsic ability to buffer itself (similarly to proteins to supernumerary chromosomes), or is a certain plasticity actually tolerable.

In short, our dataset demonstrates that the proteomes of natural isolates show diversity in both attenuation and turnover, and thus, both processes have plasticity. In general, the diversity and noise at the proteome level is lower than at the transcriptome level, which attributes a general buffering function to the proteome. This principle not only applies to aneuploidy. For example, we previously found that proteomic changes resulting from genetic perturbations are buffered³, while others found that co-expression signals resulting from the 3D-structure of the genome are buffered as well⁵. Our results concerning aneuploidy in natural isolates thus align well with the notion of the reviewer and the recent literature, which attributes a general buffering function of the proteome relative to the transcriptome. We apologize if these important discussion points were not highlighted well enough in the revised version of the manuscript, and again improved on this point by now including them in the discussion.

2) While the first section of the article reads very well and is interesting, the manuscript becomes confusing, and the message becomes less clear in the middle, especially when the results in Figure 3 are presented. The type of analyses presented in Figure 3 does not seem to be fundamentally different from the results presented in Figure 2a-c. Unless there is some important nuance that I have missed, are the results in Figure 3c not just another way to confirm the data in Figure 2a? Importantly, because these results are presented after Figs. 2d-g, in which a "higher level" analysis is provided, the Figure 3 results seem in part redundant with the prior section and interrupt the overall flow.

In revising the manuscript, we have considerably shortened and focused the manuscript while incorporating the reviewer's suggestions. Relevant to this specific point, we combined previous Figs. 2 and 3 into a single figure (Fig. 2), which addresses the quantification of dosage compensation in natural aneuploids. To improve clarity, we shortened the section

on the “higher level” analysis that the reviewer refers to considerably. We hope these improvements further increase the accessibility of our manuscript.

Overall, the data in 3a-b seem to show a more minor nuance between natural isolates and engineered strains, which weakens the main message (this was one of the reason I was previously not enthusiastic about this manuscript).

Please see the response to reviewer #1 in the previous revision, upon which we expanded the comparison between synthetic strains and natural isolates based on their input, and we apologize if the important point of the analysis in previous Fig. 3a-b (now Fig. S4a, b) did not get across. The previous literature did attribute dosage compensation in the disomes to the degradation of surplus protein complex subunits ². At the same time, other studies in

yeast did not detect an enrichment of protein complexes ⁶, resulting in debate which remained largely unresolved ⁷.

Previous Fig. 3a-b (now Fig. S4a, b) put a new light on these conclusions, because it shows that in natural isolates, dosage compensation affects proteins chromosome-wide. This is important for the impact of the manuscript because it demands for new mechanistic explanations: while the degradation of surplus protein complex subunits can be attributed to non-exponential degradation ¹, this principle cannot be applied to the broad set of proteins that are attenuated in the natural isolates. In order to make this message more clear, we rewrote, shortened and focused the section on specific protein properties. Specifically, we

reduced emphasis on the function of attenuated proteins, which was an important aspect in published work on synthetic aneuploids, and is less critical for natural isolates, because they dosage-compensate the majority (~70%) of proteins encoded on aneuploid chromosomes, including most functional groups in dosage compensation.

Is the distribution of protein abundance on supernumerary chromosomes in Figure 3a really binomial?

We apologize for the inaccurate terminology. The shoulder is explained by the members of protein complexes that are predominantly dosage-compensated in the synthetic aneuploids ². We used the term bimodal, because a "shouldered distribution" is often referred to as a bimodal distribution in the context of histograms. However, we agree that the term is suboptimal, and now refer to the “shouldered distribution”, and note that it is caused by the attenuation of specific protein groups, in particular protein complex subunits (page 5, lines

165-169).

It definitely has a shoulder, but that could be driven by a few disomic strains where the attenuation amplitude is greater (this is important because the distributions of natural isolates mostly rely on aneuploidy from three different chromosomes). More appropriate would be to compare the distribution of proteins on the same chromosome. Indeed, when comparing the data of chromosome 9 (between the disomic and selected aneuploid isolate

shown; Figure 3d), there is no apparent difference in the distribution. Instead, the main difference is in the amplitude, which was already nicely shown in Figure 2b-c.

We studied the concern of the reviewer. However, we can confirm that the shouldered distribution observed for the disomic strains is not caused by selected disomes exhibiting a stronger amplitude of attenuation; instead, the shouldered distribution can be observed for all disomes (Fig. S10a).

Therefore, the manuscript would be strengthened if results in Figure 3 were integrated with results in Figure 2a-c, and the second part of Figure 2 were presented afterward.

We adapted this suggestion as detailed above (this document, lines 457-463). Instead of moving the second part of Fig. 2 after the chromosome-wide quantification part, we have significantly shortened this section so that it is more to the point, and we hope it does not interrupt the flow.

Some of these analyses need to be better executed and presented, as these results are very interesting. First, I am confused about the number of proteins analyzed. In Figure S6, the authors present data for 680 proteins from disomic strains but only 107 proteins from natural isolates. In the previous section, it was indicated that 827 genes were analyzed from natural isolates (line 283). Why are only 107 proteins considered?

The number in Fig. S2d (previous Fig. S6b) is smaller than the total number of proteins quantified because in this analysis, we compared the 680 proteins from 9 strains that were lab-engineered disomes (each is haploid with one extra chromosome duplication) with the 107 proteins found on duplicated chromosomes in 13 natural isolates (haploid or diploid background). The overlap between genes for which we could perform a linear regression analysis and that are expressed on duplicated chromosomes in disomic lab strains ($n = 680$) as well as on duplicated chromosomes in natural isolates ($n = 107$) is 52 proteins. We apologize for any confusion and now provide this information in the figure legend.

The method sections also indicates that 630 (and not 680) genes were analyzed for disomic strains (line 99).

We apologize, and thank the reviewer for spotting this typo in the methods section – it is indeed 680 (not 630 proteins) and was corrected.

Then in the next panel (Figure S6b), 283 proteins are then shown for natural isolates, without any good explanation to describe their origin. In this case, the number of proteins in each category (attenuated vs. not) should also be specified.

The number of 283 proteins shown for natural isolates in Fig. S2d derives from the 52 genes described in Fig. S2c that are located on duplicated chromosomes in *multiple* aneuploid isolates. Each one of the 52 genes appears in – on average – 5.4 isolates on an aneuploid chromosome.

We have added the number of proteins in each category to the figure legend, which now reads: “*For the disomic strains, all 680 proteins were used to draw the protein expression distributions shown in (d); for natural isolates, the 52 overlapping proteins from (c) were used. Since these 680 proteins appear exactly once on a duplicated chromosome in the disomic strain (only one disomic isolate per chromosome in the collection), and the 52 overlapping proteins appear on duplicated aneuploid chromosomes in multiple natural isolates, the total number of datapoints used to draw the distributions is at $n = 680$, and $n = 283$ (> 52), respectively. The number of attenuated vs. not attenuated values in the distributions shown for the disomic strains was 325 and 355, respectively; and for the natural isolates, 145 and 138, respectively.*”

Thirdly, statistical significance in Figure S6c should be tested. One issue is that only 52 proteins are considered in this analysis. This is likely due to the fact that proteins are only considered if they are represented in an aneuploid from at least three isolates, which basically only allows deriving data from proteins on chromosomes I, III, and IX. In contrast, proteins in engineered strains are not filtered with the same criteria because a disomy is only represented by one strain. In other words, why not include proteins from all aneuploid isolates?

We apologize if Fig. S6c (now Fig. S2e) was confusing. The bar plot simply illustrates the counts of attenuated vs. not attenuated proteins from Fig. S2d (previous Fig. S6b). Since we are only comparing proteins expressed on *duplicated* chromosomes in lab-engineered strains and in natural isolates in Fig. S2d, e, the reviewer is correct that we are only looking at distributions and absolute numbers from chromosomes I and IX – these are the only chromosomes with duplications ($1n+1$ or $2n+2$) in both disomic strains and natural isolates. Importantly, the reviewer’s suggestion to change the filters used for the regression analysis of natural isolates would therefore not change the number of proteins available for the analysis presented in Fig. S6.

We have looked into the filtering strategy as the reviewer suggested. The reviewer is correct that our choice of parameters set a stringent filter on the number of data points that have to be available for the linear regression of relative expression levels over relative chromosome copy number changes in the natural isolates – we decided to include only gene expressed on an aneuploid chromosome in at least three different isolates that had not reverted to euploidy. However, this includes genes from 10 chromosomes (chromosomes I, III, IV, V, VI, VIII, IX, XI, XII, and XIV), apologies if our wording created the impression this comes from 3 chromosomes. Indeed, we could not apply the exact same filter to the engineered disome strains; rather, relative mRNA and protein expression values were derived from biological replicate measurements and thus exhibit lower potential deviation from the biological “ground truth” than do the single replicate measurements in the larger set of

600 natural isolates. We therefore conclude that the filters used in the regression analyses are sensible. We have revised the paragraph for clarity, and hope is satisfactory.

*The results in Figure S8 are very interesting. However, the analysis in 8b has to be redone because a t-test is not adequate, as data are not normally distributed in many cases (the non-parametric Wilcoxon test would be more appropriate in these conditions). The p-values would also need to be corrected as the same data are tested multiple times. The number of proteins in each bin should also be specified. As indicated in point 1) some of these results are very interesting but not well emphasized.*

Thank you for the suggestion. We now performed non-parametric Wilcoxon tests for this figure, and adjusted the p-values using the Benjamini-Hochberg method. We also explicitly state the number of proteins that fall in the “attenuated” (671) and “not attenuated” (260) categories in the respective main and supplementary figure legends (Fig. 2d and S3b). We also highlight the finding that proteins that are more tightly controlled in euploid strains are
more attenuated in the aneuploids in the main results section.

*3) Finally, the data on protein turnover cannot be properly evaluated due to a lack of details, and there are some important considerations that may have been overlooked. Therefore, I cannot agree with the authors' conclusions. Measuring protein turnovers in so many isolates is a major effort, and the authors should be commended. But, unless I missed it, there are no details in the methods section that explain how the calculations were made for protein turnover and turnover rates (the unit has not been specified for the latter).*

We thank the reviewer for appreciating our considerable efforts to measure protein turnover in natural isolates. The following methods section was provided:

*“RawFiles were analyzed with MaxQuant 1.6.7.0 using standard settings, with match between runs and requantify enabled, and the Uniprot S. cerevisiae protein database including isoforms (downloaded 02/09/23) selected for the database search. Complexity was set to 2 with Lys-8 set as heavy label. Further processing of the data and calculation of half-lives was done in R. First, the evidence.txt was loaded with the fread function from the data.table package, filtered for Lysine-containing peptides and cleaned from potential contaminants and remaining reverse hits. Due to many
proteins in yeast being very stable a correction for doubling times is not applicable to most identified proteins. Similar to Martin-Perez & Villén 2017 81 we therefore calculated turnover rates (kdp) and the corresponding half-lives without doubling time (SI Table 19) correction. In more detail, protein turnover rates were calculated for proteins with valid SILAC ratios in at least two time points per strain by building a
linear model from the different sampling time points against the log-transformed H/L ratios, thus calculating kdp. The corresponding slopes from each fit depict the kdp value for each strain. Half-lives were calculated from the resulting kdp as $\log(2)/kdp$ (SI Table 18). Isolate CLN was excluded due to a very low number of valid SILAC ratios obtained at $t = 135$ min. Furthermore, three proteins (ERP1, NEO1, and MDE1)*

were excluded from the dataset since they were measured only in few isolates (6, 4,
and 3, respectively) and exhibited very high variability in half-lives across these
strains.”

The unit of the median turnover rate per strain was inadvertently left out in panels Fig. 6d, e
of the previous version. They are now present in Fig. 4d, e.

*Surprisingly, some strains display very low average protein turnover in the profiles shown in
Figure 6a. Could some of these differences be caused by much slower or faster uptake of
SILAC labeled lysine residues? SILAC is normally done in lys2Δ (and arg4Δ) cells that are
isogenic. In this case, none of the analyzed strains were deficient in lysine synthesis, and
amino acid transport could be very variable across different genetic backgrounds. More
information needs to be provided, and the authors should ensure that lysine uptake does
not influence the results.*

We thank the reviewer for this thoughtful suggestion. To rule out that the results could be
confounded by lysine uptake, we performed lysine uptake experiments with all isolates used
for protein turnover measurements. In brief, we cultivated the natural isolates as well as a
lysine-auxotroph lab strain in minimal medium supplemented with 80 mg/L labeled lysine
(Lys-8, same concentration as used for the turnover experiments) for multiple generations
(overnight pre-culture, dilution to a median OD600, and subsequent incubation for 8h). We
harvested the cell pellets by centrifugation, and extracted and measured intracellular lysine
levels compared to a labeled internal standard by liquid chromatography selective reaction
monitoring (Materials and Methods). The experiment was performed in biological triplicates
for the natural isolates, and with six replicates for the lab strain. We observed that lysine
was taken up by all natural isolates (Fig. S7c).

**Fig. S7c:** Ratio (internal standard response ratio) between unlabeled (Lys-0, red) or labeled (Lys-8, cyan) intracellular
lysine and an internal quantification standard (Lys-4, Materials and Methods) in prototroph natural isolates and a lysine-
auxotroph laboratory strain (BY4742-HLU) after continuous growth in minimal medium (SM) supplemented with 80 mg/L
labeled lysine (Lys-8).

We also performed a second experiment where we assessed the ratios of intracellular
labeled and unlabeled lysine three hours after switch from unlabeled to labeled medium (as
performed when measuring protein turnover, see Materials and Methods). As above, these
experiments were performed across the natural isolates and the auxotroph lab strain in
biological triplicates (six replicates for the lab strain). We observed that for all isolates, the

685 fraction of intracellular labeled lysine present after three hours was much higher than the
 fraction of intracellular unlabeled lysine (Fig. S7d). Notably, the auxotrophic lab strain - which
 is fully dependent on extracellular lysine supply - exhibited a rather low Lys-8/Lys-0 ratio
 compared to the natural isolates. In addition, we did not observe any correlation between
 690 the Lys-8/Lys-0 ratio and the median turnover rates for the natural isolates as determined in
 the dynamic SILAC experiments (Fig. S7e). These results have been included in the
 manuscript.

**Fig. S7d:** Ratio between labeled and unlabeled intracellular lysine (Lys-8/Lys-0) in prototroph natural isolates and a lysine-
 auxotroph laboratory strain (BY4742-HLU) three hours after switch from growth in unlabeled (SM + 80 mg/L Lys-0) to
 labeled (SM + 80 mg/L Lys-8) medium. Aneuploid (red) and euploid (blue) natural isolates, as well as the lab strain (green)
 are highlighted. In (a-d), boxplot hinges mark the 25th and 75th percentiles, and whiskers show all values that, at maximum,
 fall within 1.5 times the interquartile range.

Fig. S7e: The relationship between Lys8/Lys-0 ratios from (d) and the median protein turnover rate per isolate (black dots)
 shows no correlation.

*The capstone results presented in Figure 6e are very weak. The correlation seems to be
 driven only by a few data points. Importantly, the marginal difference between aneuploid
 and euploid isolates in Figure 6d is simply not significant.*

We would like to refer to the first point from reviewer #2 (this document, lines 160-216). We
 have added additional data and illustrations that provide additional evidence, and clearly

substantiate the relationship of protein turnover and chromosome-wide dosage compensation in natural isolates.

Additionally, the authors overlooked the fact that part of their argument is somewhat "circular." Because there is a large number of proteins on supernumerary chromosomes that are attenuated (i.e., degraded), these proteins could reduce the average protein half-life in these strains. Therefore, the analysis would need to be redone by excluding these proteins from the analysis and/or showing that the increased turnover rate applies to all proteins regardless of whether they are on a supernumerary chromosome.

We thank the reviewer for this suggestion. Reassuringly, performing the analysis, we did not find a change in the correlation described in Fig. 4e when genes expressed on aneuploid chromosomes in aneuploid isolates were excluded (PCC = 0.31, $p = 0.037$, as before). This is now noted in the legend to Fig. 4e.

Another element that the authors may not have taken into consideration is that while proteasome subunits are, on average, expressed at higher levels in aneuploid isolates, this is only the case in 1/3 to 2/5 of the cases (Figure S12a). In about 1/2 of aneuploid isolates, average proteasome subunit levels are not altered, while they are even reduced in ~1/8 of the strains. I am not sure how the strains were selected in Figure 6, but it could be that the proposed increased turnover may only be a strategy adopted in some isolates where proteasomes are expressed at higher levels.

We agree that one part of the results reveals that natural isolates are highly diverse in their proteomes. However, natural isolates are also diverse in dosage compensation, and despite the diversity, the induction of the structural components of the UPS is statistically significant across all isolates.

The (degree of) induction of the UPS was no selection criteria for the turnover experiments; indeed our choice was conservative in order to not artificially affect the results. For the turnover experiments, we selected all 46 diploid aneuploid isolates with a single chromosome gain (trisomic strains) for which we quantified attenuation, two randomly chosen haploid aneuploid isolates with a single chromosome gain, as well as ten diploid euploid and two haploid euploid isolates with a similar range of growth rates as the aneuploid isolates (SI Table 16). We apologize if the isolate selection was not clear, we have expanded the respective paragraph to describe this on page 6 lines 222-224.

Minor comments

I don't understand the significance of the first paragraph in the introduction. The reproducibility crisis is an issue but how do the authors address this and why is that related to their study is not clear to me. This should be rewritten so that the manuscript is better introduced.

The introductory paragraph has been reorganized with far less emphasis on the reproducibility crisis.

*It is unclear which portion of aneuploidy is from a complete chromosomes gain/loss and what threshold was used (I may have missed that info in the method section). More importantly, how often smaller portions of chromosomes are duplicated or loss, which could account for additional change in RNA/protein levels. This is also related to point 1.*

The manuscript is based on the genome sequences, assemblies and aneuploidy annotations presented in Peter et al., 2018⁸, with mismatching data points removed in our quality control procedures. We did not consider dosage compensation in strains with partial duplications or deletions, in order to avoid the concern raised here (Methods). Thus, we agree that the manuscript does not answer the relationship of mRNA and protein levels for smaller CNVs. However, data for all isolates is available in our resource, such that this interesting problem can be studied in the future. A respective note was added to the
discussion.

*While the proteomic results are shared in the supplementary tables, I was not able to see the RNA-seq results. It would be important to provide these results as well.*

We thank the reviewer for paying attention to data availability. The RNAseq data is available in the European Nucleotide Archive (ENA) under the accession number PRJEB52153.

*The results ~line 205 should also specify the number of precursors/peptides quantified across the strains.*

Thank you. We added the number of precursors used to quantify the proteins across all strains to the sentence. It is 7946.

Figure S1f. Specify number of replicates

We now specify the number of replicates in the figure legend of Fig. S1f.

Figure S1g. Redo Venn diagram to better represent the high overlap

The Venn diagram was modified as suggested.

Figure S8c. What data was compared to derived the p-values in this analysis?

The p-values in Fig. S2i (previous Fig. S8c) are derived from one-sample t-tests comparing the mean of each group to the expected value if no attenuation occurs ($\mu = 1$), and adjusted using the Benjamini-Hochberg method. We added this information to the figure legend.

*Figure 4c. Many data points have the same p-values with jumps in between. This should not be the case. Perhaps some values were rounded or not properly converted in the pipeline (p-values seem relatively low).*

We performed gene-wise one-sample Wilcoxon tests to derive the fold-changes and significance levels for the data displayed in Fig. 3c (previous Fig. 4c). The rank-based character of this test on nine samples (nine disomic strains) explains the limited number of p-values and the “jumps” the reviewer describes. However, in this revision, we decided to change the performed test to a one-sample t-test since the normality requirement is fulfilled for the relative *trans* protein distributions that are compared, and using a t-test, which is not rank-based, will avoid these jumps that might confuse readers. We thank the reviewer for
bringing this issue to our attention and have changed all affected figures and legends accordingly (Fig. 3c, S5b, c, d, S6j, SI Table 14).

*Figure 6b and 6c should be moved to SI*

Fig. 6b, c are now found in Fig. S8a, b.

*Fig S14f-h. I do not understand why these results are shown. They need to be cited in the text or more information should be provide in legend. In h, the authors indicate that only strain with chromosome IV are analyzed but then indicate 92 strains were considered that correspond to total number of aneuploids strains*

The results shown in the former Fig. S14f-h (now Fig. S6h-j) were added in response to a specific request from reviewer #2 in the previous revision, addressing whether isolates with a chromosome IV aneuploidy could be skewing the reported results of proteasome enrichment across natural isolates, and were cited only briefly in the main text. In Fig. S6j (previous Fig. S14h), we analyzed all natural aneuploid isolates except for the ones that contain a chr IV aneuploidy. In total, we included 95 natural aneuploids in the integrated
dataset; of those, three carry a chromosome IV aneuploidy, leaving 92 isolates that were analyzed for the volcano plot shown in Fig. S6j. We improved the figure legends for Fig. S6h-j.

*The order of figure panels often don't match the order of appearance in the text*

This has been corrected in the revised manuscript.

References

- 1. McShane, E. *et al.* Kinetic Analysis of Protein Stability Reveals Age-Dependent Degradation. *Cell* **167**, 803–815.e21 (2016).
2. Dephoure, N. *et al.* Quantitative proteomic analysis reveals posttranslational responses to aneuploidy in yeast. *Elife* **3**, e03023 (2014).
3. Messner, C. B. *et al.* The proteomic landscape of genome-wide genetic perturbations. *Cell*
**186**, 2018–2034.e21 (2023).
4. Finley, D., Ulrich, H. D., Sommer, T. & Kaiser, P. The ubiquitin-proteasome system of *Saccharomyces cerevisiae*. *Genetics* **192**, 319–360 (2012).
5. Kustatscher, G., Grabowski, P. & Rappsilber, J. Pervasive coexpression of spatially proximal genes is buffered at the protein level. *Mol. Syst. Biol.* **13**, 937 (2017).
- 6. Pavelka, N. *et al.* Aneuploidy confers quantitative proteome changes and phenotypic variation in budding yeast. *Nature* **468**, 321–325 (2010).
7. Berman, J. Evolutionary genomics: When abnormality is beneficial. *Nature* vol. 468 183–184 (2010).
8. Peter, J. *et al.* Genome evolution across 1,011 *Saccharomyces cerevisiae* isolates. *Nature*
**556**, 339–344 (2018).

Reviewer Reports on the Second Revision:

Referees' comments:

Referee #1 (Remarks to the Author):

The manuscript 2022-04-05410B by Ralser and colleagues with the title: The natural diversity of the yeast proteome links aneuploidy tolerance to protein turnover has been improved significantly in this second revision. I particularly appreciate the shorter, more compact and focused text, clearer figures and an interesting discussion without overly general statements or unnecessary digressions. While the authors do not provide clear molecular mechanisms, they are three important outcomes of their work:

- naturally occurring aneuploids showed increased dosage compensation on protein level, likely to mitigate the negative effects of the aberrant chromosome numbers;

- this dosage compensation is associated with increased proteasome - mediated protein turnover;

- the provided datasets are unique and will serve the scientific community in many different fields.

The two findings will result in novel testable hypotheses which will contribute to better understanding of regulation of protein abundance in cells, and to adaptation of aneuploidy.

There are still several typos, which need to be corrected.

The authors should cite Shukken&Sheltzer, 2022, also for their discovery of increased abundance of potential ubiquitination sites in dosage-compensated proteins, and Donnelly et al, 2014, for their discovery of increased proteasome abundance and activity in human aneuploid cells.

I recommend publication of the manuscript in its current form.

Referee #2 (Remarks to the Author):

This paper now makes its argument more convincingly to me. I think the revisions have been effective. Two significant points that I would mention: First, there is a running argument in the paper about levels of ubiquitination and ubiquitination site usage. (In one spot called "potential" ubiquitination sites--as opposed to observed??) Obviously this is a key signature of degradation pathways and deserves the attention given to it. I may have missed it, I did look quite a bit, but I had trouble locating a concrete description of what measurements had been made. Even the methods section was unclear on this point. By what criterion were proteins assigned as ubiquitinated? Was that done at the protein or peptide level? The authors might have followed the signature mass of -GlyGly residuals following trypsinization (on the SILAC samples??), but if so that should be spelled out. One reason to note the method, if that is how it was done, is that it's not the most sensitive, you would have to do GG pulldowns to get better read depth (but quite difficult to do....).

The second question concerns Fig 4g. I personally doubt that it helps the paper as there are too few

points to ensure that the line that was fit to the data reflects a true trend.

Minor points of English....."induction of the ubiquitin proteasome (101)" how about "induction of the ubiquitin proteasome system" "data" should be plural..... "both collections differ" (80).....if I take the meaning it should read "the two collections differ".....

Referee #3 (Remarks to the Author):

This is a second revision of the manuscript by Muenzner et al. The article is much more coherent and covers an impressive amount of work. The authors have also adequately addressed all my concerns, and I recommend this article for publication. The authors should double-check the following minor points.

I am confused about why the distribution of the Log₂ fold change is so broad for the disomic stain in Fig 3C (left panel). In Figure 2f, the distribution of the changes appears much tighter for the lab disomic strain in comparison to the natural isolate. The authors should double-check the scale.

It is unclear how Fig S6h-j are related to RNF4. Additional information should be added in the legend, or the figures should be reassigned to a different statement in the text.

I wonder whether strains with higher proteasome levels also show faster turnover and/or attenuation. Similarly, could that be related to the trends of expression signature for ESR/APS/CAGE that are either up or down.

Author Rebuttals to Second Revision:

Comments to reviewers

Referee #1 (Remarks to the Author):

The manuscript 2022-04-05410B by Ralser and colleagues with the title: The natural diversity of the yeast proteome links aneuploidy tolerance to protein turnover has been improved significantly in this second revision. I particularly appreciate the shorter, more compact and focused text, clearer figures and an interesting discussion without overly general statements or unnecessary digressions. While the authors do not provide clear molecular mechanisms, they are three important outcomes of their work:

a) naturally occurring aneuploids showed increased dosage compensation on protein level, likely to mitigate the negative effects of the aberrant chromosome numbers;

b) this dosage compensation is associated with increased proteasome - mediated protein turnover;

c) the provided datasets are unique and will serve the scientific community in many different fields.

The two findings will result in novel testable hypotheses which will contribute to better understanding of regulation of protein abundance in cells, and to adaptation of aneuploidy.

We thank the reviewer for working on the manuscript and for the concise summary of the main advances of the manuscript.

There are still several typos, which need to be corrected.

We apologize for the typos that escaped our attention. We have carefully reviewed the revision.

The authors should cite Shukken&Sheltzer, 2022, also for their discovery of increased abundance of potential ubiquitination sites in dosage-compensated proteins, and Donnelly et al, 2014, for their discovery of increased proteasome abundance and activity in human aneuploid cells.

We have included both suggested references.

I recommend publication of the manuscript in its current form.

We thank the reviewer for their constructive comments and input throughout the review process.

Referee #2 (Remarks to the Author):

This paper now makes its argument more convincingly to me. I think the revisions have been effective.

We thank the reviewer for working on the manuscript and for the helpful input provided throughout the revision process.

Two significant points that I would mention: First, there is a running argument in the paper about levels of ubiquitination and ubiquitination site usage. (In one spot called "potential" ubiquitination sites--as opposed to observed??) Obviously this is a key signature of degradation pathways and deserves the attention given to it. I may have missed it, I did look quite a bit, but I had trouble locating a concrete description of what measurements had been made. Even the methods section was unclear on this point. By what criterion were proteins assigned as ubiquitinated? Was that done at the protein or peptide level? The authors might have followed the signature mass of -GlyGly residuals following trypsinization (on the SILAC samples??), but if so that should be spelled out. One reason to note the method, if that is how it was done, is that it's not the most sensitive, you would have to do GG pulldowns to get better read depth (but quite difficult to do....).

We apologize that the description of the ubiquitination data analysis and experiments was cut too short when we prepared the second revision. The manuscript contains two analyses, the first analysis uses the ubiquitination site annotation of the yeast proteome from Uniprot (<https://www.uniprot.org/>) which was compiled from both experimental and computational sources. We found, similar to human cell lines ¹, that ubiquitination is a strong predictor of attenuation (Fig. 2e). The second analysis was conducted using mass spectrometric *Gly-Gly* profiling, using a method that we (M. Steger, M. Ralser) published earlier ². The results obtained show, in natural aneuploid strains, a stoichiometric increase in the ubiquitination of proteins encoded on aneuploid chromosomes (Extended Data Fig. 7a, b). We have now added to the methods sections, as well as to the figure legends, by describing both analyses. We hope that these edits make the procedures much clearer to the reader.

The second question concerns Fig 4g. I personally doubt that it helps the paper as there are too few points to ensure that the line that was fit to the data reflects a true trend.

The number of data points for the AGE2 example corresponds to the number of respective aneuploid (bottom panel) or euploid isolates (top panel) in the collection. We agree that a higher number of isolates would have further increased the statistical robustness of the observed correlation in the aneuploids. However, the trend for this protein is consistent with the overall picture obtained across all aneuploids and proteins. We have now moved the previous Fig. 4g to the supplement (now Extended Data Fig. 8c) as suggested by the editor.

Minor points of English....."induction of the ubiquitin proteasome (101)" how about "induction of the ubiquitin proteasome system"....."data" should be plural....."both collections differ" (80).....if I take the meaning it should read "the two collections differ".....

We thank the reviewer for spotting these language inconsistencies, which are now corrected.

Referee #3 (Remarks to the Author):

This is a second revision of the manuscript by Muenzner et al. The article is much more coherent and covers an impressive amount of work. The authors have also adequately addressed all my concerns, and I recommend this article for publication. The authors should double-check the following minor points.

We thank the reviewer for the constructive review process and helpful input, which certainly improved the manuscript.

I am confused about why the distribution of the Log2 fold change is so broad for the disomic stain in Fig 3C (left panel). In Figure 2f, the distribution of the changes appears much tighter for the lab disomic strain in comparison to the natural isolate. The authors should double-check the scale.

We confirm that the plots are correct as presented. In short, the effect sizes appear different, because in Fig. 2f (now Fig. 3a) we show analysis of data from a single strain, while in Fig. 3c (now Fig. 4c) we display the average values across nine disomes or 95 aneuploid isolates, respectively. Averaging across many

strains explains the (expected) “tighter” appearance of the distribution in Fig. 3c (now Fig. 4c) for the natural isolates vs. the lab aneuploids (compare also Extended Data Fig. 5d).

It is unclear how Fig S6h-j are related to RNF4. Additional information should be added in the legend, or the figures should be reassigned to a different statement in the text.

These results, analyzing whether chromosome IV aneuploids exhibit particular induction of the proteasome, were included for the second revision in response to a reviewer question. We have now expanded the figure legend to start with “*RPN4 is located on chromosome IV. We assessed whether natural isolates carrying a chromosome IV aneuploidy show particularly prominent induction of the proteasome due to increased gene dosage of RPN4.*” to better explain the context of the three panels.

I wonder whether strains with higher proteasome levels also show faster turnover and/or attenuation. Similarly, could that be related to the trends of expression signature.

A related analysis was part of the manuscript, but was not highlighted clearly; we thank the reviewer for making this evident to us. We do report that levels of the ubiquitin-proteasome system correlate with increasing turnover in euploid isolates. This is now highlighted better on lines 244-249. The data is provided in Extended Data Fig. 8d.

1. Schukken, K. M. & Sheltzer, J. M. Extensive protein dosage compensation in aneuploid human cancers. *Genome Res.* **32**, 1254–1270 (2022).
2. Steger, M. *et al.* Time-resolved in vivo ubiquitinome profiling by DIA-MS reveals USP7 targets on a proteome-wide scale. *Nat. Commun.* **12**, 5399 (2021).

Reviewer Reports on the Third Revision:

Referees' comments:

Referee #2 (Remarks to the Author):

My comments have been addressed and I have no further comments.